# Understanding ultrafast free-rising bubble capturing on nano/micro-structured super-aerophilic surfaces

Yue Hu[1], Zhenbo Xu[2], Haotian Shi [1], Benlong Wang [1,3], Liqiu Wang [4] ✉ & Lu-Wen Zhang [1] ✉

Rapid bubble capture is essential for collecting targeted gaseous media and eliminating floating impurities across aquatic environments. While the role of nanostructures during the collision of free-rising bubbles with super-aerophilic surfaces is well established, the fundamental contribution of microtextures in promoting initial capture, even before contact, has yet to be fully understood. We report the rising bubble-induced large deformation of the entrapped gas layer, rapidly thinning the liquid film to its rupture threshold and thus achieving an ultrafast bubble capture down to about 1 ms with an array of microcones, decorated with nanoparticles as a convenient example to obtain super-aerophilicity. This rapid capture is also very stable due to the hysteresis movement of three-phase contact lines that inspired a critical pressure criterion for ensuring gas-layer stability and capture efficacy. The present nano/microstructured surface supports prolonged, loss-free gas transport in challenging shear flow as well, providing robust bubble control strategies for diverse systems.

Free-rising bubble capture, transcending its origins in nature processes such as the respiration of diving insects[1] and gas exchange in aquatic plants[2], has solidified its importance across an expansive range of industries, including ocean methane capture[3,4], drag reduction[5,6], energy harvesting[7], and water treatment[8]. Significant bubble adsorption is achieved on aerophilic surfaces by combining rough textures with low surface energy materials to form gas layers within characteristic structures[9,10], distinguishing them from hydrophobic surfaces that are non-wetting but not necessarily absorb bubbles[11]. Micro/nano-scale textures, such as concave pits[12], asperities[13,14] or random porous structure[15,16], greatly enhance aerophilic surfaces to exhibit super-aerophilicity, allowing for rapid capture of rising bubbles on first contact, rather than bouncing or escaping.

From release at a sufficiently large distance to full absorption into the plastron on surfaces in quiescent liquid, a free bubble's life unfolds through distinct stages: fluid dynamic dominated early approach[17–19], a decelerated motion phase culminating in collision with or without bouncing[20,21], colloidal forces involved rupture[22,23], and absorption at equilibrium[24,25]. From a system-level perspective, this process encompasses liquid film drainage, three-phase contact line formation, and contact line spreading, respectively[24,26], reflecting the interplay between the bubble dynamics and surface morphology. Among these, accelerating drainage of the thin film is the most crucial for enhancing capture efficiency, as this process can vary by two orders of magnitude across diverse surfaces even in pure water, and by up to 5~6 orders of magnitude due to both the liquid composition and the physicochemical property across surface[27,28]. Therefore, we define and focus on the capture time as the interval from the bubble's initial contact with the surface to the formation of a spreadable three-phase contact line[24,29], and also examine the

[1]Department of Engineering Mechanics, School of Ocean and Civil Engineering, Shanghai Jiao Tong University, Shanghai 200240, China. [2]Department of Architecture and Civil Engineering, City University of Hong Kong, Hong Kong 999077, China. [3]Key Laboratory of Hydrodynamics (Ministry of Education), School of Ocean and Civil Engineering, Shanghai Jiao Tong University, Shanghai 200240, China. [4]Department of Mechanical Engineering, The Hong Kong Polytechnic University, Hong Kong 999077, China. ✉e-mail: liqiu.wang@polyu.edu.hk; lwzhang@sjtu.edu.cn

capture stability, quantified by the coefficient of variation in capture time.

The capture time can be dramatically reduced by evolving surface design strategies. Compared to bare hydrophobic surfaces modified only with nanoparticles, where capture times range from 10 to 100 ms[26], a breakthrough bubble capture in 10 ms was achieved on micro-nano structured artificial lotus leaf surfaces[30]. Further optimization of microstructure dimensions beyond the surface slip length has reduced the capture time to less than 5 ms[21]. Employing diverse techniques such as electrodeposition, femtosecond laser, and liquid flame spray to refine nanostructures between 60 and 200 nm[13,15,31], has further shortened the capture time to a current minimum of 2 ms.

As a key strategy to enhance bubble capture, refining and miniaturizing surface structures is widely recognized. Nanoprotrusions increase gas fraction and introduce nano-slip boundaries that accelerate fluid flow in liquid films, a crucial mechanism for fast capture[32]. This has led to efforts to even smaller structures, aiming to maximize local Laplace pressure differentials and further expedite liquid film rupture[29,31]. However, nanoscale structures primarily contribute at or after the moment of contact, governing only the instant rupture and absorption during the bubble's rise, leaving the decelerated approach and fluid dynamics dominated phases unaffected. Presumably, this explains why despite nanoscale dimensions being reduced by 1000 times compared to microscale structures through various complex fabrication processes, the reduction in capture time in general[13,21] has not shown a proportionally significant decrease. An opposing trend, even, exists where hydrophobic microprotrusions can capture free-rising bubbles (with an approach velocity of ~0.35 m/s) an order of magnitude faster than the nanoparticle-decorated surfaces[21]. Moreover, the challenges of maintaining uniformity and addressing complex interfacial effects in large-scale fabrication limit the effectiveness and extent of size reduction of nanoscale structures, and atomically precise engineering on large surfaces remains impractical.

Upon reevaluating microstructures, which share similarities with nanostructures in terms of slip effects, TPL pinning, and other factors, likely exert pronounced effects throughout the rising bubble capture process. Key features such as thicker air mattress[33], rougher textures[24], and larger gas storage space[34] in comparison with nanostructures, particularly influence the critical drainage process before film rupture and absorption. We believe that the role of micrometer structures in promoting free-rising bubble trapping offers much space for exploration. The internal drainage state between micro-protrusions— the smallest characteristic units of the surface—is unclear. Particularly, how the multi-phase interfaces confined between microscale asperities interact with bubble dynamics remains largely unexplored. Non-strategic design results in notable fluctuations in capture times across these periodically structured surfaces. Such uncertainty in capturing further impedes the efficiency of aerophilic surface applications in practical environments such as complex flows.

We aim to elucidate how basic yet ubiquitous conical structures can maximize ultrafast free-rising bubble capture on super-aerophilic surfaces. Inspired by the Salvinia leaves, the microcones capitalize on the distinctive scale of trichome assemblies, hundreds of microns in size (Fig. 1a). Such features are observed to facilitate rapid bubble capture upon initial contact with the Salvinia leaves (Supplementary Movie 1). Despite the variability in the fine apex structure across different species, they all demonstrate high efficacy in bubble capture, achieving times below 2.5 ms (detailed data shown in Supplementary Fig. 1). This consistent performance motivates a deeper investigation into the universal mechanisms underlying bubble entrapment by microstructures.

## Results
### Ultrafast bubble capture
We achieve robust ultrafast capture of free-rising bubbles on a model surface by integrating basic conical structures of a hundred-microns in size, and enhancing aerophilicity through nanoparticle coating (see Methods). This surface with microconical super-aerophilic (MA) arrays is fabricated through 3D printing, where optimally configured microcones are arranged in a square lattice with a center-to-center distance of $L = 1$ mm, a base radius of $a = 346$ μm, and a height of $b = 600$ μm (Fig. 1b, c). MA surface features ~2 μm roughness (detailed morphology characterization in Supplementary Fig. 2), which is two orders of magnitude lower than the microcone size. Bubbles (dimensionless diameter $D_0/L = 2.4$, where $D_0$ is the initial diameter of bubble, inset of Fig. 1e) released from 2 cm beneath the surface are rapidly absorbed upon first contact with the MA surface, reaching an exceptional capture time ($t_c$) of 1.5 ms (Fig. 1d), a reduction of over 47 times from the 71.5 ms observed on a flat hydrophobic (FH) surface modified with nanoparticles only, where the bubble rebounds four times upon first impact (Supplementary Movie 2). This difference in capture time magnitude is minimally influenced by further varying releasing distances from 60 mm to 5 mm, resulting in bubble approach velocity ranging from 0.37 to 0.16 m/s (Supplementary Fig. 3a). This marked reduction in capture time demonstrates that simple hundred-micron conical protrusions with plastron are sufficient to ensure ultrafast trapping, even in the context of continuous bubble capture (Supplementary Fig. 4) and on the tiled MA surface (Supplementary Fig. 5), augmented by the synergistic effects of nanoparticles. In this process, nanoparticles serve to provide expedient capture sites by stabilizing the gas layer and elevating nano-local pressure differentials, yet they cannot trap rising bubbles upon the first contact independently. We also show that nanoparticle-free but super-aerophilic silanized microstructured surfaces achieve similar rapid capture performance ($t_c = 2.1$ ms ± 0.6 ms, Supplementary Fig. 6a), yet they are more susceptible to hydrostatic pressure, which may destabilize the gas layer and prevent full bubble spreading (Supplementary Fig. 6b). As a result, we further focus on MA surfaces decorated with nanoparticles as a model surface, thereby avoiding such interference with gas layer loss.

The MA surface also exhibits remarkable stability and robustness, consistently achieving an ultrafast capture time $t_c$ (average 1.5 ms) across the broad $D_0/L$ range of 1.6–3.5 (Fig. 1e and Supplementary Fig. 7), with a coefficient of variation of 13.7% in $t_c$. One-hundred repeated capture events at a fixed $D_0/L$ on the same location also indicate a slight fluctuation of <10% (Supplementary Fig. 8). On the contrary, $t_c$ with the FH surface varies over a much wider range, from 63.6 ms to 132.6 ms, due to prolonged bouncing before bubble rupture. Compared with six types from eighteen reported aerophilic surfaces[13,16,21,26,29,30,35–38] (the bubble approach velocity mainly ranges from 0.37 to 0.14 m/s), most of which primarily focus on controlling nanoaggregate sizes and shapes through intricate bottom-up fabrication to form hierarchical structures with characteristic size $S_c$ ranging 10–200 μm, typically fluctuating up to 21 times in capture time on the same surface[13], our model surfaces achieve the extremum of minimum capture duration ($t_{min} = 0.8$ ms). Furthermore, they demonstrate consistent capture time with only up to 2 times difference across the widest range of roughness ratio ($S_c/S_{min}$) (Fig. 1f), a performance derived from leveraging primary features from Salvinia leaves' trichomes and utilizing a straightforward manufacturing process. These comparisons suggest that capture time does not strictly diminish with the reduction in texture size, calling for a thorough understanding of the mechanism that enables basic conical textures to achieve high trapping efficiency to fully excavate their potential.

### Enhanced water film drainage within a characteristic unit
How does the liquid-gas interface evolve in a single characteristic unit? This is crucial as it influences the liquid film drainage process during free-rising bubble capture. Utilizing confocal microscopy, we clearly observe the liquid-gas interfaces (LGIs) and the three-phase contact lines (TPLs) formed as gas layer contacts the cones within a typical microconical unit (Fig. 2c). These interfaces evolve to collectively

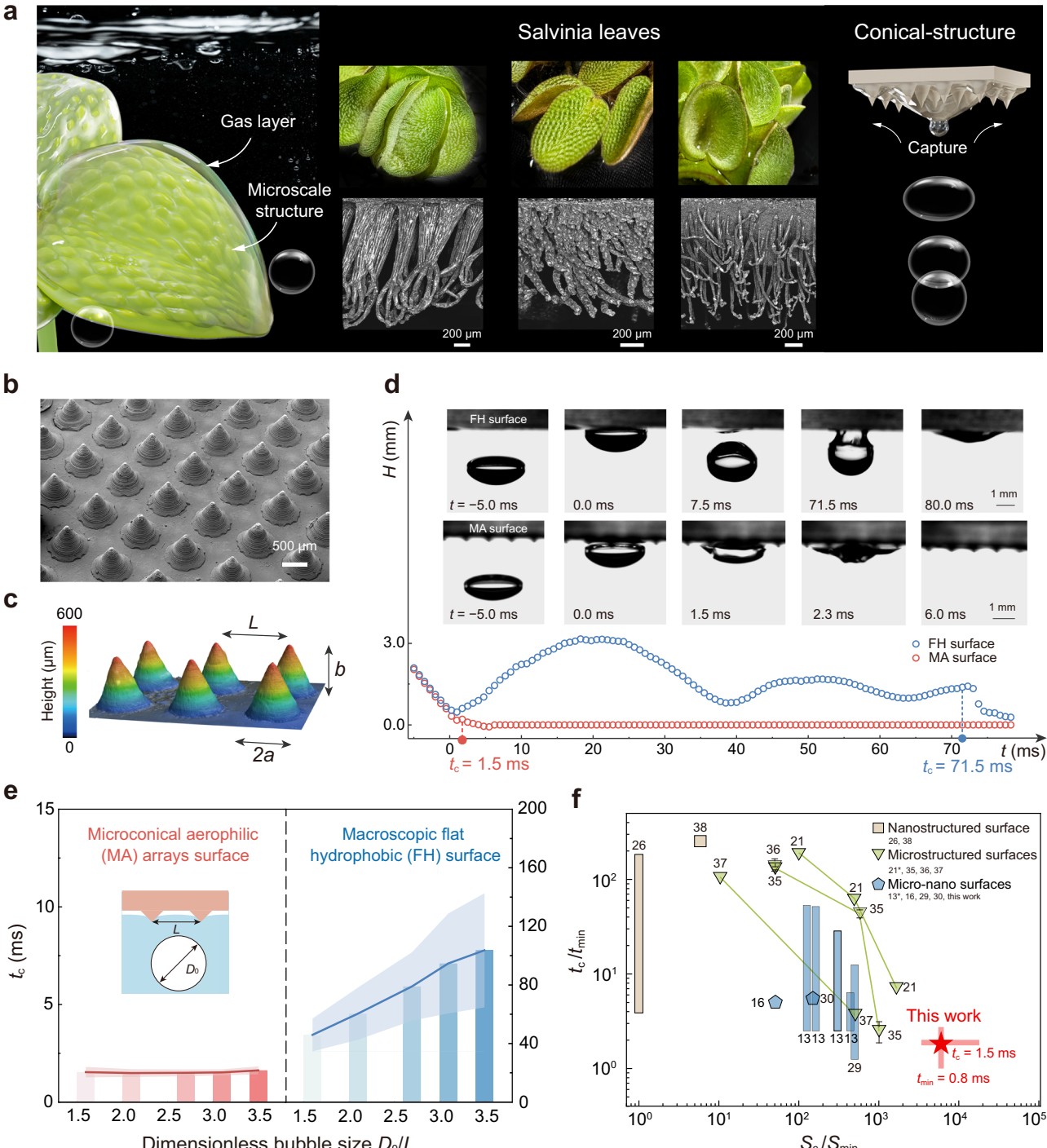

**Fig. 1 | Ultrafast bubble capture on MA surface. a** The design of ultrafast bubble capture surface is inspired by the Salvinia leaves, which feature hundred-micron structures and show excellent bubble trapping efficiency. **b** A scanning electron microscope (SEM) image of the conical-structured surface. **c** Height map from a laser confocal microscope and structural characteristics of the microcones. **d** The bubble centroid $H$ evolutions versus time $t$. The inserted images display capture behaviors when bubbles ($D_0/L = 2.4$, where $D_0$ is the initial diameter and $L = 1\,\text{mm}$ represents the distance between the two cones) released 2 cm from the test surfaces. **e** Capture time $t_c$ with varying sizes of bubble from $D_0/L = 1.6$ to 3.5 on the MA and FH surfaces. The insert illustrates the initial bubble radius $D_0$ and the conical distance $L$. Error bands represent standard deviation calculated from 20 independent experiments, indicating larger uncertainty in $t_c$ on the FH surface than that on the MA surface. $t_c$ on MA surface ranges in 1.5 ms–1.6 ms with an average 1.5 ms ±

0.1 ms. On FH surface that ranges in 45.9 ms–103.8 ms with an average 76.8 ms ± 21.4 ms. **f** Dimensionless capture time $t_c/t_{min}$ among the reported structural aerophilic surfaces with varied roughness ratio $S_c/S_{min}$. $S_c$ is the maximum characteristic structural size on each surface and $S_{min} = 0.1\,\mu\text{m}$ is the minimum characteristic size among the all. Each long bar represents the range of capture time on the same surface. Each error bar shows the reported measurement error. Our work shows a record-setting minimal time $t_{min} = 0.8\,\text{ms}$, a stable average $t_c$ of 1.5 ms, and the time bars covering wide parameter ranges $S_c/S_{min} = 3.4 \times 10^3$–$1.8 \times 10^4$ and $D_0/L = 1.6$ to 3.5. Star labels (*) represent raw data that cannot be transformed to the capture time defined in this work. All the raw data are provided in Supplementary Table 1. A comparison of detection methods on capture time is summarized in Supplementary Table 2.

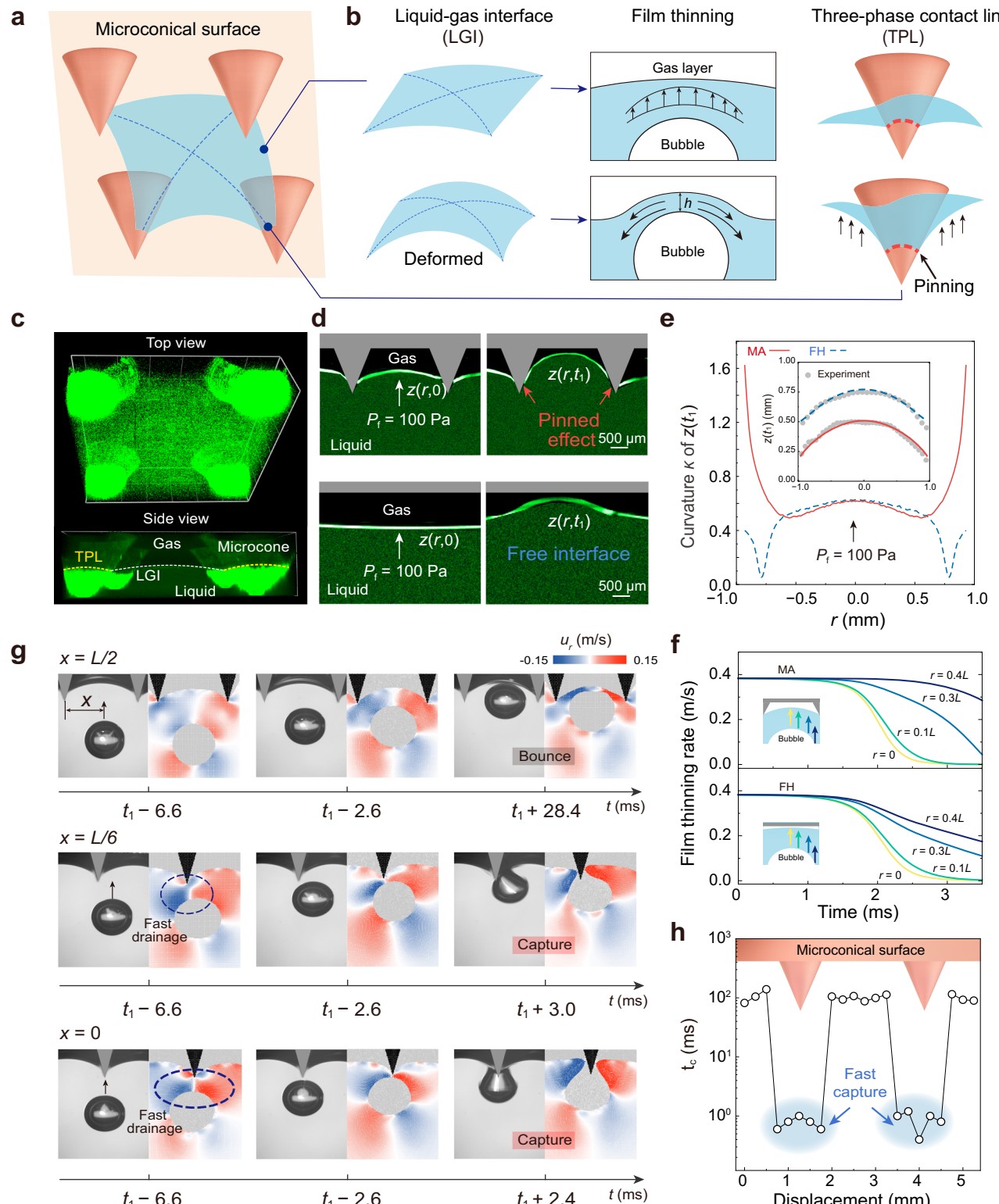

**Fig. 2 | Mechanism of fast capture within microstructure unit. a** Scheme of the interface formed within a unit of four microcones. **b** Illustrations of the drainage process show that the LGI deforms when bubbles approaching accompanied with film thinning, while the TPL pins on the microstructures. **c** Confocal fluorescence images of the conical microstructured surface when immersed underwater with clear LGI and TPLs. **d** The fluorescent reflection images of the LGI motion on the MA ($b = 1.2$ mm and $L = 2.5$ mm) and FH surfaces. The interface dynamics is driven by bubbles ($D_0 = 1.6$ mm) approaching with the averaged film pressure equals around 100 Pa. **e**, The theoretical predictions of the curvature $\kappa$ of the ultimate interface profiles on the FH and MA surfaces. The inset shows that theoretical estimations of

the ultimate interface profiles $z(t_1)$ exhibit high consistency with the experimental results extracted from the fluorescent-reflecting images in **d**. **f** The film thinning rate $\partial h/\partial t$ predicted by our model at locations $r = 0 - 0.4\,L$. **g** High-speed image sequences of bubbles approaching at three horizontal distances to the conical tip, with horizontal velocities $u_r$ map obtained by SPH simulations showing large$_r$ $u_r$ when bubble released closer to TPLs. **h** The capture time $t_c$ under several displacements with a horizontal increment of 200 µm, showing fast capture regions adjacent to the TPLs. The zero displacement corresponds to the tip of the first protrusion.

reflect alterations of surface states induced by the bubble's ascent until captured (Fig. 2a, b, Supplementary Movie 3). At the initial state ($t = 0$), the deformable gas layer inside the microstructure is anchored by the TPLs, with liquid subtly infiltrating and forming a meniscus at the LGI. After releasing a bubble ($D_0 = 1.6$ mm) between protrusions spaced 2.5 mm apart and just before contacting the gas layer ($t = t_1$), the interface exhibits a notable curvature increase under the rigid constraint of the TPLs. The interface contours from the fluorescence signal enable a straightforward recognition of these changes (top two images in Fig. 2d). In contrast, on the FH surface, the LGI experiences a noticeable migration upwards with a slight deformation as the bubble approaches, due to the gas film initially forming higher than the nanoprotrusions (Supplementary Movie 2) and the absence of pinned contact lines, indeed resembling a free liquid surface (bottom two images in Fig. 2d).

To quantify the gas layer deformation and assess its impact on film thinning dynamics, we develop a model of liquid-gas interface dynamics incorporating the TPL pinning effect based on Stokes-Reynolds-Young-Laplace equation[18,39,40]. The interface profiles, including gas layer profile $z(r, t)$ and the bubble surface profile $z_b(r, t)$ are used to determine the thickness of liquid film $h = z - z_b$, which is subsequently used to derive the film pressure $P_f$ within liquid film, expressed as $P_f = \frac{2\sigma}{D_0} + \frac{\rho g z}{2} - \frac{\sigma}{2r}\frac{\partial}{\partial r}\left(r\frac{\partial h}{\partial r}\right)$, where $\rho$ is the liquid density, $g$ is the acceleration of gravity, $\sigma$ is the liquid-gas interfacial tension. Then the drainage dynamics of the liquid film can be described by determining the drainage rate based on the lubrication theory[40], i.e., $\frac{\partial h}{\partial t} = \frac{1}{3\mu r}\frac{\partial}{\partial r}\left(rh^3\frac{\partial P_f}{\partial r}\right)$, where $\mu$ is the liquid viscosity (detailed solving procedure is shown in Supplementary Note 1 and 2).

The ultimate interface profiles $z(r, t_1)$ from the theoretical analysis demonstrate high consistency with experimental measurements (Fig. 2e). TPLs pinning of the gas layer diminishes the peak displacement of liquid-gas interfaces by 34% at the film center (inset). The curvature $\kappa = |z''|/[1 + (z')^2]^{3/2}$ for $z(r, t_1)$ increases up to four times of that with the FH surface near the film boundary ($r = 0.4\,L$), subjecting the liquid film to greater compressive forces and accelerating the film thinning. Furthermore, the thinning rate $\partial h/\partial t$ near the bubble basal perimeter peaks at 1.6 times that of the free interface, owing to passive retention from TPLs (see Fig. 2f, Supplementary Fig. 9 for the detailed spatiotemporal evolution of film thickness and Supplementary Fig. 10 for a case involving a larger bubble). This fast thinning ensures a prerequisite for liquid film rupture upon first contact, thereby obviating a prolonged bouncing process and substantially reducing capture times by 10 to 100 times. Building on the results from Fig. 2e, f, we unveil that the interdependent increase in curvature of the gas layer and the rate of liquid film thinning within the minimum characteristic unit is the fundamental mechanism underlying bubble entrapment.

The extent to which the mechanism of accelerated film drainage is affected by bubble release distance and approach velocity warrants further consideration. For the free-rising bubbles discussed in both our study (approach velocity $v = 0.37-0.16$ m/s) and the cases presented in Fig. 1f, the release distance generally ranges from 300 to 2.4 mm, with corresponding $v = 0.37-0.14$ m/s (Supplementary Table 1). Within these comparable velocity ranges, bubbles have enough time and space to deform, generating sufficient pressure that accelerates liquid film thinning, all driven by non-contacting microstructures. When bubbles are released over a very short distance with an ultra-low approach velocity of around 0.04 m/s (Supplementary Fig. 3b), MA surface maintains a short capture time of ~1.7 ms, while on FH surfaces, bubbles no longer rebound, resulting in a capture time decreasing to the same order of magnitude as that on the MA surface, albeit with greater variance. However, as release distance is further reduced beyond the above range, the effectiveness of microstructures in driving drainage may diminish. Notably, ultra-short release distances have a non-negligible probability of introducing uncontrolled factors, such as gas injection speed and needle tip adhesion, necessitating cautious

interpretation of these results. Another scenario involves needle-fixed bubbles[23,41]. While these can also produce very low velocities, the bubble dynamics differ fundamentally from free-rising bubbles, making direct comparison within the same theoretical framework inappropriate. This refined needle-fixed bubble setup is particularly well-suited for analyzing instantaneous rupture upon surface contact[23].

Interestingly, we observe a further increase in liquid film drainage rate by manipulating the bubble release position. Figure 2g illustrates a shift from bouncing to rapid captured as bubbles are released from the center of the unit ($x = L/2$) to directly beneath the cone tip ($x = 0$). This improved capture efficiency mutually reinforces the accelerated horizontal drainage velocity field $u_r$ and a 90.4% ± 18.5% increase in the maximum velocity (cloud plot in Fig. 2g and Supplementary Fig. 11) produced from our Smoothed Particle Hydrodynamics model (see Methods and Supplementary Note 3). The enhanced drainage rate stems from the highly compressed liquid film at the pinned boundaries, restricted by the constrained upward movement of the interfaces (see the interfacial deformations under three release locations in Supplementary Fig. 12). We highlight a sharp reduction in capture time $t_c$ from ~100 ms to ~1.5 ms near TPLs at positions in 200 μm horizontal increments (Fig. 2h). This observation, with asymmetric capture patterns mainly in bubbles released by slightly offset cone tips (Supplementary Fig. 13), shows that the gas bridge tends to form at the specific positions where the bubble meets the TPL. At these precise points, the liquid film thins to a local minimum, a critical thickness near 100 nm[29]. Within the range, the liquid film adjacent to nanoparticles generates attractive disjoining pressure[42,43], leading to spontaneous rupture and subsequent coalescence between the bubble and the gas layer. These findings show the necessary condition for consistent capture efficacy: rising bubbles must contact with the stationary pinned TPLs formed by the encapsulated gas and microstructure, located on the inner wall. At the moment of rupture, the effect of nanoparticles or nanostructures remains consistent across both MA and FH surfaces due to their comparable minimum feature sizes. On the other hand, during the drainage phase before rupture, the microstructure irreplaceably contributes to confining the gas to a constrained space, within which significant deformation of the gas layer can occur, thereby accelerating drainage.

## Threshold criteria for bubble capture

Insights into the gas layer state within a single unit raise a new question: what are the coupled dynamics between multi-anchored gas layer across several units and large, significantly deformed bubbles? This coupling introduces considerable uncertainties in capture behavior, thereby guiding us toward the optimization of the microstructures. We employ particle image velocimetry (PIV) to characterize flow field structures near the microconical array surfaces (Supplementary Fig. 14), suggesting that fluid between the bubble and microcones is expelled outward as the bubble rises, forming a stronger side vortex ring and wake vortex at a larger cone height-to-radius ratio of $b/a = 3.5$ compared to 0.3, notably enhancing velocity field below the bubble (Fig. 3a and detailed velocity and vorticity in Supplementary Fig. 15). This enhancement stems from increased drainage rates within the cone array above the bubble, where the fluid velocity $u$ of outward flow in the liquid film is influenced by cone geometry. In contrast, on the FH surface featuring only nanoparticles, bubbles' approach velocity is reduced, hindering their touch with the TPLs upon first contact. Elongated microcones, with a higher $b/a$, not only create a thicker gas layer that expands deformation space and enhances interface curvature, thus accelerating drainage, but also increase the bubble's maximum velocity $V_m$ before contact (Fig. 3b), contributing further to the overall thinning of the liquid film.

We can gain a clear understanding of how conical structural variations affect liquid film evacuation from energy conservation during the bubble's ascent. This analysis involves assessing the efficiency of

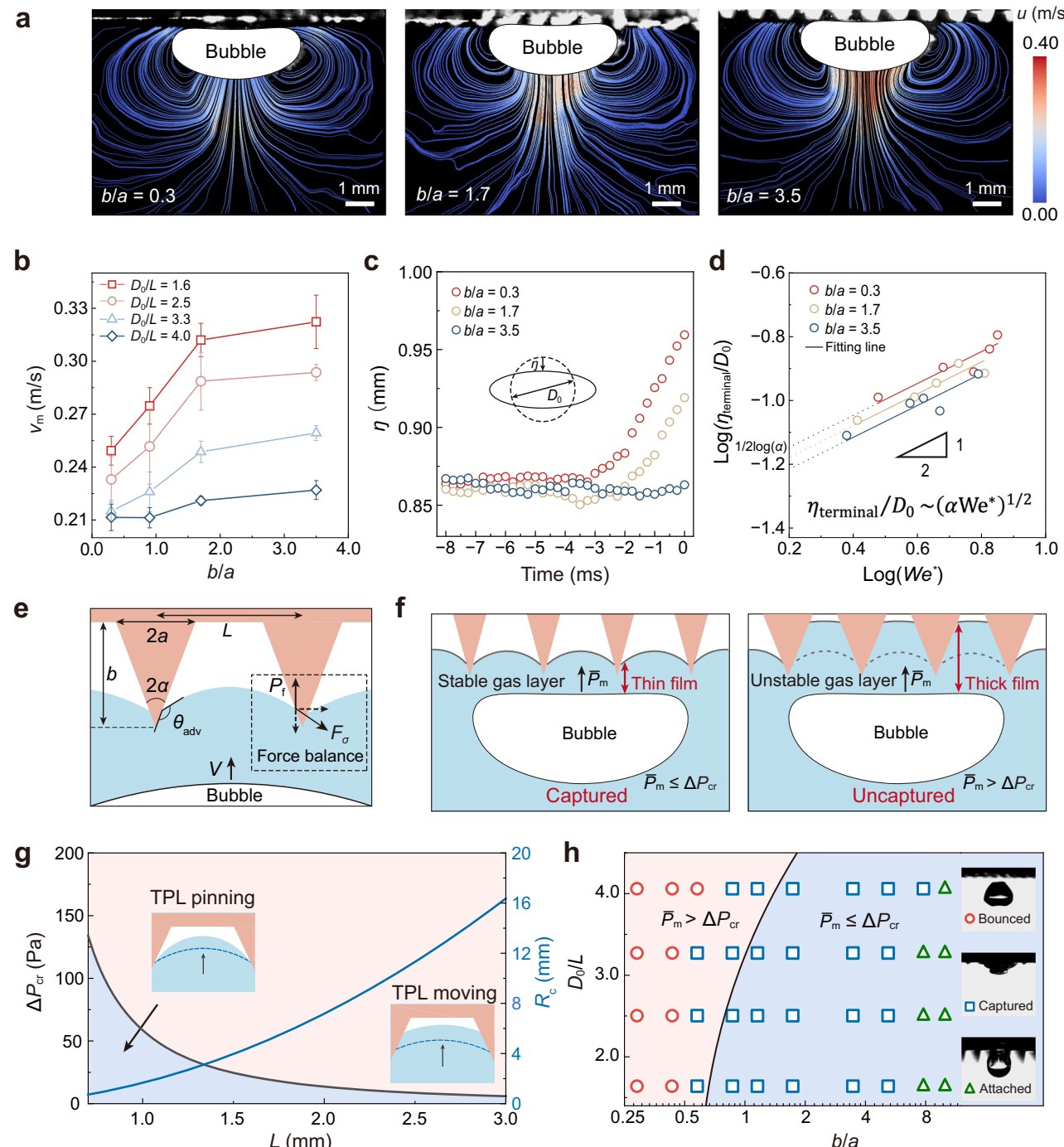

**Fig. 3 | Interactions between bubbles and gas layer on MA surfaces. a** The streamlines colored in the velocity magnitude around a bubble (dimensionless equivalent bubble diameter $D_0/L = 3.0$) approaching the MA surface with $b/a$ increasing from 0.3 to 3.5 obtained by particle image velocimetry tests. **b** Maximum velocities $V_m$ before bubbles decelerate as a function of the conical geometric parameter $b/a$. Error bars represent standard deviation calculated from three independent experiments. **c** The elongation parameter $\eta$ versus time $t$ during the approaching process on MA surfaces with varying $b/a$ ratios. The inset shows the change in bubble radius after deformation. **d** Correlation of the dimensionless terminal deformation $\eta_{terminal}/D_0$ as a function of $We^* = C_m\rho D_0 V^2_m/\sigma$. $\eta_{terminal}$ is the terminal deformation of the bubble upon surface contact, and $C_m$ is the added mass coefficient. **e** Force analysis of the TPLs pinning on the microstructures. **f** The schematic diagram of the gas layer stability. A stable gas layer under film pressure ensures the formation of thin film that facilitate the fast bubble capture. **g** The relationship of the conical distance $L$ and the $\Delta P_{cr}$ under $b/a = 1.7$, predicted by the augmented Young-Laplace equation (Supplementary Note 5). **h** Phase diagram summarizing the dimensionless bubble size $D_0/L$ and the $b/a$ required for the efficient capture. Each case is repeated twenty times and the fast-capture is considered successful if the bubble is captured on the first contact with the surfaces for all cases. The black line separating the cases of $\bar{P}_m \leq \Delta P_{cr}$ and $\bar{P}_m > \Delta P_{cr}$ is obtained by theoretical evaluation.

kinetic energy conversion to surface energy of the bubble. In this process, bubble deformation and the associated change in liquid film area are pivotal. A large $b/a$ ratio induces the reduced bubble deformation $\eta$, defined as the difference between the deformed ellipsoidal bubble's short semi-axis, and its equivalent radius, indicating a diminished conversion of kinetic energy to bubble surface energy while a larger proportion diverted to the gas layer's surface energy (Fig. 3c). We establish a scaling law that connects the dimensionless

terminal deformation of bubble $\eta_{terminal}/D_0$ with energy conversion efficiency $\alpha$, defined as $\eta_{terminal}/D_0 \sim (\alpha We^*)^{1/2}$, where $We^* = C_m\rho D_0 V_m^2/\sigma$ (see details in Supplementary Note 4). This relationship is confirmed in Fig. 3d, where decreasing y-intercepts, marked as $1/2 \log(\alpha)$, illustrate a decrease in $\alpha$ with a larger $b/a$, associated with pronounced bubble deformation and smaller basal diameter $D_b$ (evolution of $D_b$ can be seen in Supplementary Fig. 16), consequently limiting the area of liquid film accessible to the bubble. Notably, the rate of film thinning, $dh/dt$, and the liquid film radius, $R_b = D_b/2$, are inversely related as $-dh/dt \propto R_b^{-2}$, according to the classical Scheludko equation[44]. Therefore, a smaller $D_b$ significantly accelerates film thinning. On a global scale, energy conversion within this multi-interface system, particularly the dynamic interaction among bubble, liquid film, and gas layer, serves as a critical mechanism through which periodic microconical arrays facilitate drainage and enhance bubble capture. Within each unit, this process is driven by the synergetic effects at the liquid-gas interfaces, where we isolate the correlation between deformations in entrapped gas layers and bubble dynamics (shape and speed). At a more localized scale, as discussed in the Fig. 2f, the protruding features of the microstructure dictate the local film thickness, thinning the liquid film at the tips and thereby influencing the drainage dynamics, ultimately affecting the time required for liquid drainage.

Can the gas layer remain consistently stable by pinning three-phase contact lines on multiple microcones? Determining the stability of this highly deformable gas layer under drainage pressure during bubble ascent is crucial. This is quantified by a critical pressure threshold, $\Delta P_{cr}$, which is derived by establishing a mechanical equilibrium between the interfacial tension force and the liquid film pressure exerted on the gas layer (Fig. 3e):

$$\Delta P_{cr} = \frac{\sigma \cos(\pi + \alpha - \theta_{adv})}{R_c}, \qquad (1)$$

where $\alpha$ is half the apex angle of the cone, $\theta_{adv} = 162° \pm 3°$ is the advancing angle determined by the nanocoating's wettability, $R_c = (L^2 - \pi a_c^2)/2\pi a_c$ is the capillary radius of meniscus' spatial curvature[45] and $a_c$ is the radius of the cone section corresponding to the liquid filling depth (see the detailed derivative in Supplementary Note 5). Notably, appropriate microstructural features $\alpha$ and $R_c$ enable the contact angle to undergo greater variability before reaching the critical 162°, thereby amplifying the hysteresis effect and enhancing the stability of the contact line. According to Eq. (1), we define two regions divided by the $\Delta P_{cr}$ curve to ascertain the gas layer's stability as shown in Fig. 3g, varying $L$ while fixing $a/b$ (see the diagram by varying $b/a$ in Supplementary Fig. 17). Below $\Delta P_{cr}$ (blue zone) represents stable gas layer with the pinning TPL, where the liquid film thins rapidly within microstructural units by accelerated drainage, enabling fast bubble capture on initial contact. Above $\Delta P_{cr}$ (pink zone), TPL movement may cause gas layer collapse or even disappear, maintaining a thick liquid film that fails to reach the critical rupture thickness, thus hindering bubble trapping (Fig. 3f).

Based on our theory, we provide the geometric parameter space for microstructure optimization that enables ultrafast capture of free-rising bubbles. We introduce a conservative pressure criterion, $\bar{P}_m \leq \Delta P_{cr}$, where $\bar{P}_m$ is defined as the maximum average liquid film pressure the bubble undergoes its maximum deceleration upon surface contact, variable with microstructure geometry (see the detailed calculation of $\bar{P}_m$ in Supplementary Note 1 and Supplementary Fig. 18). This criterion is represented by a curve within the phase space delineated by dimensionless parameters $D_0/L$ and $b/a$ (black solid line in Fig. 3h), distinguishing a capture zone (blue) where $\bar{P}_m \leq \Delta P_{cr}$ from a non-capture zone (red) characterized by bubble bounce. Extensive statistical evidence robustly correlates our model with experimental results across a broad parameter range ($t_c$ is provided in Supplementary Fig. 19). Observations show rapid captures (blue squares)

predominantly fall within the blue region, while all bouncing events (red circles) are confined to the red zone. Increasing $b/a$ obviously facilitates the rapid capture of larger bubbles; however, too large $b/a$ (>7.8) causes bubbles to suspend over the microstructures, inhibiting capture (green triangles, process illustrated in Supplementary Fig. 20), mirroring observations on other surfaces with high aspect ratio textured structures[30,46]. This failure is due to the liquid-filled gap from the cone tip to the TPLs surpassing the bubble's buoyancy threshold (experimental images in Supplementary Fig. 21). In our identified optimal parameter space, TPL pinning stabilizes the gas layer, ensuring enhanced drainage between the gas layer and ascending bubbles. This significantly boosts the likelihood of reducing the liquid film to its critical thickness upon initial contact.

## Bubble collection in liquid flow

We further demonstrate sustainable gas collection on optimized surfaces in a water flow environment, which represents typical features applicable to gas sensors, drug delivery, and electrode reactions. A flow channel is designed to evaluate efficiency at different flow rates, with controllable gas injection and a real-time gas collection device (Fig. 4a and Supplementary Fig. 22). The MA sample strip patterned with optimized conical arrays ($b/a = 1.7$, $L = 1$ mm as shown in Fig. 4b) is positioned invertedly in the channel. Injected air bubbles attach to the aerophilic sample and are absorbed into the gas layer, which progressively thickens, detaches, and directs into the syringe cylinder (Fig. 4c and released bubble sizes shown in Supplementary Fig. 23). This setup allows precise measurement of gas volume increase (liquid level drop) on the scale (Fig. 4d).

The MA surface showcases ultra-high efficiency in gas collection with increasing Reynolds numbers (Re = 2083–3750), maintaining a collected volume ratio of 100–96.5% (Fig. 4g), whereas that with the FH sample fluctuates between 76.5–39.8%. When Re increases to 5417, MA surfaces' capture rate reduces to 50%, but still 6.25 times larger than FH surface. This significant performance advantage is due to the clear gas channel formed on the MA surface, where its microstructure facilitates continuous gas transportation by rapidly trapping bubbles and allowing gas medium to propagate along the flow direction in the interconnected gas layer (Fig. 4e, Supplementary Fig. 24 and Supplementary Movie 4). This gas transport process exhibits high sensitivity, resulting in a transient phase (less than 1 s) that has negligible impact on the statistics of gas collection. Conversely, the FH surface exhibits scattered, unattached bubbles and a limited number of dispersed capture sites, resulting in most bubbles bypassing the surface without being captured (Fig. 4f, Supplementary Fig. 24 and 25). Furthermore, the gas collection volume on MA surfaces shows a nearly perfect linear relationship with time at all three injection rates (Fig. 4h and Supplementary Movie 5), and the collection rates (slopes of the curves) align with the injection rates, demonstrating continuous, loss-free gas transport over long distance (>10 cm), consistent with our experimental observations (Supplementary Fig. 26). Notably, after 12 h, the capture rate of the MA surface still maintains 98.8% (Re = 2083), implying its robust efficiency and potential for long service in practical applications (Supplementary Fig. 27 and Supplementary Movie 6).

## Discussion

We have explored the upper limits of free-rising bubble capture rate achievable with the super-aerophilic nano/microstructured surfaces. This study demonstrates that a typical conical array of hundreds of micrometers, formed a plastron through nanoparticle coating, enables stable and fast bubble adsorption within 1.5 ms upon first contact. We provide compelling evidence that the mechanism lies in the substantial deformation of the gas layer within each characteristic unit, while a stable plastron is ensured. This local deformation permeates throughout the entire array surface and boosts energy conversion from bubble kinetic to gas layer surface energy, significantly contributing to accelerating the thinning of the liquid film between the gas

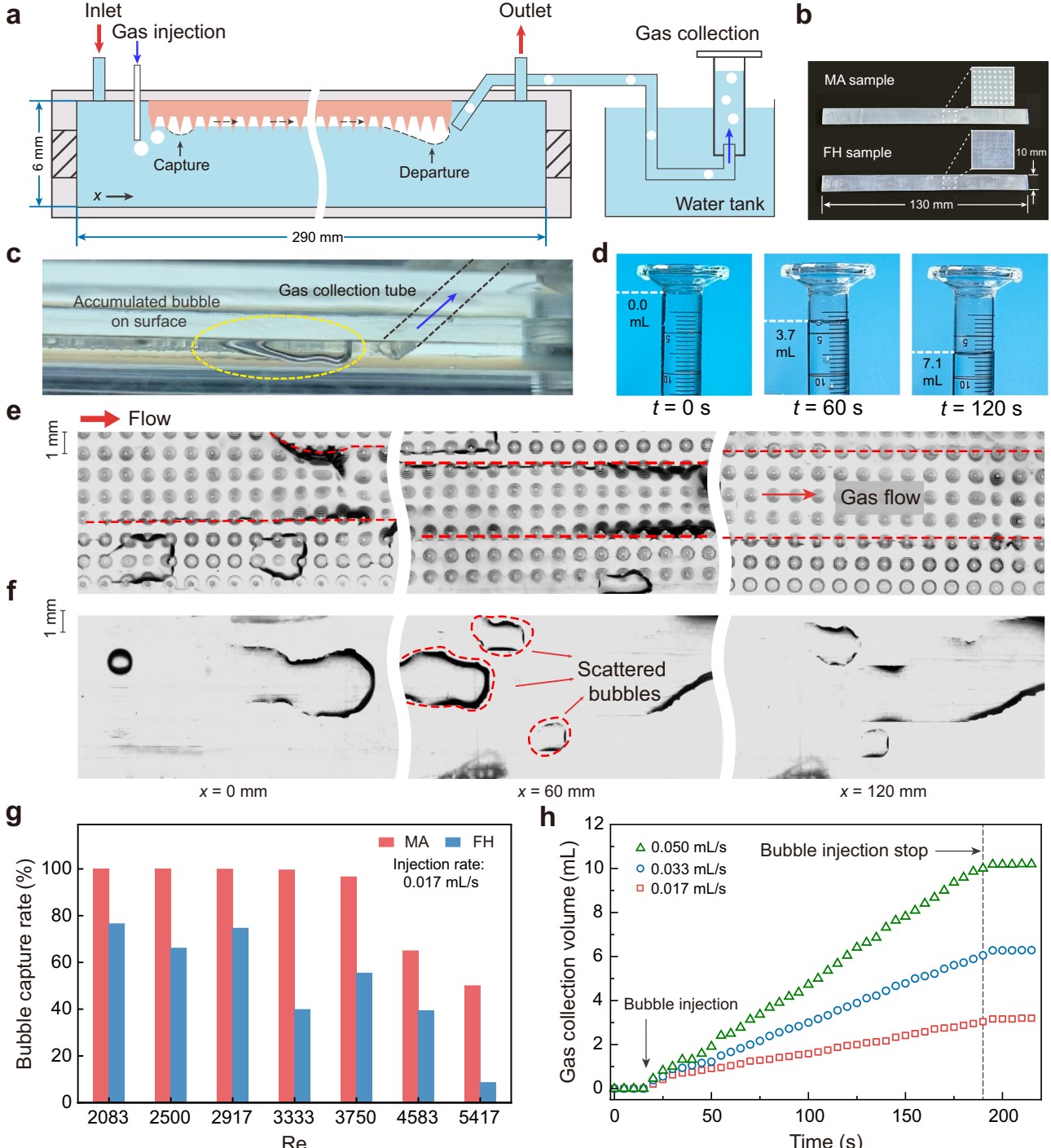

**Fig. 4 | Fast bubble capture and collection in flow environment. a** Experimental setup and working process for bubble capture involve constructing an inner flow channel measuring 290 × 10 × 6 mm, connected to a gas collection section, to facilitate bubble capture measurements in a flow environment. **b** The optimized MA surface with a conical array structure of $b/a = 1.7$ and the FH surface, both measuring 130 × 10 × 1.6 mm. **c** An optical image of the accumulated bubble at the end of the MA surface, with a gas collection tube connected to the immersed volumetric cylinder. **d** Gas collection setup for an immersed volumetric cylinder. The optical image sequence of gas collection in the immersed volumetric cylinder during a 120 s test. **e** Optical images display stable continuous gas flow formed on the MA surface recorded after 5 s bubble capture process. **f** Optical images show separated bubbles captured by the FH surface at varying locations along the surface recorded after 5 s bubble capture process. **g** Bubble capture rate (captured volume/gas injection volume) on the MA and FH surfaces as a function of $Re = \rho v d_e / \mu$, where $v$ is the flow rate and $d_e$ is the equivalent diameter of chamber, ranging from 2083 to 5417. The bubble injection rate is 0.017 mL/s. **h** The collection gas volume under varying bubble injection rate accumulates with time at $Re = 2083$.

layer and the bubbles. The role of microstructures in accelerating drainage remains robust across a range of bubble approach velocities, on the order of $10^{-2}$–$10^{-1}$ m/s in our study and most of the literature (Fig. 1h), though the $10^{-2}$ m/s cases need further validation, as bubbles

released over very short distance ($10^{-1}$ mm) may not reach a steady state and could be highly sensitive to experimental conditions. The critical pressure criterion, which is necessary for the mechanism to function effectively and essential for the gas layer's stability and

controllability, is defined. From this definition, we further establish a capture capacity spectrum to optimize microstructures. Interestingly, the bouncing and capture of bubbles share some similarities in liquid/gas bridge formation with the bouncing and adhesion of droplets[47]. However, the dynamics of the three-phase contact line and adsorption during bubble capture, influenced by the surrounding liquid, are more complex than droplet adhesion, warranting special attention.

We underscore that rapid capture of free-rising bubbles does not rely solely on complex and finely detailed nanostructures. Beyond the well-acknowledged microstructural roles, such as slip boundaries that accelerate liquid film drainage[21], rough surfaces that create uneven liquid films[24], and sufficient gas storage space that promotes bubble merging[48], we reveal the dominant role of the stationarily pinned three-phase contact lines, in rapidly reaching the threshold thickness for liquid film rupture, a fundamental aspect of liquid film dynamics within microstructures that has been overlooked. Through a comparative analysis of bubble capture on the nanoparticle-free silanized super-aerophilic surface, we also highlight the role of nanostructure in stabilizing the gas layer. These findings, together with our proposed mechanism of microstructures that facilitate accelerated film drainage, collectively offer a fundamental understanding of the ultrafast capture process of free-rising bubbles.

These insights on the role of microstructures may offer alternative perspectives in aerophilic materials, enabling broad applications ranging from microfluidics to subsea methane harvesting and drag reduction. We also demonstrate the efficient entrapment of bubbles in challenging shear flow environments, ensuring prolonged, loss-free gas transport with our model surface. As moving forward, tackling shear flow perturbations and gas dissolution will be pivotal. Stabilizing the gas layer and its three-phase contact line at high Reynolds numbers will broaden the technology's applications, from delicate lab-on-a-chip devices to robust industrial processes.

## Methods
### Natural materials
The *Salvinia* leaves were all purchased from Crazy Aquatic Plant Trading Co. Ltd. Three *Salvinia* species show different apex features on trichomes including Cucullata, Oblongifolia, and Molesta types[49], but all species show a large height-to-diameter ratio ranging from 4.0 to 22.7. Each leaf was cut into 10 mm × 10 mm specimens for the bubble capture tests.

### Surface fabrications
We fabricated the microconical structured surfaces by a projection micro stereolithography 3D printing technique (nanoArch P140, BMF Precision Tech Inc.), and general HTL resin (BMF Precision Tech Inc.) was exposed to a 405 nm laser to be photocured with a layer resolution of 10 μm. A commercial spray, Glaco (Soft99), was employed to decorate the samples with silanized silica nanoparticles. The surfaces were further dried at room temperature for 120 min to complete the coating.

We converted the microconical array structure into triangular ridges to observe the evolution of liquid-gas interface profiles. It was composed of parallel triangular ridge protrusions with a conical spacing $L$ of 2500 μm, a base radius $a$ of 346 μm, and a height $b$ of 1200 μm, using the same preparation process as above.

### Surface morphology characterizations
The laser confocal microscope images of the microstructures on both the *Salvinia* leaves and our MA surfaces were measured by an upright laser confocal microscope (VK-X3000, KEYENCE). Scanning electron microscopy (SEM) images of the nanoscale structures were obtained on JSM-7800F at 5 kV (JEOL Ltd.) and SEM images of the microcones were captured by GeminiSEM 560 (ZEISS).

### Bubble capture test
The single bubbles rising and bursting on the aerophilic surfaces were analyzed optically using a high-speed camera with 9000 fps (X190, Revealer). The testing surfaces were invertedly immersed in deionized water secured by an aluminum bracket (Supplementary Fig. 14). Bubbles were released through quartz tubes at the bottom of the chamber. The bubble sizes were tuned by the varying inner diameters of quartz tubes (0.1–2 mm) attached to a syringe. A syringe pump was used to control the bubble-releasing velocity. In order to release bubbles at varying horizontal distances to the conical tip, the test surface was additionally connected to a displacement platform with a positional precision of 20 μm, with bubbles formed by the fused silica capillary tube (Polymicro Technologies), featuring an inner diameter of 100 μm and an outer diameter of 165 μm. The surface was unidirectionally moved 200 μm per test, and microbubbles were released beneath the surface at a vertical distance of 3.2 mm to precisely contact the intended location on the test surface.

### Film drainage PIV test
Polyamide particles with a diameter of 20 μm and a mass density of 1.03 g/mm$^3$ (Arkema Inc.) were added to deionized water in a low concentration ($< 10^{-4}$ w/w). Two semiconductor lasers (1600 mW, 450 nm) were placed on both sides to create a laser sheet, which focused on the meridian plane of the liquid film between the bubble and the test surface. The film drainage process was recorded by a high-speed camera at 4000 fps (X190, Revealer). PIVlab software[50] was used to measure the velocity field and subsequent PIV analyses. The fast Fourier transfer (FFT) window deformation technique with two passes and a Gaussian sub-pixel estimator was employed to ensure an adequate particle number in the initial pass ($32 \times 32$ pixels).

### Characterizations of the liquid-gas interfaces
To gain insight into the 3-dimensional liquid-gas interface configuration, we utilized confocal microscopy to observe the aerophilic surface underwater. The aerophilic surface was immersed underwater, where water was dyed by Rhodamine B (95%, Energy Chemical Corp.) with a concentration of 0.1 mg/L. An upright confocal microscope (LSM 900, ZEISS, 4X, dry objective) was used to scan the liquid-gas interface on the test surface. The fluorescence signal in water appeared as a green color, while no signal was detected in the gas layer.

To examine the deformation of the interface when the bubble approached, the test surface was submerged invertedly in water labeled with fluorescein isothiocyanate (isomer I, 90%, Aladdin Corp.) at a concentration of $10^{-5}$ w/w. The light reflection method was used to obtain the reflected light signals of the gas-liquid interfaces via a laser with a wavelength of 450 nm (1600 mW).

### SPH simulations of the film drainage
We carried out numerical simulations based on an SPH solver enhanced by MUSCL scheme[51] to investigate the film drainage velocity distribution within the microconical units. The Tammann equation of state was used to describe the compressibility of the gas and liquid phases. A particle regeneration technique[52] was employed to maintain the liquid-gas interface distinct throughout the simulation. The continuum surface force model[53] was introduced to consider the macroscopic surface tension effect, with a correction force parallel to the wall surface associated with the advancing angle (162° ± 3°) applied near the TPL to realize the dynamic control of the contact angle. The detailed descriptions are provided in Supplementary Note 3. The bubble radius $D_0 = 1.6$ mm is the same as the experimental bubble radius, and the surface tension coefficient of pure water $\sigma = 0.0728$ N/m.

## Bubble capture and transportation in water flow

We established the bubble capture system in a flow environment and examined its gas collection performance. The system mainly consists of a flow channel, a submerged pump, a syringe pump, a bubble collection section, and a high-speed camera system (Supplementary Fig. 22). The flow channel consists of a base and a top cover with an inner flow channel measuring $290 \times 10 \times 6$ mm. The top cover is equipped with a 2 mm deep groove for the inverted tight adhesion of test surfaces. A gas injection port was located at $x = 80$ mm of the top cover, and the inner and outer diameters of the injection tubes were 1 mm and 1.7 mm, respectively. Gas bubbles were released at a rate of 1 mL/min–3 mL/min, and their behavior was recorded by a high-speed camera at 200 fps (X190, Revealer). Meanwhile, a submersible pump (25 W) was used to drive the water flow, and the flow rate could be adjusted from 0 to 1.6 L/min to achieve a Re number range of 0 to 5417. To test the efficiency of bubble collection, the tube close to the end of the test surface was connected to an inverted cylinder filled with water, and the volume of collected gas was obtained by recording the scale of the cylinder.

To complete a 12-hour bubble trapping experiment, our experimental procedure involves using a 30 mL syringe with a syringe pump to maintain a consistent output rate of 1 mL/min. Every 30 min, the syringe is briefly disconnected ($\leq 10$ s) for air replenishment, ensuring minimal disturbance to the surface air layer and continuity in the 12 h bubble-trapping experiments.

## Data availability

All raw data related to the main text and supporting information are provided in the Source Data File (https://github.com/Luwen-Zhang/Bubble-capturing.git). Source data are provided with this paper.

## Code availability

The codes used in this study for solving liquid-gas interface deformation within microstructures and directly simulating liquid film drainage[54] are publicly available at https://doi.org/10.5281/zenodo.15083029, which links to the GitHub repository: https://github.com/Luwen-Zhang/Bubble-capturing.git.

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

## Acknowledgements
We acknowledge financial support from the Marine Equipment Foresight Innovation Union Project (2-A3, L.W.Z.), the National Natural Science Foundation of China (12272228, L.W.Z.), the Research Grants Council of Hong Kong (17213823, and 17205421, L.Q.W.), and the State Key Program of National Natural Science Foundation of China (92252205, B.L.W.).

## Author contributions
L.W.Z. conceived and directed the project. Y.H. and L.W.Z. designed the experiments. Y.H. set up the experimental system, and performed confocal fluorescence, scanning electron microscope, and particle image velocimetry measurements. Z.X. and Y.H. developed the theoretical model. H.S. conducted the Smoothed-particle hydrodynamics simulation. Y.H., L.W.Z. and H.S. analyzed the data. L.W. and B.W. contributed to the interpretation of the results. Y.H., L.W.Z. and L.W. wrote the manuscript with input from all the other authors.

## Competing interests
The authors declare no competing interests.
