## [Transparent Peer Review file · Nature Communications]

Understanding ultrafast free-rising bubble capturing on nano/micro-structured super-aerophilic surfaces

Corresponding Author: Professor Lu-Wen Zhang

Version 0:

Reviewer comments:

Reviewer #1

(Remarks to the Author)

The manuscript presents an intriguing study on the dynamics of ultrafast bubble capture using microtextured aerophilic surfaces. The authors propose using conical microstructures to achieve rapid bubble capture, which has potential applications in various aquatic environments. However, there are several areas where the manuscript could benefit from further clarification and refinement, supported by recent literature. In addition, I cannot accept that the surfaces are defined as "simply microtextured" since there is clear usage of nanoparticles (Glaco spray coating). This must be corrected for scientific accuracy. I also found that the focus of the work misplaced. It is significantly more novel in the context of rupturing bubbles at high approach velocity, which relies on the nature of the microcones decorated with nanoparticles. This is where the emphasis should be. Detailed comments as follows: Please see attached review.

(Remarks on code availability)

Simulations are not my area of expertise, but i have made general comments on the use of various governing equations which should be addressed.

Reviewer #2

(Remarks to the Author)

This manuscript, wherein the bubble capture of aerophilic conical microstructures is analyzed, does demonstrate cutting-edge air capture rates. This work has clear significance, for example their microconical surfaces can clearly capture and transport more gas underwater than an equivalent aerophilic flat surface, which the authors show through experiments.

Moreover, I was on the whole impressed with the quantification of the gas layer and interface profiles, particularly as it seemed to show fairly good agreement with actual experimental results. Additionally, I found the experiments quantifying bubble capture dynamics to be well done.

However, I do have some qualms with the manuscript. An argument you pose in the introduction is that microstructural features 'overshadow' the role of nanostructure for bubble capture. You don't prove that. You prove that a nanostructured surface with a corresponding microstructure performs better than a nanostructured surface without that corresponding microstructure. That is to say, you prove that microstructure is relevant to bubble capture. For you to prove that microstructure overshadows nanostructure, you'd have to find some way of isolating the effect from microstructure and the effect from nanostructure in the improved bubble capture of the microstructured surface, which you haven't done. The conclusion addresses this point appropriately, in that you state 'that rapid bubble capture does not necessitate on overly complex [...] nanostructures'. This you do prove.

On a more surface level, significant chunks of the manuscript, particularly the introduction, include numerous grammar/language mistakes. Examples of poor phrasing include page 1 (starting from introduction) lines 15-16 'random porous' should be 'random pores' or 'random porous structures'. Likewise the page 1 lines 9-10 'the respiratory of diving insects' should be 'the respiration of diving insects'. These are not particularly egregious on their own, but they are just a couple examples of a recurring problem. Similarly, you refer to your methods section consistently as 'materials and methods'. On the whole, the writing of this manuscript should be reviewed and thoroughly edited.

(Remarks on code availability)

Code provides instructions through a README file.

Reviewer #3

(Remarks to the Author)

Comments on

Ultrafast bubble capturing on simply microtextured aerophilic surfaces

The targeted collection of bubbles from an aqueous environment is of current interest and important for various industrial and environmental applications. Whereas the bubble capture by the plastron sustained on superhydrophobic surfaces has been in the literature for some time, the present work is a detailed study building on some of the prior designs and presenting an optimized approach for fast bubble capture. The approach relies on the combination of nanoparticle superhydrophobic coating and microscopic conical features, inspired by the nature of *Salvinia* leaves. The study is of high quality and covers various aspects of the surface's design and performance. It has the outreach and potential for a Nature Communications paper. I have a few, rather general comments the authors might want to consider before the final recommendation:

1. I hesitate to use a record stating "capture time of 0.83 ms". The capture times will vary from experiment to experiment. In different works it is defined in various ways, starting from different bubble positions and sometimes including and sometimes not the time for the bubble spreading on the surface, so one should be careful when comparing. Use more general statements such as "capture times down to about 1 ms" and explain how it was determined (time for the film rupture?).

2. What is the practical advantage of lowering the capture time from a few ms to under 1 ms – anyway there are slower stages of the processes than the time required for the bubble to reach the interfaces. I guess that the major time-shortening effect is the switch from bouncing before capture to capture without bouncing.

3. The ultrafast capture suggested is tested and works only for pure water. However, in practical situations as in flotation, there are often surfactant additives, natural or added. Some comments on how the surfaces will perform if such additives are present. The same holds for high electrolyte concentration water solution as seawater which is known to significantly inhibit the bubble coalescence times.

4. Another major issue with using a microtexture supported gas layer is the tendency of the gas layer to dissolve with time. How robust are the current surfaces to high pressure and extended underwater time?

5. The force balance model used to estimate the film drainage captures the major physical features of the system. However full numerical simulation might be more accurate. The interfaces' mobility is of great importance. See further discussions in:

I.U. Vakarelski, F. Yang, S.T. Thoroddsen. Free-rising bubbles bounce more strongly from mobile than from immobile water-air interfaces, *Langmuir* 36 (2020) 5908-5918.

I.U. Vakarelski, K.R. Langley, F. Yang, S.T. Thoroddsen. Interferometry and simulation of the thin liquid film between a free-rising bubble and a glass substrate, *Langmuir* 38 (2022) 2363–2371.

6. Literature is generally well referred to. One more recent related work:

Wong, W. S. Y. et al. Designing Plastrons for Underwater Bubble Capture: From Model Microstructures to Stochastic Nanostructures *Adv. Sci.* 2024, 2403366

7. Last – should it be "simply microtextured aerophilic surfaces" or "simple microtextured aerophilic surfaces"?

(Remarks on code availability)

Reviewer #4

(Remarks to the Author)

Hu et al. present an insightful study on bubble capturing, with a specific focus on the effects of microstructuration. Their research primarily examines gas bubbles ranging from $D_0 = 1.6\text{--}3.5$ mm and demonstrates that microstructuration with a 1 mm pitch can significantly reduce capture time by enabling bounce-free capture. The authors identify three distinct regimes—bounce, capture, and attachment—and develop a pressure threshold model based on bubble size and the substrate's advancing contact angle. This study is thorough, well-constructed, and presents valuable insights into the microstructuration regime, making it a commendable contribution to the field.

I believe the manuscript is suitable for publication in Nature Communications, as it 1) includes fundamental considerations about gas bubble capture, which not only lowers the limit of capture time but also offers a new perspective on efficient strategies for bubble capture, and 2) is urgent as it may redirect the current focus of the scientific community - from nano to the importance of micro - thereby potentially optimizing resource utilization. The findings are relevant to any field where gas capture is utilized and potentially for the wetting community as well.

Overall, I recommend acceptance after a major revision.

Major comments

1. Reading your manuscript, I believed for a long time that your surface was purely microstructured—there is no mention of nanofeatures in the introduction. Only in the section “Ultrafast Bubble Capture” do you mention the coating with hydrophobic nanoparticles. Please make it clear very early that you are utilizing a hierarchical surface, comprising both nano and microstructuration. This is particularly important as you later highlight the critical role of the nanoparticles in fast capturing. Please go over the manuscript to ensure that this is clear from a very early point and no matter where the reader starts from in your manuscript. It will make the manuscript clearer.
2. In large parts of your manuscript, you are exploring $D_0/L = 1.6$ to 3.5 . Please argue why this is the relevant range because it may otherwise appear to simply be a range where your choice of structure length scale seems to work. Likewise, please argue even more explicitly why your capture mechanism wouldn't work if everything were scaled down by a factor of 10-100. What do I mean by this? Your main claim in the paper is that microstructuration is necessary, but I am not convinced that this need is not just caused by the use of 1.6–3.5 mm gas bubbles, and you indeed find that it doesn't work for bubbles that are too small or too large. But would it work if you scaled the structures down proportionally to the bubble sizes? A related note: Do you have any thoughts on how to broaden the range of bubble diameters that experience ultrafast capture?
3. Page 7: “...shows that the gas bridge tends to form at the specific positions where the bubble meets TPL.” If pinning is critical, please add data or references to NP-free control samples, showing how triple-line pinning is indeed critical. You present a convincing theory, but this control experiment is missing. Removing the nanoparticles would also alter your advancing contact angle, thereby providing a second data point for your critical pressure theory (which has advancing contact angle as a parameter). This is critical for supporting one of your main claims. One way to address this is to explore $L \ll 1$ mm to bridge the micro to nano regime. The question is whether the absolute size of the structure is important or if it is just the ratio D_0/L that matters.
4. For the bubble collection in liquid flow, it is unclear what role the transient phase plays. I imagine it takes some time to set up the gas-flow channel (is that the 5 seconds you mention in the extra material)? Please include details on the transient versus steady-state behavior.
5. Is there a possibility to add a prediction for the bubble trapping as well?

Minor comments

1. Page 5: Please add errors to 1.5 ms, 71.5 ms, and 1.54 ms (Is this the same as 1.5 ms? If yes, why don't you use the same rounding?).
2. Page 5 (and caption of Fig. 1F): For Sc , is this a characteristic size or a roughness? It appears that you refer to it as both, which can be confusing for people in other fields, as roughness is also defined as the actual surface area normalized to the projected surface area. Please clarify and also elaborate on the benefits of normalizing with respect to S_0 .
3. For bubble capture, there is the question of how fast the bubble merges with the plastron, but there is also the subsequent air transport to prepare the surface for the next bubble, which is important for continuous bubble capture. You define capture time as “the interval from initial contact to forming a spreadable three-phase contact line,” but given the transport argument above, another logical choice would be the interval from initial contact until the system is ready to accept the next bubble. Please provide an estimate for this second definition and elaborate on how the transport/build-up may be a bottleneck for continuous bubble capture.
4. Your introduction goes over the development in the field of bubble capture, and you also repeatedly emphasize that you are pushing the limit for capture time. However, as you mention in the introduction, previous work in the field includes micro-nano structuration, so it is unclear why previous microstructures were not as effective as those in the presented work.
5. Page 7: Is the precision of 129.9% justified? Please provide an error estimate.
6. Page 7: “We highlight a sharp reduction in capture time t_c from 100 ms to 1.5 ms near TPLs.” Here your surface shows a variation in capture times similar to other works. Is this also the cause of variation in other works? This is relevant for understanding why no one before you achieved similarly small capture time variation.
7. Page 8: 162° —Please add the errors to your numbers. I've seen in your supporting information that you have them all, so please add them to the main article to make it easier for the reader to judge the reliability/trustworthiness of the study.
8. Page 8: As I understand, your theory (eq. 1) is a 2D model. How does the actual 3D case differ from your model? Please also make it clearer that you are using “parallel triangular ridge protrusions” (i.e., a different geometry than your initial cones) for this part of the study. It was only obvious to me after reading the methods section on page 19.
9. Page 11: “Still maintains 98.8%”—what Re is this for?
10. Conclusion: “The most basic microstructure” — Define how to measure “most basic,” or perhaps just call it a “simple conical structure” throughout the manuscript.
11. Can you perform continuous flow-free capture? What is the impact of MA substrate tilting on bubble capture and transport?
12. Fig 1f: Please make it clearer that the small numbers are paper references and that your structure also belongs to the micro-nano group.
13. Fig 3g: Please include $Re > 3750$ and $Re < 2083$, which I believe you have data on but only put in the extra figures. It won't take up more space, and it will give the reader a better idea of the system's limitations. Please also add the injection rate so it is easier to compare with Fig 4h.
14. Fig 4f: “...at varying locations along the surface after a 5-second bubble capture process.” What does this mean? Is it the

time it takes to establish a steady state? And starting from a completely wet surface or starting from a plastron? Or?

15. Fig 4c-d: I saw the two figures are not aligned with each other, and I wonder if this is the case for more figures. Improve it if you feel perfectionistic.

16. Page 20: Please explain how you provide gas bubbles for 12 hours straight. It says: "a submerged syringe pump." I'm not sure what this is, but it suddenly struck me that it was not clear to me how you provide air for 12 hours straight. If you use a syringe, it has to hold $3 \text{ mL/min} \times (60 \times 12) \text{ min} \sim 2 \text{ L}$, which is a syringe bigger than I have ever seen. I expect that I have misunderstood, but more readers will misunderstand, so I would suggest that you make this clearer. It should be easy 😊.

17. Page 20: "The flow rate could be adjusted from 0 to 1.6 L/min to achieve a Re number range of 0 to 3333." However, you investigated higher Re numbers, so how did you do this?

18. Ex Data Fig 2b: What is the unit on the x-axis? Does this range actually match the A-A in (a)?

19. Ex Data Fig 2c: "10 μL hangs on the FH surface." In the photo, it looks very much like a sessile droplet and not a pendant/hanging droplet. Please clarify.

20. Ex Data Fig 3: "t0 ms" in the figure. Is the "ms" unit supposed to be there in the 2nd column?

21. Ex Data Fig 8: What is the reference point for this "contact location," i.e., where is "contact location" = 0?

22. Ex Data Fig 22: "all present nearly straight lines at all three injection rates." In (a) 0.017 mL/s, the curve does not look straight. Please comment and correct if you made an error.

23. The work reminds me of the work by Prof. Quéré on bouncing droplets, where they also identify various regimes for bouncing and sticking, which depend on bridge formation. It would be interesting to have a short reflection on how looking at techniques for reducing bouncing of water droplets can inspire further developments in the field of gas bubble capture.

a. Mouterde, T., Lecointre, P., Lehoucq, G. et al. Two recipes for repelling hot water. *Nat Commun* 10, 1410 (2019).

<https://doi.org/10.1038/s41467-019-09456-8>

24. The title could be made more informative, such as: "Ultrafast bubble capture on microcone-textured aerophilic surfaces." Even better if there is mention of the nanostructuration, which may be hinted at by calling it "super-aerophilic" instead; similar to the field of water wetting, where superhydrophobic implies that some structuration has been applied to the materials.

25. Consider having an extra round of proofreading done, focusing on writing easily digestible sentences.

Sincerely,

Dr. Nikolaj K. Mandsberg

(Remarks on code availability)

Version 1:

Reviewer comments:

Reviewer #1

(Remarks to the Author)

Over the course of the review, I believe it is now clear from opinions of the reviewers and the authors that there remains two main points of contention.

First, the manuscript does employ the use of nanostructures, albeit claims in stabilization of the Cassie-state, they should still be explicitly described to avoid misleading readers. This is as of yet (see last section of the review attached) not entirely clear in the primary sections of the manuscript.

Second, the surfaces proposed appear to work primarily with fast moving free bubbles, and not so well with slow moving bubbles (in fact they take an order of magnitude more time than state-of-the-art nanotextures). However, it is also difficult to justify if they are indeed the best with slow bubbles without a direct comparison to the state of the art. Therefore, claims should be made within such limits until more evidence is clearly presented. As far as glaco is concerned, it also has an unclear chemistry makeup, and that makes it a difficult material to benchmark.

With a reorganization of the manuscript around its key points of discovery, and even with glaco surfaces as a control, I would be happy to recommend this work for publication. However, the claims must be accurate and not as universal-sounding as they currently are, which I would still suggest a major revision on. Reduction of the universal-sounding claims is helpful to prevent premature limitation of the field and avoid confusing the benchmark set in current literature.

(Remarks on code availability)

Reviewer #2

(Remarks to the Author)

The authors have produced a noteworthy manuscript on the impacts of microtexture on bubble capture underwater. This work has clear significance and displays an impressive quantification of the gas layer and interface profiles.

While I had some concerns with their writing regarding the relative contributions of micro- and nanotexture, they have since both moderated the tone and conducted additional rounds of experimentation. Given that, I find no further major issues with this publication.

(Remarks on code availability)

Reviewer #3

(Remarks to the Author)

I am satisfied with the responses given by the authors to the comments in my first report. The paper can be published as it is.

(Remarks on code availability)

Reviewer #4

(Remarks to the Author)

I have reviewed the point-by-point responses and the revised version of the manuscript by Hu et al. and carefully considered the points raised by all four reviewers, with particular attention to the comments made by reviewer 4. The authors have satisfactorily addressed the concerns raised by reviewer 4 and appear to have responded comprehensively to the feedback from the other reviewers as well. Collectively, these revisions have significantly improved the manuscript, and I now believe it meets the standards for publication in Nature Communications.

However, I would like to highlight two minor issues:

- 1) There appears to be a grammatical issue in the revised abstract: 'surpassing the available most minimum capture times.' I suggest revising this for improved clarity.
- 2) The figure titled 'Fig. R10. Continuous bubble capture on the tilted MA surface' is missing from the revised supplementary information. Please ensure that it is included for completeness and clarity.

Finally, I would like to acknowledge the authors' considerable effort in addressing the reviewers' comments and strengthening the manuscript. I also encourage the authors, in future work, to revisit their statement: "We feel it's unrealistic to design a structure that can capture all bubble sizes." Exploring this challenge could lead to exciting new opportunities in this field.

Sincerely,
Dr. Nikolaj K. Mandsberg

(Remarks on code availability)

Version 2:

Reviewer comments:

Reviewer #1

(Remarks to the Author)

Review of "Ultrafast bubble capturing on microtextured superaerophilic surfaces" by Hu et. al.

I have looked through the descriptions of this work again, and this round, really do appreciate the efforts spent by the authors in addressing and critically assessing revisions. I believe that the clarifications and the re-bracketing of the research context now fully justifies this work for publication in its current form. The discussions in the main manuscript, comparisons to the literature in this area, alongside concluding statements are well-structured.

Based on the review document and line-by-line responses (I did not spend additional time looking through revisions in the manuscript except the abstract and conclusions), this work is ready for publication.

I look forward to reading it in formal print.

Thanks!

(Remarks on code availability)

I did not run the code, but i looked through the key logic behind how the code is structured, alongside the descriptions provided by the authors, and i believe that it should be conceptually correct. Details of course, will be harder to assess without a full run.

Point-to-point responses to reviewers' comments

First and foremost, we would like to thank the reviewers for your thoughtful evaluation of our manuscript. Your insightful comments and constructive suggestions are very helpful in improving the quality of our work. We have carefully revised the manuscript and supplementary information in response to all the your comments, addressing concerns where appropriate.

Reviewer: 1

The manuscript presents an intriguing study on the dynamics of ultrafast bubble capture using microtextured aerophilic surfaces. The authors propose using conical microstructures to achieve rapid bubble capture, which has potential applications in various aquatic environments. However, there are several areas where the manuscript could benefit from further clarification and refinement, supported by recent literature. In addition, I cannot accept that the surfaces are defined as “simply microtextured” since there is clear usage of nanoparticles (Glaco spray coating). This must be corrected for scientific accuracy. I also found that the focus of the work misplaced. It is significantly more novel in the context of rupturing bubbles at high approach velocity, which relies on the nature of the microcones decorated with nanoparticles. This is where the emphasis should be.

Response: We are grateful that the reviewer found our work intriguing, and appreciate the insightful suggestions for further clarification and refinement. We have thoroughly addressed the points raised, in terms of clarifying the mechanisms and refining terminology, by adding a large number of experiments. After careful consideration, the title is revised to “*Ultrafast bubble capturing on microcone-textured super-aerophilic surfaces*” by removing “simple.” This change aims to specify microcones used as textures, and indicate that “*super-aerophilic*” is intended as a juxtaposed condition, rather than a subordinate one, to “microcone-textured” for achieving ultrafast bubble capturing. We have also ensured that this revision implicitly acknowledges the contribution of nanoparticles, while highlighting the central role of the microstructures. The term “*super-aerophilic surface*” encompasses, but is not limited to, the way in which *super-aerophilicity* is achieved through nanoparticles, as supported by the nanoparticle-free silanized control experiments (see detailed response to Comment 13). Based on our findings and supplementary results, the focus of our work remains the investigation of microstructures' drainage enhancement role in bubble capture, which we will elaborate on in subsequent responses and discussions.

1. Abstract: “Current aerophilic surface designs focus predominantly on complex nanostructures, and largely overlooks the fundamental role of microtextures.”

On this exact topic, a recently published article (Adv. Sci. 2024, 2403366) might be of interest, where model microtextures were specifically investigated for their efficacy in bubble rupture-and-absorption. The article also describes the differences observed between microtextures and nanotextures in the context of the entire bubble capture process. The conclusions made there can perhaps help to improve how the authors' abstract is written?

Response:

We thank the reviewer very much for the constructive comments. We have carefully studied the work in Adv. Sci. 2024, 2403366 and included it in the Ref. [27], which indeed refined our narrative on the contributions of nano/microstructures to bubble capture in the revised manuscript.

Our study addresses a different scope compared to Ref. [27]. We focus on freely rising bubbles, addressing the fundamental issue of whether they can be captured or not upon the super-aerophilic surface, particularly with emphasis on the role of microstructures. Ref. [27] discusses needle-fixed bubbles, which inevitably rupture and absorb upon active contact with a super-aerophilic surface, with a comprehensive analysis of impact factors such as microtexture size and gas fraction.

It is clear that in these two studies, the initial state of the bubbles differs, leading to a divergent understanding of the 'entire process' of bubble capture. In our study of free bubble capture, the plastron and bubbles are influenced by microstructures long before contacting the surface. This significant 'pre-capture' stage, which greatly impacts whether a capture event can occur upon first contact, is one of the key aspects of our analysis. This aspect has not been well understood before.

Accordingly, we have refined our abstract, with the intriguing study of Ref. [27] offering some helpful perspectives:

“Current super-aerophilic surface designs often emphasize complex nanostructures, while the fundamental role of microtextures in first-contact bubble capture has not been well understood.”

2. Abstract: “Achieving an ultrafast bubble capture in 0.83 ms with a simple array of hundred-micron conical structures, surpassing the available minimum capture times.”

3. Introduction: “We report the fastest and most stable bubble capture among all the available microstructured surfaces with exceptionally simple conical arrays.”

The manuscript claims in the above two areas that the extremely fast rupture behavior was captured using microstructures only. This claim is easily overstated, especially considering the role of nanoparticulate structures due to the Glaco spray.

Recent studies have demonstrated that nanostructure-coated micropillars behave similarly to

bare nanostructures, which suggests that the controlling factor in the rapid capture may be the nanostructures rather than the microstructures alone (Adv. Sci. 2024, 2403366). Without demonstrating results of micro-cones without nanoparticles coated on top, these claims should be moderated.

Response: We appreciate the critical comments raised by the reviewer. To enhance the rigor, we have moderated the tone and more accurately defined our surface to incorporate the role of nanoparticles in the above-mentioned two parts, as well as relevant sections in the manuscript.

After reviewing the Ref. [27] (Adv. Sci., 2024, 2403366), we would like to point out that the claimed similarity between perfluoroalkylated silane functionalized micropillars (not nanostructure-coated) and hierarchical nanostructures occurs only during the final stage--- contact line spreading, rather than entire bubble capture process. Please refer to its Section 2, titled “Micro to Nano: Understanding Contact Line Advancement and Pinning in Bubble Absorption”, where the whole discussion in the third paragraph of page 9, “Hierarchical Nanostructured Surfaces: ... Notwithstanding minor differences in contact line velocity, such behavior is analogous to our observations with microstructured surfaces” is entirely within the context of contact line spreading. Therefore, we cannot suggest that “the controlling factor in the rapid capture may be the nanostructures rather than the microstructures alone” is the case.

In our original manuscript, we focus on the stage of film drainage when capturing free-rising bubbles (rather than capturing needle-fixed bubbles in Ref. [27]), where the presence or absence of microstructures makes a significant difference, even when both surfaces are decorated with the same nanoparticles. This indicates that microstructures are the control factor. We also present new results of microstructures without decorated nanoparticles, which will be discussed in detail in response to Comment 13.

We have revised the related sentence in the Abstract as:

“We report the rising bubble-induced large deformation of the entrapped gas layer, rapidly thinning the liquid film to its rupture threshold and thus achieving an ultrafast bubble capture in down to about 1ms with a simple array of microcones, decorated with nanoparticles as a convenient example to obtain super-aerophilicity, surpassing the most capture times.”

We have also removed the first short paragraph of the introduction, as it serves more as an unnecessary lead-in. The information is already conveyed in the abstract.

4. Introduction: The former refers to the interval from initial contact to forming a spreadable three-phase contact line, and can be varied by two orders of magnitude across surfaces.

I believe, that, the introduction should be more comprehensive towards what has been explored. I believe that research involving the addition of surfactants can delay rupture behavior, with different surfaces experiencing up to 3-4 orders of magnitude with the same liquid or up to 5-

6 orders of magnitude with different liquids (See Adv. Mater. 2023, 2300306, Figure 5f). This has been explained nicely in terms of the nature of compatibility between surface chemistry and liquid composition.

Response: We acknowledge the valuable point raised. Previous work indeed found that surfactants can cause a difference of 5-6 orders of magnitude in rupture time (Adv. Mater. 2023, 2300306) and has been included in Ref. [20]. However, even in pure water, significant time disparities (>1-2 orders of magnitude) are reported in the paper referred to by the reviewer (Adv. Mater. 2023, 2300306, Figure 5f) and many other works (Phys. Rev. Lett., 2016, 094501; Nat. Commun., 2021, 5358; Adv. Mater., 2021, 2101855; Adv. Mater. Interfaces, 2020, 1901599), emphasizing the incompleteness of the current understanding of bubble capture. Our current statistics also demonstrate notable time disparities even in pure water scenarios, as shown in Fig. 1e (FH surface cases).

To enhance the comprehensiveness of the introduction, we have expanded the text in the Introduction section:

“The three critical stages of bubble capture include collision after bubble rises, instantaneous rupture, and absorption at equilibrium, corresponding to drainage, solid-liquid-gas phase contact line formation, and contact line spreading, respectively^{17,18}. Among these, accelerating drainage between the bubble and surface is the most crucial for enhancing capture efficiency, as this process can vary by two orders of magnitude across diverse surfaces even in pure water, and by up to 5~6 orders of magnitude due to both the liquid composition and the physicochemical property across surface^{19,20}. Therefore, we define and focus on the capture time as the interval from the bubble’s initial contact with the surface to the formation of a spreadable three-phase contact line^{17,21}.”

5. Introduction: Employing diverse techniques such as electrodeposition, femtosecond laser, and liquid flame spray to refine nanostructures between 60 and 200 nm^{13,15,22} has further shortened the capture time to a current minimum of 2 ms.

I now recognize that the analysis of absolute rupture time does differ across different literature sources (author’s original references of 13, 15, 22). For instance, in this work, the authors use the point of initial contact, up until a spreadable contact. In other works, some authors use a “more conservative” estimate of having the largest bubble center-mass motion. Is it possible to align them by downloading the supporting information videos and then abstract the rupture time the same way for each reference project. Thereafter, a global plot of recent (and past) research’s achievements vs. this manuscript’s findings would provide a very informative plot towards the true advancements.

Response: We thank the reviewer for this very constructive comment.

As point out by the reviewer, the definition of capture time is not uniform. Here we summarize some common definitions in Table R1. Our definition aligns more closely with the objectives of our study and is also widely adopted in previous works.

Table R1. Summary of capture time definitions.

	Article	Capture Time Definition	Detection Method
1	Advanced Materials Interfaces, 2020, 7, 1901599	The time from the initial deceleration from terminal velocity to the point at which the surface captures the bubble.	
2	Advanced Materials, 2021, 33, 2101855	The interval from the minimum decelerated velocity to the sharp increase in velocity detected.	The velocity method
3	Nature Communications, 2021, 212, 5358	The time between contact (determined by the local minimum of the velocity–time graph) of the bubble with the surface and its rupturing (determined by the local maximum of the velocity–time graph).	
4	Soft Matter, 2019, 15, 5819-5826	The instance of bubble detachment from the tip of the syringe needle to a sudden decrease in the bubble contact angle.	
5	Physics of Fluids, 2016, 28, 057103	*The time of the TPC formation is the time span from the moment of the first collision to the three-phase contact line formation.	
6	International Journal of Mineral Processing, 2007, 81, 205–216	*The time of attachment, i.e., the time period from a moment of the first collision to the TPC formation.	The visual method
7	Physical Chemistry Chemical Physics, 2013, 15, 2586-2595	*The time of the TPC formation is the time interval from the moment of the bubble first collision with the surface until the TPC perimeter assuring the bubble attachment is formed.	
8	Langmuir 2009, 25(24), 14129–14134	*The time that the air bubble just contacted the lotus leaf surface before deformation to this complete spreading-out process.	

* Although different terms are used, these definitions of capture time are essentially the same as ours.

Based on the applied definition of bubble capture time in our work, we re-extract the capture

times from previous papers using images or videos, and provide a unified capture time in the following Table R2 and the updated Fig. 1f in the revised manuscript. Yet we acknowledge that complete alignment may not be feasible due to limited raw data sources provided in some literature, and we have endeavored to adopt consistent statistical standards. This statistical difference is mentioned in Fig. 1f caption, and the revised SI is enriched by these data, allowing the reader to have a clearer understanding.

Table R2. Summary of bubble capture time in previous work

Classification	Article	Materials	Structure Characteristic Size* (mm)	Bubble Diameter (mm)	Unified Capture Time (t_c , ms)	
1	Porous membrane	Soft Matter, 2019, 15, 5819-5826	Hydrophilic porous membrane + hydrophobic nanoparticle coating	0.005	2.6	4.0
2	Nano-coated plate	Physical Review Letters, 2016, 117, 094501	Glass substrate + hydrophobic nanoparticle coating	0.0001	2.0	3.1 – 146.2
3	PTFE plate	Physics of Fluids, 2016, 28, 057103	PTFE substrate	0.0050	1.5	115.0 ± 17.0
4	Rough Teflon surface	International Journal of Mineral Processing, 2007, 81, 205–216	Sandpapered commercial Teflon substrate	0.0500	1.5	2
		Physical Chemistry Chemical Physics, 2013, 15, 2586-2595	Sandpapered commercial Teflon substrate	0.0050	1.5	105.0 ± 4.0
				0.0600		37.0 ± 3.6
				0.1000	2.0 ± 0.5	

			Micropillared Silicon substrate + silane modification	0.0050	2.1	41.0
				0.0100	2.2	145.0
		Advanced Materials Interfaces, 2020, 7, 1901599		0.0500	2.0	30.0
			Microconical Silicon substrate + nano-coating	0.2040	2.6	5.7
				0.0440		2.0 – 5.0
5	Micro-nano surface	Advanced Materials, 2021, 33, 2101855	Glass substrate + nanoparticle deposition	0.0305	1.8	2.0 – 22.8
				0.0163		2.0 – 41.3
				0.0126		2.0 – 42.5
		Nature Communications, 2021, 212, 5358	Glass substrate + powder coating	0.0500	2.0	1.0 – 10.0
		Langmuir 2009, 25(24), 14129–14134	Silicon substrate + etched nano texture + chemical modification	0.0150	3.0	4.4
6	Nanofiber mat	ACS Omega 2022, 7, 39959–39969	PMMA nanofiber	0.0006	4.0	203.0

* Structure Characteristic Size S_c is the maximum characteristic structural size on each surface.

6. Introduction: However, despite nanoscale dimensions being reduced by 1000 times compared to microscale structures through various complex fabrication processes, the reduction in capture time has not shown proportionally significant decrease.

I noted that this claim of lack of further reduction in “capture time” (or rupture time?) is not backed by references. Perhaps this is a good time to include some references detailing performance of both micro- and nano-structured surfaces.

General definitions: Terms such as "capture stability" and "capture time" are used frequently

but are not well-defined. If the authors believe that they are measuring film rupture, perhaps it should be “rupture time”. My considerations may be border on semantics, but it appears that “capture” as a definition might be more suitable for having the bubble completely absorbed into the surface, therefore “captured”? On this note, perhaps there is some further data that can be presented in terms of the rupture time and the capture time?

Response: In accordance with the reviewer’s suggestions, we add references (Adv. Sci. 2024, 2403366; Adv. Mater. Interfaces, 2020, 7, 1901599, Adv. Mater., 2021, 33, 2101855) that detail the performance of both micro- and nano-structured surfaces.

While capture time t_c was already defined in the Introduction as the interval from initial contact to forming a spreadable three-phase contact line, we have now explicitly defined "capture stability" as “the coefficient of variation in capture time” in the revised Introduction. In this way, we can quantify how effectively the bubble can be captured on the surface. These definitions have also been widely used in previous studies (Physical Review Letters, 2016, 117, 094501; Soft Matter, 2019, 15, 5819-5826; Langmuir 2009, 25(24), 14129–14134).

Regarding the term “rupture”, which is more like an instantaneous event followed by absorption, we feel that retaining the use of "capture time" is more appropriate, as it clearly and accurately represents the phenomenon of capture or non-capture (bouncing) we aim to describe. While different terms or definitions are used in the literature to describe similar processes, we recognize that each study operates within its unique context and purpose, leading to variations in definition and terminology, such as “capture time”, “rupture time” and “three-phase contact line formation time”. After careful consideration, we believe it is more appropriate to retain the original definition of “capture time” to ensure coherence and consistency throughout the manuscript. In addition, as show in Table R1, this definition allows for meaningful comparison with a wide range of results reported in the literature. We have summarized the data as a unified capture time in previous works, presented in Table R2 and Supplementary Table 1 of the SI.

7. Results: “Coated with hydrophobic nanoparticles...”

I believe this description does influence how this manuscript is presented. If authors claim that only microstructural geometries are responsible for the various observations, it violates the accuracy of description. Therefore, I strongly insist that the authors revise both the title and main claims in abstract, and introduction, to avoid misleading statements.

Response: We appreciate the reviewer's critical comments. The need for a more accurate description of our surface is fully recognized. We have thus revised the title, and main claims in the abstract, introduction, and conclusion.

We would like to take this opportunity to give a brief clarification. It was not our intention to omit the discussion on nanoparticles; rather, our use of the term “aerophilic” in the original title

was meant to emphasize the necessity of forming a plastron on the microstructured surface for instant bubble capture and subsequent absorption. However, the formation of this plastron does not necessarily require nanoparticles. For example, the suggested paper (Adv. Sci. 2024, 2403366) demonstrates bubble rupture and absorption on microstructures functionalized with a perfluoroalkylated silane, rather than using the nanoparticle coating technique. Although our study focuses on different aspects, we share the same argument that the formation of a stable gas layer is needed, and the nanoparticles are not the only method to achieve this. In our response to Comment 13, we have presented additional experiments to show it.

8. Results: “Reduction of over 47 times from the 71.5 ms observed on a flat hydrophobic (FH) surface modified with nanoparticles only, where the bubble rebounds four times upon first impact.”

In Figure 1d, I noted that the bubble does not experience complete absorption into the nanostructured, albeit macroscopically flat surface. Even at 80 ms. This contrasts what others have found with nanostructures. Could this be caused by the high bubble approach velocity? Typically, a smaller approach velocity results in almost instantaneous rupture with nanostructures.

We also see here the most interesting observation, where the authors show rebounding bubbles. however, this is an induced condition - i.e., creating the trapped film of liquid between the bubble and the solid using high speeds of contact. At lower approach velocity, it is very likely that minimal differences will be seen. The authors should 1) indicate the velocity of bubble approach, and 2) try much lower velocities of contact (avoiding the bouncing effect) to prove or disprove this point: i.e., what is the effect of approach velocity on bubble rupture behavior?

Response: We thank the reviewer for the detailed observations and thoughtful insights. Regarding the concern about absorption on the control nanostructures in Fig. 1d, the bubble fully spreads and reaches equilibrium within ca. 183.8 ms, as shown in Fig. R1. As noted, since we are capturing free-rising bubbles with high approach velocity, which are inherently more difficult to capture, the interface between the bubbles and the plastron through coalescence is likely to oscillate relatively violently, which can be clearly observed from Supplementary Movie 2. This oscillation, in turn, may influence the rate of absorption. However, this final state of bubble absorption in the control experiments is not presented in the original manuscript, as such cases of repeated bubble bouncing are already recognized as a failure in rapid capture. Therefore, further concern about how quickly the bubble absorbs after rupture may not be required, as the primary objective of the control experiments is to highlight their limited effectiveness, rather than to delve into the specifics of their absorption dynamics.

Fig. R1. The optical image of a bubble after complete absorption into the nanostructured macroscopically flat surface.

We appreciate the reviewer’s insightful emphasis on the importance of approach velocity in bubble rupture behavior. It is certainly a critical factor that deserves further clarification. To investigate this effect, we conducted two groups of experiments. The first group still focuses on the issue of free-rising bubbles, providing supplementary evidence to the original manuscript. We set up four cases with different bubble release distances, 20 mm, 15 mm, 10 mm, and 5 mm, labeled RD-20 to RD-5, to achieve varying approach velocities. Each set of experimental conditions is repeated 20 times. Results in Fig. R2 indicate that the capture time on the MA surface remains relatively stable at around 1.5 ms, which is still significantly shorter than that on the FH surface under different velocities. These results have been added within Supplementary Fig. 3 of SI.

The second group of experiments, designed to avoid the bouncing effect as requested by the reviewer, shifts from the free-rising bubble capture issue to a fixed bubble approaching the surface, analogous to the investigation of plastron-induced bubble coalescence reported in the reviewer’s referenced paper (Adv. Sci. 2024, 2403366). The approach velocity is estimated to be less than 1 mm/s, and we ensure that no bubble bouncing occurs. Our findings (Fig. R2b) align closely with the reviewer's expectations, showing minimal differences between the MA and FH surfaces.

Fig. R2. a Capture time of free-rising bubbles on MA and FH surface under varying releasing distances, i.e. 20 mm, 15 mm, 10 mm, and 5 mm, labeled as RD-20 to RD-5, to achieve varying approach velocities. Bubble bounce will still occur on FH surfaces in these cases. **b** Capture time of needle-fixed bubbles on MA and FH surfaces under approaching velocity < 1 mm/s. No bubble bouncing occurs. The data box displays the mean value with standard deviation, and the black vertical line reaches the maximum and minimum values.

9. I noted that the authors only repeated their experiments 5 times, as indicated in the captions of Figure 1. It is therefore difficult to conclude that the process is not as statistical as what others have described before. I recommend a single statistical test of between 50 - 100 rupture events to visualize the spread of data while better justifying their claim. For strong claims like these that covers the fastest times reported, this is not only essential, but important. For instance, some of the cited studies in this manuscript perform at least 20 or more rupture experiments to identify statistical significance.

Response: We thank the reviewer very much for the valuable comment. To ensure statistical significance, we perform substantial additional experiments to guarantee 20 times for each case in updated Fig. 1e, Fig. 1f, Fig. 3h, and 100 rupture events on the MA surface shown in Fig. R3 (Supplementary Fig. 7 in the revised SI). The spread of data indicates that capture time on the MA surface can indeed stabilize at $1.5 \text{ ms} \pm 0.1 \text{ ms}$, with the coefficient of variation being less than 10%. We hope these results provide further confidence in the enhanced reliability of our conclusions.

Fig. R3. Statistical tests of capture time for 100-repetitive bubble capture events on the same location of the MA surface. The marginal plots show distribution curves of bubble diameter D_0 (averaged in 2.3 mm) and capture time (averaged in 1.5 ms). Coefficient of variation in t_c is 9%. The microcone array parameters on the MA sample surface is $b/a = 1.7$,

$L = 1$ mm. The bubble released distance is 20 mm.

10. One contribution to why the apparent rupture time does not decrease beyond a few milliseconds is because of detection algorithms (see author's references 13 by Vollmer – automated detection or 20 by Jiang – visual detection). These are often limited by the way automated detection is performed (one common method is detecting when the rapid and sudden shape change of the bubble occurs). This typically results in a jump in center-mass location. However, due to the nature of study (high speed cameras), this is often still limited. For examples, at 10000 fps, one is limited to the resolution of 0.1 ms. Therefore, under a moving average analysis of 10 frames, the resolution is limited to 1 ms. – i.e., a performance plot using other manuscripts' supporting information videos etc. While visually identifying rupture is perhaps more accurate, it becomes challenging to determine reliably where to start timing and when to end (a process that has been so-far automatized by researchers before with velocity detection algorithms). Notably, Quere also published a timescale of 0.16 ms (see author's reference 18, Figure 4a), but this depends on when the contact time is. Limitations in these methods, and why the authors chose their current method, should also be commented on.

Response: We thank the reviewer for the insightful comment on the detection methods for determining the capture time. Most of our experiments were recorded at a frame rate of 4000 fps, resulting in a time error of ± 0.25 ms due to the capture of the start and end moments within 1 frame. For the short capture time of 1.5 ms observed on the MA surface, this corresponds to a maximum deviation of 16.7%, which decreases to less than 1% on the FH surface. Given our efforts to standardize capture times across different studies as much as possible (Table R1), we believe the reliability of our comparative results is robust.

Regarding the 0.16 ms timescale in Quere's publication (reference 18, Figure 4a), it appears that this time does not represent the actual capture time, as our observation from Supplementary Information Movie 5 of the article reveals that the duration from the moment the bubble touches the surface to its bursting is approximately 1 second in the video. We think that the 0.16 ms timescale merely represents two frames before and after the bubble bursts.

Regarding the detection methods, we agree that the two main detection methods are the velocity method and the visual method. The distinctions between the two and the reasons for our choice of the visual method are outlined below:

- The velocity method defines the start time when the bubble velocity begins to decrease or reaches zero, and the end time when the bubble experiences a sudden change in velocity due to capture. While this method is relatively automated and allows for quantitative detection, it can lead to inaccuracies. This is because the bubble may not exhibit a noticeable shift in the center of mass at the moment of rupture. Additionally, on the MA surface, the bubble continues to ascend even after it contacts the cone tip, and its velocity

does not decrease to zero. Therefore, it is also difficult to accurately determine the starting time based on velocity changes.

- The visual method generally identifies the start time when the bubble first touches the surface and the end time when the three-phase contact line forms (typically marked by a sudden decrease in the bubble contact angle). It is more accurate than the velocity method because the change in velocity may lag behind the bubble rupture event. While this method may be subject to some error due to limited spatial resolution, the error is basically less than 1 ms. Therefore, we find that the visual method is more accurate and reliable for our study.

11. “Then the drainage dynamics of liquid film can be described by determining the drainage rate based on the lubrication theory³¹, i.e., $\partial h/\partial t = 1/3 \mu r \partial^2 r / \partial r^2 (r h^3 \partial P / \partial r)$, where μ is the liquid viscosity (detailed solving procedure is shown in SI note 2).”

The description of the slip condition is not included. Authors should justify when and why they use a certain slip condition here, which in turn, influences the prefactor of this equation.

Response: We thank the reviewer for giving us the chance to clarify our theoretical model. We have included the description of the slip condition in the revised SI Note 2. Our aim was to consider the thinning process of the liquid film within a microstructural unit and to determine the drainage rate of the film using lubrication theory. In this case, the lower boundary of the liquid film is the bubble surface, and there is no added medium in the water column, so the pure bubble surface can be approximated as the slip boundary. Meanwhile, since the model considers bubbles smaller than the spacing of the microstructure, and the upper boundary of the liquid film is the gas-liquid interface of the gas layer inside the microstructure, then the upper boundary is approximated as the free liquid surface, i.e., no slip boundary condition (Langmuir, 2019, 35: 8294-8307). Based on the boundary conditions in the lubrication theory, the corresponding prefactor of this equation is chosen to be 1/3 in our work.

12. One primary area I noted was, that in the experiments, bubble bounce significantly off the surfaces (particularly the FH surfaces, See Figure 1D). this would invalidate the timescale computed by the SRYL equation since a bubble that has lifted off no longer satisfy the assumptions of a thin film that SRYL requires. In that sense, is it still possible to make this experimental to theory link?

When bubbles bounce off surfaces, the equation cannot be used alone without accounting for the bouncing time during which the thin film is not established and draining. Therefore, most of the research in this area that even make attempts to link these aspects try to have a bubble come in contact and have its film slowly thin. Bouncing dynamics must be included and explained alongside the SRYL.

As discussed before, the solution is to perform a slow and gradual contact between a bubble and the test surfaces and report these results experimentally. I personally suspect that differences will be much lesser in those cases since rupture will be defined by the smallest length scale of contact, which is identical in both cases. Nonetheless, it does not mean that the findings do not have novelty, as rupturing of high-speed bubbles remains a highly meaningful endeavor. I feel that the primary message of this manuscript is better placed there.

Response: We thank the reviewer for the thoughtful comments. There are several points embedded in this comment, for which we would like to address one-by-one.

We understand the reviewer's concern that the drainage process during the rebound of the bubble is not applicable to the SRYL model. Actually, SRYL is only applied to the initial approach and contact period with the surface, rather than the subsequent rebound stage. This focus is based on our observation that bubbles are captured upon their first contact on MA surfaces, whereas FH surfaces exhibited bouncing after first contact. Therefore, the moment of first contact is the key point, and the SRYL model is sufficient to explain the differences in bubble behavior at the moment of first contact between the two surfaces. The subsequent bouncing dynamics, which occur after the bubble's initial contact with the FH surface, are treated as a control experiment, so we did not specifically analyze them theoretically.

Concerning the bubble capture experiments with a slow and gradual contact, as discussed in our previous response to Comment 8, this scenario is highly analogous to the investigation of plastron-induced bubble coalescence reported in the previous study (Adv. Sci., 2024, 2403366). It should be noted that this manually controlled quasi-state bubble motion unintentionally manipulates the drainage velocity before bubble rupture, thereby inadvertently eliminating the difference in drainage rates brought about by FH and MA surfaces. This drainage rate between bubble and plastron, as along with the corresponding plastron deformation profile, bubble deformation, and even the bubble rising velocity, have already been demonstrated to be influenced by microstructures as early as the moment the bubble begins to release, well before it makes contact with the surface. The experimental evidence and quantitative analysis of these key variables have been presented in the sections "Enhanced water film drainage within a characteristic unit" and "Threshold criteria for bubble capture". We are therefore concerned that this slow contact between the bubble and the test surfaces may act as an intervention with the system of plastron, bubble, and the intermediate liquid film, thereby no longer addressing or explaining what happens before the moment of bubble rupture, which is the key to distinguishing between bubble capture and non-capture. Nonetheless, when it comes to the instant of bubble rupture, we appreciate the reviewer's valuable insight. Since both the MA and FH surfaces have the same smallest contact length scale, the rupture times are expected to be similar. Our results (Fig. R2b) show minimal differences, as anticipated. We also believe this is the focus of the previous study (Adv. Sci., 2024, 2403366), which helps differentiate our

work.

Additionally, from an application perspective, most cases involve the capture of free-rising bubbles. If a bubble is not captured on first contact, it is likely to be carried away by the water flow, making subsequent capture unlikely. Therefore, we believe that our focus should remain on exploring the capture mechanisms that incorporate how the surface features influence the bubble-rising process, rather than on discussing the rupture of high-speed bubbles.

13. “These findings emphasize the key to stable capture efficacy: when bubbles touch the inner wall of the microstructure and form a stationary pinned TPL, the constrained deformation of the gas layer is created, thereby accelerating drainage.”

Ultimately, it remains unclear if the rupture can be attributed solely to the microstructures or the nanostructures. The use of the microcones certainly forces the bubble to interact with the nanostructures in a way that was not possible with macroscopically flat variants. However, conventional knowledge would indicate significant contributions by the nanostructuring (see Adv. Sci. 2024, 11, 2403366). To prove or disprove this hypothesis, the use of undecorated cones alongside slower bubble approach velocity can provide further insight.

Response: We appreciate the reviewer's insightful comments. Although the focus of our analysis is on the evolution of the system before rupture, we fully respect the reviewers' suggestions for further explanation of the roles of the micro and nanostructures at the moment of rupture. We have attempted to organize our perspectives from a combination of original results, new experiments, and the referenced paper (Adv. Sci., 2024, 11, 2403366).

The two new sets of control experiments are shown in Fig. R4b and c. One set involves an undecorated microstructured surface exhibiting hydrophilicity (contact angle = $72^\circ \pm 2^\circ$). The other involves a silanized microstructured surface exhibiting super-aerophilicity but not stable (no nanoparticle, contact angle = $153^\circ \pm 1^\circ$); this metastable Cassie-state will easily collapse into the Wenzel-state when submerged at water. These new experiments, as well as the original ones, i.e. nanostructured surface ($155^\circ \pm 2^\circ$) and nanoparticles coated microstructured surface (contact angle = $154^\circ \pm 2^\circ$) as shown in Fig. R4a and d, are used to clarify the raised issue.

At the moment of bubble rupture rather than the entire process of bubble capture:

- The fundamental mechanism of bubble rupture is the same on nanoparticle coated microstructured surfaces and nanostructured surfaces. Ultimately, it all comes down to the localized high pressure created by the nano extrusions. This leads to the thinning of the liquid film between the plastron and bubble to a critical size, causing the film to rupture, followed by gas bridge formation and bubble coalescence. This mechanism have been widely accepted.
- When microstructures are not coated with nanoparticles but are instead functionalized with

silane to create a chemically super-aerophilic surface, the bubble can also rupture rapidly ($t_c = 2.1 \text{ ms} \pm 0.6 \text{ ms}$ under hydrostatic pressure of 19.6 Pa and $t_c = 2.6 \text{ ms} \pm 0.3 \text{ ms}$ under hydrostatic pressure of 98.1 Pa, Fig. R4c), compared to the nanoparticle coated microstructure ($t_c = 1.5 \text{ ms} \pm 0.1 \text{ ms}$, Fig. R4d). The surface of the microstructure is not atomically smooth but rather has some roughness. Although this microscale roughness does not generate as significant local pressure as the nanoparticles, it still helps stabilize the gas layer and the three-phase contact line, allowing the bubble to rupture at the three-phase contact line on the inner wall of the microstructure. Interestingly, the referenced paper (Adv. Sci., 2024, 11, 2403-366) reports a similar finding, noting that bubble rupture is more likely to occur at the corners and edges of silanized micropillars. To extend the content of the original manuscripts, we have added an elaboration of the role of nanostructures at the moment of rupture, as shown at the end of this response to Comment 13.

- Without hydrophobic treatment on the microstructures (Fig. R4b), a gas layer can't form, thus preventing bubble rupture entirely.

Now, we would like to return to the complete picture of bubble capture:

- Bubbles rebound on both nanostructured surfaces (Fig. R4a) and microstructured surfaces (Fig. R4b), suggesting that capture can not be attributed solely to the microstructures or the nanostructures.
- Bubbles can be captured instantly upon first contact with both silanized microstructured surfaces ($t_c = 2.1 \text{ ms} \pm 0.6 \text{ ms}$ under 19.6 Pa and $t_c = 2.6 \text{ ms} \pm 0.3 \text{ ms}$ under 98.1 Pa, Fig. R4c) and nanoparticle-coated microstructured surfaces ($t_c = 1.5 \text{ ms} \pm 0.1 \text{ ms}$, Fig. R4d), as both surfaces can form a gas layer (plastron), either thick or thin. We think this strongly supports our claims that microstructure, as well as stable plastron, are two necessary

Fig. R4. The high-speed image sequences of bubble behaviors on the three surfaces. **a** Hydrophobic flat surface with nanostructures. **b** Hydrophilic microstructured surfaces. **c** Super-aerophilic silanized microstructured surfaces without nanostructures. The substrate was first treated for plasma treatment (250 W, 180 s) to obtain surface-active hydroxyl groups after cleaning and dry, followed by a silanization reaction for 2 hours under the conditions of a silane concentration of 5.0×10^{-3} M (1H, 1H, 2H, 2H-perfluorodecyltriethoxysilane), a temperature of 110°C , and a reduced pressure of 0.2 atm. The surface distance from the liquid is tested at two values: i) 2 mm, corresponding to a hydrostatic pressure of 19.6 Pa, and ii) 10 mm, corresponding to a hydrostatic pressure of 98.1 Pa. **d** Super-aerophilic microstructured surfaces with both microstructures and nanostructures used in the manuscript. The bubbles with a diameter of 2.4 mm were identically released 2 cm beneath the test surface. All surfaces are tested at a depth of 10 mm underwater, except for Fig. R4c (ii).

conditions for rapid bubble capture. Although the plastron formed on the silanized surface appears to be metastable underwater and tends to be easily lost under hydrostatic pressure or long-time water immersion (Fig. R4c (ii)), leaving only a thin gas layer insufficient for complete bubble absorption, it is still sufficient for quick coalescence.

- It seems that the referenced paper (Adv. Sci. 2024, 11, 2403366) does not specifically conclude the significant contribution of nanostructures during the bubble rupture and spreading stages. Rather, it shows that nanostructures lead to faster contact line progression compared to microstructures, with both showing similar effects on rupture and spreading. This focus on rupture and spreading differs from our emphasis on the film drainage process before rupture. Therefore, we believe that our work, together with theirs, collectively provides a comprehensive picture of the bubble capture process.

These analyses do not change the mechanisms proposed in our manuscript. We support the perspective that microstructures have a more significant impact on the overall system (plastron, liquid film, and bubble) during the rise of free bubbles when a gas layer can form, while nanostructures serve to enhance the stability of this gas layer.

Based on the clearer understanding provided by these new experiments, we have accordingly revised and improved the corresponding claims regarding Fig. 2g raised by the reviewer, particularly the role of nanostructures in stabilizing the gas layer under our mechanism.

“These findings show the necessary condition for consistent capture efficacy: rising bubbles must contact with the stationary pinned TPLs formed by the encapsulated gas and microstructure, located on the inner wall. At the moment of rupture, the effect of nanoparticles or nanostructures remains consistent across both MA and FH surfaces due to their comparable minimum feature sizes. On the other hand, during the drainage phase before rupture, the microstructure irreplaceably contributes to confining the gas to a constrained space, within which significant deformation of the gas layer can occur, thereby accelerating drainage.”

14. “This enhancement stems from increased drainage rates within the cone array above the bubble, where the fluid velocity u of outward flow in the liquid film is influenced by cone geometry.”

The PIV technique is indeed quite an interesting characterization method to understand microscopic flow. However, I could not understand why a smaller bubble (red squares vs black diamonds), as represented by the dimensionless number, would lead to a higher velocity. Isn't there less liquid to push out and hence the flow velocity should be smaller? I ask this because the physical intuition illustrated above is running in contrary to the findings.

Response: We thank the reviewer for the appreciation of the PIV experiments presented in our

study. Regarding the question about the apparent observed higher velocity of smaller bubbles (represented by red squares) compared to larger bubbles (black diamonds), the key to understanding this phenomenon lies in the interplay between bubble size, buoyancy force, and drag force. The force balance model is established to determine the motion of the rising bubble, where buoyancy force F_B , drag force F_D , added mass force F_A , and film drainage force F_F are considered in this process. Due to the negligible density of gases relative to liquids, assuming the mass of bubble is zero, the equilibrium of the four forces is represented as

$$F_B + F_D + F_A + F_F = ma \approx 0.$$

Smaller bubbles experience less drag due to their reduced cross-sectional area, overcoming the drag force more efficiently. As a result, they tend to rise faster than larger ones, leading to the overall higher velocities V_m , which is conducive to liquid drainage. We hope this explanation addresses the concern and clarifies the underlying physics behind the observed trends.

15. “Elongated microcones, with a higher b/a , not only create a thicker gas layer that expands deformation space and enhances interface curvature, thus accelerating drainage, but also increase the bubble’s maximum velocity V_m before contact (Fig. 3B), contributing further to the overall thinning of the liquid film.”

I found that this is likely the primary contribution to the literature as this shows how microstructural texturing is able to enhance nanostructure efficacy in bubble rupture under high We or Bo number conditions, where bubbles are moving much faster than those stated in the current literature. More focus should be placed here, instead of the unwarranted focus on “simply microtextured” which clearly isn’t the case.

Response: We sincerely appreciate the reviewer's valuable comments on our work. As outlined in our detailed discussion in response to Comment 13, we agree that both microstructures and nanostructures contribute to the overall bubble capture process. However, during the drainage stage, microstructures are the dominant factor, primarily through their role in enhancing film drainage. While the reviewer correctly highlighted the fast bubble capture under high We or Bo number conditions as a significant contribution, our core objective has always been to uncover a novel mechanism by which microstructures accelerate bubble capture efficiency through optimizing liquid film dynamics. It is true that the relatively severe high We or Bo number conditions we employed more prominently indicate the critical role of micro-textures in the liquid film thinning process (as also detailed in our response to Comment 8 and Fig. R2), but we believe this conclusion does not deviate from the original focus of our study.

In the revised manuscript, we have expanded on this core mechanism with additional data and analysis to strengthen our conclusions. We have also refined the terminology to ensure clarity and avoid misunderstandings about the role of microstructures in liquid film dynamics.

16. “We establish a scaling law that connects the dimensionless terminal deformation of bubble

$\eta_{\text{terminal}}/D_0$ with energy conversion efficiency α , defined as $\eta_{\text{terminal}}/D_0 \sim (\alpha We^*)^{1/2}$, where $We^* = C_m \rho D_0 V_m^2 / \sigma$.”

Scaling laws are often rooted in more broad theories. can the authors illustrate where this stems from? Either here or in the supporting information.

Response: The scaling law is rooted in the conservation of energy. A fraction of kinetic energy at terminal velocity V_t is transformed into the bubble’s surface energy during the bubble’s approach:

$$\alpha E_k = E_\sigma ,$$

where $E_k \sim \rho^* D_0^3 V_t^2$ and $E_\sigma \sim \gamma \eta_{\text{terminal}}^2$. Here, α denotes the energy transfer efficiency, ranging from 0 to 1, η_{terminal} is the ultimate deformation of the bubble contacting the surface, and $\rho^* = C_m \rho$ is the density considering total inertia in the motion of the bubble, where ρ is the liquid density and C_m is the the added mass coefficient. Accordingly, substitute E_k and E_σ into the formula, we derive that $\eta_{\text{terminal}}/D_0 \sim (\alpha We^*)^{1/2}$, with $We^* = \rho^* D_0 V_t^2 / \gamma$.

This content is added in the revised SI Note 4.

17. “Notably, the rate of film thinning, dh/dt , and the liquid film radius, $R_b = D_b/2$, inversely relate as $-dh/dt \propto R_b^2$, according to the classical Scheludko equation.”

Interesting, this correlation would indicate that planar drainage (so called Reynold's flat film model) would be dominant, and that the drainage occurs as though a flat film is present. Ultimately, at the speeds of bubble approach involved - 40 mm/s, one would expect a macroscopic dimple (Soft Matter, 2011, 7, 2235). However, other works cited by the authors have also suggested that local film thinning can occur, thus maintaining the flat film model. Can the authors further describe what their findings suggest? Where does drainage dominate?

Response: We thank the reviewer for the insightful comment on the drainage model. For the relation $-dh/dt \propto R_b^2$ discussed here, we solely use this to quantitatively infer the trend of liquid film thinning under the variation of bubble deformation. The classical Scheludko equation is derived according to the Stefan–Reynolds flat film model. However, the flat film model is widely recognized as inaccurate for calculating the evolution of the liquid film (Soft Matter, 2011, 7, 2235–2264; Adv. Colloid Interfac., 2016, 235, 214-232). On one hand, we agree that bubbles will deform as macroscopic dimples at high rising velocities. On the other hand, the local film thinning suggested by the cited papers mainly results from the topographical feature (Adv. Colloid Interfac., 2009, 147–148, 155–169; Phys. Chem. Chem. Phys., 2013, 15, 2586-2595; Nat. Commun., 2021, 12, 5358). When the solid surface exhibits roughness, e.g., pillars and protrusions across distinct localized regions, the liquid layer thinned between a colliding bubble and the surface can locally, specifically atop the protruding features of the rough surface,

attain a thickness that is substantially smaller than the evaluated average value. These findings suggest that the protruding features, especially microstructures, can change the film thickness distribution, thus influence the time required for the liquid film to drain.

Therefore, we use the Stokes-Reynolds-Young-Laplace equation, instead of the flat film model, to quantitatively obtain the spatio-temporal evolutions of the film thickness and the drainage rate within a characteristic microstructural unit, which has not been unveiled before.

In terms of where drainage dominates, our results indicate that drainage is a combination of both macroscopic and local effects. The macroscopic aspect is mainly influenced by the overall bubble dynamics and approach speed. However, the local effects, such as bubble deformation and structure-caused local thin film, play a crucial role in determining the accurate detailed evolution of the liquid film.

18. “This is quantified by a critical pressure threshold, ΔP_{cr} , which is derived by establishing a mechanical equilibrium between the interfacial tension force and the liquid film pressure exerted on the gas layer (Fig. 3E): $\Delta P_{cr} = \sigma \cos(\pi + \alpha - \theta_{adv}) R_c$, (1)”

The derivation of equation 1 is not entirely clear. Is there a reference or is there a derivation somewhere in the supplementary materials?

Response: We have provided a detailed derivation in the revised SI Note 5.

For the microconical surface, the three-phase contact lines pinned on microstructures provide a force that can resist pressure, which can be expressed as

$$F_{\sigma} = 2\pi a_c \sigma \cos \theta_E, \quad (S10)$$

here a_c is the cross-section radius of the microstructure where the three-phase contact line formed. $\theta_E = \pi + \alpha - \theta_{adv}$, is the angle between the interfacial tension force F_{σ} and the vertical axis, where α equals to half the apex angle of the cone, and θ_{adv} is the advancing angle of the surface. In a structural unit, the critical pressure can be obtained by dividing the three phase wire pinning force by the area of the gas-liquid interface, written as

$$\Delta P_{cr} = \frac{F_{\sigma}}{A} = \frac{2\pi a_c \sigma \cos \theta_E}{L^2 - \pi a_c^2}. \quad (S11)$$

Based on equation S11, the general Laplace expression can be evolved as

$$\Delta P_{cr} = \frac{\sigma \cos \theta_E}{R_c}. \quad (S12)$$

where $R_c = A/C$ is the capillary radius (36) of meniscus' spatial curvature with the projected area of the liquid-gas interface $A = L^2 - \pi a_c^2$, and the length of the three-phase contact line $C = 2\pi a_c$.

Reviewer: 2

This manuscript, wherein the bubble capture of aerophilic conical microstructures is analyzed, does demonstrate cutting-edge air capture rates. This work has clear significance, for example their microconical surfaces can clearly capture and transport more gas underwater than an equivalent aerophilic flat surface, which the authors show through experiments.

Moreover, I was on the whole impressed with the quantification of the gas layer and interface profiles, particularly as it seemed to show fairly good agreement with actual experimental results. Additionally, I found the experiments quantifying bubble capture dynamics to be well done.

Response: We sincerely thank the reviewer for the appreciation of this work.

1. However, I do have some qualms with the manuscript. An argument you pose in the introduction is that microstructural features 'overshadow' the role of nanostructure for bubble capture. You don't prove that. You prove that a nanostructured surface with a corresponding microstructure performs better than a nanostructured surface without that corresponding microstructure. That is to say, you prove that microstructure is relevant to bubble capture. For you to prove that microstructure overshadows nanostructure, you'd have to find some way of isolating the effect from microstructure and the effect from nanostructure in the improved bubble capture of the microstructured surface, which you haven't done. The conclusion addresses this point appropriately, in that you state 'that rapid bubble capture does not necessitate on overly complex [...] nanostructures'. This you do prove.

Response: We thank the reviewer for the thorough examination and valuable feedback. We find that the claim of "overshadow" may have been somewhat overstated, so we have moderated the tone in the revised manuscript. We have also clarified that the effect of microstructures appears to diminish the contribution of nanostructures, particularly during the drainage stage of the bubble capture process. We hope these revisions help avoid potential misunderstandings.

Building on the constructive suggestions from the reviewer, which align closely with Reviewer 1's comments, we have conducted two sets of control experiments to further investigate the individual roles of microstructures and nanostructures (please see Fig. R4): (1) undecorated microstructured surface, (2) silanized microstructured surface (exhibiting super-aerophilicity but not stable). These new experiments are used to compare with our original results, including nanostructured surfaces and nanoparticles coated microstructured surfaces.

Fig. R4a and b demonstrate bubbles rebound on both nanostructured surfaces and undecorated

Fig. R4. The high-speed image sequences of bubble behaviors on the three surfaces. **a** Hydrophobic flat surface with nanostructures. **b** Hydrophilic microstructured surfaces. **c** Super-aerophilic silanized microstructured surfaces without nanostructures. The substrate was first treated for plasma treatment (250 W, 180 s) to obtain surface-active hydroxyl groups after cleaning and dry, followed by a silanization reaction for 2 hours under the conditions of a silane concentration of 5.0×10^{-3} M (1H, 1H, 2H, 2H-perfluorodecyltriethoxysilane), a temperature of 110°C , and a reduced pressure of 0.2 atm. The surface distance from the liquid is tested at two values: i) 2 mm, corresponding to a hydrostatic pressure of 19.6 Pa, and ii) 10 mm, corresponding to a hydrostatic pressure of 98.1 Pa. **d** Super-aerophilic microstructured surfaces with both microstructures and nanostructures used in the manuscript. The bubbles with a diameter of 2.4 mm were identically released 2 cm beneath the test surface. All surfaces are tested at a depth of 10 mm underwater, except for Fig. R4c (ii).

microstructured surfaces, suggest that individual nanostructures or microstructures cannot facilitate bubble capture upon the first contact with the surface. In contrast, bubble can be captured instantly upon first contact with both silanized microstructured surfaces ($t_c = 2.1 \text{ ms} \pm 0.6 \text{ ms}$, Fig. R4c, i) and nanoparticle coated microstructured surfaces ($t_c = 1.5 \text{ ms} \pm 0.1 \text{ ms}$, Fig. R4d), as both surfaces can form a gas layer (plastron), either thick or thin. We think this strongly support our claims that microstructure as well as stable plastron are two necessary conditions for rapid bubble capture. However, there are differences between the two surfaces capable of capturing bubbles. Since the plastron formed on the silanized surface is metastable and tends to be easily lost under hydrostatic pressure when immersed in water, it leaves only a thin gas layer sufficient for bubble coalescence but insufficient for complete bubble absorption ($t_c = 2.6 \text{ ms} \pm 0.3 \text{ ms}$, Fig. R4c, ii). On the other hand, bubbles could be fully absorbed on the nanoparticle decorated microstructured surface. Nevertheless, this difference in bubble absorption stage doesn't affect our claims on the microstructures' contribution during initial drainage stage of the capture process. As for the nanostructures, in terms of the core issue of bubble capture or non-capture that we are concerned with, they primarily contribute to stabilizing the gas layer, rather than being the only way of achieving super-aerophilicity thus forming plastron.

2. On a more surface level, significant chunks of the manuscript, particularly the introduction, include numerous grammar/language mistakes. Examples of poor phrasing include page 1 (starting from introduction) lines 15-16 'random porous' should be 'random pores' or 'random porous structures'. Likewise the page 1 lines 9-10 'the respiratory of diving insects' should be 'the respiration of diving insects'. These are not particularly egregious on their own, but they are just a couple examples of a recurring problem. Similarly, you refer to your methods section consistently as 'materials and methods'. On the whole, the writing of this manuscript should be reviewed and thoroughly edited.

Response: We thank the reviewer for carefully reviewing our manuscript. We have made a careful proofreading; all the issues pointed out have been thoroughly addressed and corrected in the revised manuscript.

Reviewer: 3

The targeted collection of bubbles from an aqueous environment is of current interest and important for various industrial and environmental applications. Whereas the bubble capture by the plastron sustained on superhydrophobic surfaces has been in the literature for some time, the present work is a detailed study building on some of the prior designs and presenting an optimized approach for fast bubble capture. The approach relies on the combination of nanoparticle superhydrophobic coating and microscopic conical features, inspired by the nature of *Salvina* leaves. The study is of high quality and covers various aspects of the surface's design and performance. It has the outreach and potential for a Nature Communications paper.

Response: We thank the reviewer very much for supporting this work.

1. I hesitate to use a record stating “capture time of 0.83 ms”. The capture times will vary from experiment to experiment. In different works it is defined in various ways, starting from different bubble positions and sometimes including and sometimes not the time for the bubble spreading on the surface, so one should be careful when comparing. Use more general statements such as “capture times down to about 1 ms” and explain how it was determined (time for the film rupture?).

Response: As the definition of capture time is not universal, we have chosen one that focuses on the drainage phase, which aligns better with the focus of this work. Based on the applied definition of bubble capture time in our work, we re-extract the capture times from previous papers using images or videos, and provide a unified capture time in the following Table R2 and the updated Fig. 1f in the revised manuscript.

Following the reviewer’s suggestion, we have explicitly explained the definition of capture time in the revised Introduction and modified the record stating “capture time of 0.83 ms” into the “capture times down to about 1 ms” in the Abstract.

Table R2. Summary of bubble capture time in previous work

Classification	Article	Materials	Structure Characteristic Size* (mm)	Bubble Diameter (mm)	Unified Capture Time (t_c , ms)
1 Porous membrane	Soft Matter, 2019, 15, 5819-5826	Hydrophilic porous membrane + hydrophobic nanoparticle	0.005	2.6	4.0

		coating				
2	Nano-coated plate	Physical Review Letters, 2016, 117, 094501	Glass substrate + hydrophobic nanoparticle coating	0.0001	2.0	3.1 – 146.2
3	PTFE plate	Physics of Fluids, 2016, 28, 057103	PTFE substrate	0.0050	1.5	115.0 ± 17.0
4	Rough Teflon surface	International Journal of Mineral Processing, 2007, 81, 205–216	Sandpapered commercial Teflon substrate	0.0500	1.5	2
		Physical Chemistry Chemical Physics, 2013, 15, 2586-2595	Sandpapered commercial Teflon substrate	0.0050	1.5	105.0 ± 4.0
				0.0600		37.0 ± 3.6
				0.1000		2.0 ± 0.5
5	Micro-nano surface		Micropillared Silicon substrate + silane modification	0.0050	2.1	41.0
				0.0100	2.2	145.0
		Advanced Materials Interfaces, 2020, 7, 1901599		0.0500	2.0	30.0
			Microconical Silicon substrate + nano-coating	0.2040	2.6	5.7
				0.0440		2.0 – 5.0
		Advanced Materials, 2021, 33, 2101855	Glass substrate + nanoparticle deposition	0.0305	1.8	2.0 – 22.8
				0.0163		2.0 – 41.3
		0.0126	2.0 – 42.5			
	Nature Communications,	Glass substrate + powder	0.0500	2.0	1.0 – 10.0	

		2021, 212, 5358	coating			
		Langmuir 2009, 25(24), 14129– 14134	Silicon substrate + etched nano texture + chemical modification	0.0150	3.0	4.4
6	Nanofiber mat	ACS Omega 2022, 7, 39959–39969	PMMA nanofiber	0.0006	4.0	203.0

* Structure Characteristic Size S_c is the maximum characteristic structural size on each surface.

2. What is the practical advantage of lowering the capture time from a few ms to under 1 ms – anyway there are slower stages of the processes than the time required for the bubble to reach the interfaces. I guess that the major time-shortening effect is the switch from bouncing before capture to capture without bouncing.

Response: We thank the reviewer for the insightful comment on the time-shortening effect. It is indeed true that for a given surface, the capture time without bouncing is shorter than that with bouncing. However, for surfaces that prevent bouncing, achieving a shorter bubble capture time is particularly significant. In practical applications, such as microfluidic devices, biosensors, or chemical reaction control, systems often demand high precision and rapid response. Reducing the capture time can greatly enhance the overall system's response speed and efficiency. Capturing bubbles within extremely short periods effectively mitigates flow disturbances caused by prolonged uncaptured bubbles, thereby reducing the risk of them being carried away unintentionally. This is crucial for maintaining the stability and uniformity of the flow field.

3. The ultrafast capture suggested is tested and works only for pure water. However, in practical situations as in flotation, there are often surfactant additives, natural or added. Some comments on how the surfaces will perform if such additives are present. The same holds for high electrolyte concentration water solution as seawater which is known to significantly inhibit the bubble coalescence times.

Response: We appreciate the reviewer's constructive comments, which give us the opportunity to validate the expandability of this ultrafast bubble capture environment. We fully recognize the potential impacts of surfactants according to current studies (Adv. Mater. 2023, 35, 2300306; Sci. Adv. 2019, 5, eaaw429), as they can alter the surface tension of the liquid, affecting the stability of the liquid film, and establishing no-slip boundary conditions. However, it is

important to note that the effect of surfactants is consistent across all surfaces, meaning that their influence would be the same for all surfaces involved in the comparison. As a result, this factor does not affect the comparison of bubble capture times across different surfaces, and we consider that this factor does not contradict the mechanism proposed in our study.

We have also conducted additional experiments on bubbles capture using the MA surface immersed in a high electrolyte concentration water (3.5% NaCl solution), resembling sea water. Fig. R5 presents the capture time distribution for 20 experiments under each set of conditions, with bubble diameters ranging from 1.6 to 3.5 mm. The results reveal that this environment has minimal impact on the ultrafast bubble capture for bubbles with radius less than 3.0 mm, with capture times consistently around 1.5 ms. However, as the bubble diameter increases to 3.5 mm, the capture time rises to approximately 3 ms (as noted by the reviewer). We attribute this primarily to the increased viscosity of the salt solution compared to pure water, which exerts greater resistance on the vertical motion of larger bubbles (Int. J. Chem. React Eng., 2023, 21, 701–715). Furthermore, larger bubbles have more surface area and are more likely to accumulating solutes, which may influence the stability of the liquid film to some extent (Adv. Colloid Interface, 2015, 222, 305-318).

Overall, the enhanced bubble capture efficiency remains effective in high electrolyte concentration water.

Fig. R5. Capture time with varying sizes of bubble from $D_0/L = 1.6$ to 3.5 on the MA surfaces under high electrolyte concentration water (3.5% NaCl solution). The data box displays the mean value with standard deviation, and the vertical line reaches the maximum and minimum values. Each condition is repeated 20 times with raw data distributed next to the corresponding data box.

4. Another major issue with using a microtexture supported gas layer is the tendency of the gas layer to dissolve with time. How robust are the current surfaces to high pressure and extended underwater time?

Response: We appreciate the insight comments highlighting the importance of the stability of the gas layer. The characteristic size range of our structure is within 0.2-1.8 mm, and gas layers on surfaces demonstrate robustness under a submerged conditions up to 5 cm of tap water (\approx 0.5 kPa static pressure). Optimizing our microstructures to improve resistance to hydrostatic pressure could potentially achieve performance records, e.g., the multi-layer hierarchical conical structure exceeding 300 kPa (npj Asia Mater., 2013, 5, e37). Currently, there has already been tremendous progress in this area (Adv. Sci., 2024, 11, 2308152; Adv. Funct. Mater., 2023, 33, 2206946), including our recent work (ACS Appl. Mater. Interfaces, 2024, 16, 26954–26964). However, since our focus is more on shallow-water and microfluidic devices, biosensors, or chemical reaction control applications rather than deep-water applications, we have not made further efforts specifically targeting hydrostatic pressure resistance.

In terms of another point raised by the review, the extended underwater time, we were already aware of the challenge of maintaining hydrophobicity in the real environment, such as flowing, and had performed experiments to investigate this issue. Our experiments demonstrated continuous bubble capture for up to 12 hours in a flowing environment, as detailed in the SI (Supplementary Fig. 26 in SI and Supplementary Movie 6). Throughout this period, a consistently high efficiency over 98% was observed. This performance is attributed to the continuous replenishment of the gas layer, which effectively counteracts dissolution and diffusion effects, thereby enabling long-term underwater hydrophobicity. These encouraging results provide a strong foundation for further exploration to enhance the stability and durability of hydrophobic surfaces for extended underwater applications.

5. The force balance model used to estimate the film drainage captures the major physical features of the system. However full numerical simulation might be more accurate. The interfaces' mobility is of great importance. See further discussions in:

I.U. Vakarelski, F. Yang, S.T. Thoroddsen. Free-rising bubbles bounce more strongly from mobile than from immobile water-air interfaces, Langmuir 36 (2020) 5908-5918.

I.U. Vakarelski, K.R. Langley, F. Yang, S.T. Thoroddsen. Interferometry and simulation of the thin liquid film between a free-rising bubble and a glass substrate, Langmuir 38 (2022) 2363–2371.

Response: We appreciate the reviewer's insightful comments on the numerical simulation and interface mobility. In our original manuscript, we have conducted a full numerical simulation of the drainage process using the two-phase MUSCL-SPH method and obtained the velocity field of the liquid film drainage, as presented in Fig. 2g and SI Note 3. Specifically, direct

discretization of the Navier-Stokes equations (conservation of mass, momentum, and energy) is achieved by SPH approximation. The modeling of water and air was achieved by assigning different reference mass densities, sound speeds, and adiabatic parameters within the corresponding geometric regions. In the momentum equation, the kinematics of fluid particles is dominated by the pressure gradients, surface tension force, gravity, and solid boundary constraints. The Tammann equation of state is introduced to decouple the continuity and momentum equations and the explicit predictor-corrector scheme is adopted for time integration.

Regarding the interfaces' mobility, we agree that interface mobility directly influences the drainage speed of the liquid film, significantly impacting the efficiency of bubble rise and capture. To emphasize its importance, we have added this point in SI Note 3 and cited as Supplementary Refs. [7] and [8] in SI. In our SPH simulations, we neglected the viscosity of the two-phase fluids, treating all interfaces (including gas-liquid and solid-liquid interfaces) as mobile interfaces with free slip, thereby eliminating the interference of interface mobility. This is because, in this system, the influence of capillary forces far outweighs that of viscous forces. The Morton number, $Mo = \Delta\rho g \mu_c^4 / \rho_c^2 \sigma^3$, is a dimensionless parameter describing the ratio of viscous forces to capillary forces in fluid dynamics, where $\Delta\rho$ is the absolute value of the density difference between the two phases, ρ_c and μ_c are the density and viscosity coefficient of the ambient fluid, g is the gravitational acceleration, and σ is the surface tension. In our study, $Mo \approx 3.7 \times 10^{-11} \ll 1$ ($\Delta\rho = 1000 \text{ kg/m}^3$, $g = 9.81 \text{ m/s}^2$, $\mu_c = 1.1\text{e-}3 \text{ Pa}\cdot\text{s}$, $\rho_c = 1000 \text{ kg/m}^3$, $\sigma = 7.28\text{e-}2 \text{ N/m}$), confirming that capillary forces dominate over viscous forces. We have provided a detailed description of the above analysis and the discussion of the interfaces' mobility in the revised SI Note 3.

6. Literature is generally well referred to. One more recent related work:

Wong, W. S. Y. et al. Designing Plastrons for Underwater Bubble Capture: From Model Microstructures to Stochastic Nanostructures Adv. Sci. 2024, 2403366

Response: We thank the reviewer for providing a valuable reference. We find the literature published recently indeed inspires us for the clarification of some claims in this work, and cite it as Ref. [27].

7. Last – should it be “simply microtextured aerophilic surfaces” or “simple microtextured aerophilic surfaces”?

Response: We thank the reviewer for the suggestions. The title is now revised to “*Ultrafast bubble capturing on microcone-textured super-aerophilic surfaces.*” This change aims to specify microcones used as simple textures, and indicate that “*super-aerophilic*” is intended as a juxtaposed condition to “microcone-textured” for achieving ultrafast bubble capturing.

Reviewer: 4

Hu et al. present an insightful study on bubble capturing, with a specific focus on the effects of microstructuration. Their research primarily examines gas bubbles ranging from $D_0 = 1.6\text{--}3.5$ mm and demonstrates that microstructuration with a 1 mm pitch can significantly reduce capture time by enabling bounce-free capture. The authors identify three distinct regimes—bounce, capture, and attachment—and develop a pressure threshold model based on bubble size and the substrate's advancing contact angle. This study is thorough, well-constructed, and presents valuable insights into the microstructuration regime, making it a commendable contribution to the field.

I believe the manuscript is suitable for publication in Nature Communications, as it 1) includes fundamental considerations about gas bubble capture, which not only lowers the limit of capture time but also offers a new perspective on efficient strategies for bubble capture, and 2) is urgent as it may redirect the current focus of the scientific community - from nano to the importance of micro - thereby potentially optimizing resource utilization. The findings are relevant to any field where gas capture is utilized and potentially for the wetting community as well.

Response: We thank the reviewer very much for the high appreciation of this work.

Major comments

1. Reading your manuscript, I believed for a long time that your surface was purely microstructured—there is no mention of nanofeatures in the introduction. Only in the section “Ultrafast Bubble Capture” do you mention the coating with hydrophobic nanoparticles. Please make it clear very early that you are utilizing a hierarchical surface, comprising both nano and microstructuration. This is particularly important as you later highlight the critical role of the nanoparticles in fast capturing. Please go over the manuscript to ensure that this is clear from a very early point and no matter where the reader starts from in your manuscript. It will make the manuscript clearer.

Response: We greatly appreciate the constructive comments provided by the reviewer. We have clarified early in the revised manuscript that our microstructured surface is coated with nanoparticles. However, we have intentionally avoided the term “hierarchical” to prevent any potential confusion with more complex, multi-scale structures. Instead, we emphasize the key role of microstructures in accelerating liquid film drainage for ultrafast bubble capture. Following the reviewer's suggestion, we have indicated the role of nanostructures by using the term “super-aerophilic”, and we have ensured clarity on the surface features that are described clearly throughout the manuscript, regardless of where the reader begins.

Futhermore, we have conducted more control experiments and discussed about the isolated role

of micro- and nano- structures (please see Fig. R4): (1) undecorated microstructured surface, (2) silanized microstructured surface (exhibiting super-aerophilicity but not stable). These new experiments are used to compared with our original results, including nanostructured surfaces and nanoparticles coated microstructured surfaces.

Fig. R4a and b demonstrates bubbles rebound on both nanostructured surfaces and undecorated microstructured surfaces, suggest that individual nanostructures or microstructures cannot facilitate bubble capture upon the first contact with the surface. In contrast, bubble can be captured instantly upon first contact with both silanized microstructured surfaces ($t_c = 2.1 \text{ ms} \pm 0.6 \text{ ms}$, Fig. R4c, i) and nanoparticle coated microstructured surfaces ($t_c = 1.5 \text{ ms} \pm 0.1 \text{ ms}$, Fig. R4d), as both surfaces can form a gas layer (plastron), either thick or thin. We think this strongly support our claims that microstructure as well as stable plastron are two necessary conditions for rapid bubble capture. However, there are differences between the two surfaces capable of capturing bubbles. Since the plastron formed on the silanized surface is metastable and tends to be easily lost under hydrostatic pressure when immersed in water, it leaves only a thin gas layer sufficient for bubble coalescence but insufficient for complete bubble absorption ($t_c = 2.6 \text{ ms} \pm 0.3 \text{ ms}$, Fig. R4c, ii). On the other hand, bubbles could be fully absorbed on the nanoparticle decorated microstructured surface. Nevertheless, this difference in bubble absorption stage doesn't affect our claims on the microstructures' contribution during initial drainage stage of the capture process. As for the nanostructures, in terms of the core issue of bubble capture or non-capture that we are concerned with, they primarily contribute to stabilizing the gas layer, rather than being the only way of achieving super-aerophilicity thus forming plastron.

2. In large parts of your manuscript, you are exploring $D_0/L = 1.6$ to 3.5 . Please argue why this is the relevant range because it may otherwise appear to simply be a range where your choice of structure length scale seems to work. Likewise, please argue even more explicitly why your capture mechanism wouldn't work if everything were scaled down by a factor of 10-100. What do I mean by this? Your main claim in the paper is that microstructuration is necessary, but I am not convinced that this need is not just caused by the use of 1.6–3.5 mm gas bubbles, and you indeed find that it doesn't work for bubbles that are too small or too large. But would it work if you scaled the structures down proportionally to the bubble sizes? A related note: Do you have any thoughts on how to broaden the range of bubble diameters that experience ultrafast capture?

Response: We appreciate the reviewer's valuable comments and would like to provide a clarification regarding the bubble size range and its implications.

The reviewer may have concerns about the range of bubble sizes for ultrafast capture is relatively narrow in our study. It shall be noted that the range $D_0 = 1.6 \text{ mm}$ to 3.5 mm (D_0/L is within the same range as D_0 , given that $L=1 \text{ mm}$ in our study) represents a significant and broad

range, covering the bubble sizes typically used in current related research, as summarized in Table R3. Furthermore, in a natural water environment, the relatively stable bubble size typically ranges from a few millimeters to several centimeters, while micro-sized bubbles are prone to dissolution, and large bubbles are more likely to rupture.

Table R3. Summary of the range of bubble sizes

	Article	Bubble size (mm)
1	Advanced Materials Interfaces, 2020, 7, 1901599	2.0~2.6
2	Advanced Materials, 2021, 33, 2101855	1.8
3	Nature Communications, 2021, 212, 5358	2.0
4	Soft Matter, 2019, 15, 5819-5826	2.6
5	Physics of Fluids, 2016, 28, 057103	1.5
6	International Journal of Mineral Processing, 2007, 81, 205–216	1.5
7	Physical Chemistry Chemical Physics, 2013, 15, 2586-2595	1.5
8	Langmuir 2009, 25(24), 14129–14134	3.0

In accordance with the reviewer’s suggestion, we have extended the range of bubble sizes in our supplementary experiments. As shown in Fig. R6, when the bubble size grows to $D_0/L=5.1$, we find that the bubble is also captured fast when first contact with the MA surface, although the capture time increases from an average of 1.5 ms to 11 ms. This increase is primarily due to the increased drainage pressure from larger bubbles, which decelerates the bubble velocity, thereby prolonging the film thinning process. Conversely, when the bubble size D_0/L is less than 1, the capture process becomes quite random. This is because the bubbles are more likely to directly contact the gas film between the two microcones rather than the three-phase contact line on the microcones where the bubbles are captured. In such cases, the behavior resembles the interaction between bubbles and the gas layer on a hydrophobic flat plate, where the bubbles tend to bounce off rather than being effectively captured. Furthermore, the range of bubble size in real environment are indeed limited, and the corresponding D_0/L in our study almost covers a relatively wide range.

Fig. R6. Capture time with varying sizes of bubble from $D_0/L = 0.6$ to 5.1 on the MA surface.

When the overall size of the structure and bubbles is reduced by a factor of 10 to 100 times, it brings about a series of notable effects. Firstly, the corresponding decrease in bubble volume significantly reduces buoyancy, thereby slowing bubble movement and greatly diminishing the surface's ability to enhance water drainage. Secondly, the volume of gas trapped within the microstructure also decreases significantly, resulting in a nearly negligible gas layer. Such a tiny gas layer struggles to produce sufficient deformation at the interface, thus lowering the efficiency of bubble capture. Therefore, although the surface may still capture bubbles, due to the combined effects mentioned above, the capture time is likely to increase significantly, as shown in Fig. R7. This finding does not undermine the validity of the proposed ultrafast bubble capture mechanism, which relies on microstructure-enhanced drainage. However, it suggests a relatively narrow parameter space for optimizing ultrafast bubble capture at this scale.

Fig. R7. A sequence of images of bubbles with a diameter of 0.4 mm captured on an MA surface with cone bottom radius $a = 0.025$ mm, height $b = 0.06$ mm, centers spaced $L = 0.07$ mm apart.

In summary, the bubble size range of 1.6-3.5 mm adopted in our study is reasonable, and our mechanism demonstrates high efficiency and stability in bubble capture within this range.

Furthermore, the bubble size distribution in a particular application should be considered when determining the range of bubble sizes that can be effectively captured. We feel it's unrealistic to design a structure that can capture all bubble sizes. For a targeted reasonable range of bubble sizes, our theoretical model of critical pressure (Eq.1) can be used to determine the optimal range of structural parameters that facilitate rapid capture.

3. Page 7: "...shows that the gas bridge tends to form at the specific positions where the bubble meets TPL." If pinning is critical, please add data or references to NP-free control samples, showing how triple-line pinning is indeed critical. You present a convincing theory, but this control experiment is missing. Removing the nanoparticles would also alter your advancing contact angle, thereby providing a second data point for your critical pressure theory (which has advancing contact angle as a parameter). This is critical for supporting one of your main claims. One way to address this is to explore $L \ll 1$ mm to bridge the micro to nano regime. The question is whether the absolute size of the structure is important or if it is just the ratio D_0/L that matters.

Response: We appreciate the reviewer's insightful comments on the role of pinning three-phase contact lines. Regarding the control experiments, we additionally prepared a silanized microstructured surface (Fig. R4), thereby eliminating the influence of nanoparticles but ensuring super-aerophilicity. Although this metastable Cassie-state is prone to collapsing into the Wenzel-state when submerged in water, preventing full spreading of bubbles, the bubbles still adhere rapidly upon first contact. This observation is in contrasting sharply with the bouncing phenomena observed on the bare microstructured surface (without any hydrophobic treatment), as illustrated in Fig. R4b. It is reasonable to suggest that the silanized microtextured surface maintains a thin gas layer underwater, yet not as pronounced, thus facilitating bubble adherence, particularly at the points where the three-phase contact line is pinned. In brief, these control experiments enhance our claim regarding the crucial role of microstructures in pinning the three-phase contact line during the drainage stage of bubble capture, thus facilitating ultrafast bubble capture.

Regarding the suggestion to modify the advancing contact angle by removing nanoparticles, and providing a second data point in Fig. 3h, it appears that our description of Eq. (1) might not have been as clear as intended, which may have led to a slight misunderstanding. Eq. (1) requires the presence of an observable thick gas layer on the added control sample. This is a prerequisite to Eq. (1). In this scenario, the absence of a thick gas layer means that we are unable to model the mechanical equilibrium between the interfacial tension force and the liquid film pressure, as such we cannot derive the critical pressure threshold from Eq. (1), and cannot provide a second data point in Fig. 3h.

Lastly, we would like to emphasize that both the absolute size of the structure and the ratio D_0/L are of importance for ultrafast bubble capture. When the absolute size of the structure is

too small (e.g., $L \ll 1$ mm), the surface lacks sufficient volume to encapsulate a thick gas layer, closely resembling a macroscopic flat super-aerophilic surface modified solely by nanostructures. In this scenario, the enhanced drainage effect of the surface structure is greatly diminished. Regarding the ratio D_0/L , a very large D_0/L tends to make the bubble more prone to contacting the macroscopic flat super-aerophilic surface, while a very small D_0/L results in significant variations in the local areas where the bubble comes into contact, leading to unstable bubble capture behavior as achieved by many current studies in the literature. These conclusions do not undermine the mechanism we proposed for ultrafast bubble capture.

4. For the bubble collection in liquid flow, it is unclear what role the transient phase plays. I imagine it takes some time to set up the gas-flow channel (is that the 5 seconds you mention in the extra material)? Please include details on the transient versus steady-state behavior.

Response: We understand that the mention of "5 seconds" may have caused some confusion. In the bubble capture experiment conducted in a flow environment, the process from the initial release of bubbles to the formation of a stable gas flow channel (where bubbles are continuously captured and detached) involves only a very brief transient phase, as shown in Fig. R8. Immediately following bubble release, the surface that is initially fully wetted spontaneously forms an initial gas layer. This allows bubbles to detach continuously from the surface terminus shortly after and then be collected by a cylinder. On the MA surface, this gas transport process is highly efficient, resulting in a transient phase lasting less than 1 second, which has a negligible impact on the overall process. After this phase, the system rapidly stabilizes, with bubbles being captured and detached at a steady rate, enabling continuous bubble collection. The reference to "5 seconds" in the original text was intended to indicate the point at which images were sampled, rather than to directly describe the exact time required for establishing the gas flow channel. To clarify this, we have included additional descriptions of the experimental process in the revised manuscript.

"This gas transport process exhibits high sensitivity, resulting in a transient phase (less than 1 second) that has negligible impact on the statistics of gas collection."

Fig. R8. The optical images depict the end of the MA surface in both its initial and stable states. The images at the top are captured from a side view, while the images at the bottom are from an upward view. Prior to stabilization, an initial gas layer is already present, and upon stabilization, bubbles tend to detach.

5. Is there a possibility to add a prediction for the bubble trapping as well?

Response: We thank reviewer for the suggestion. We understand that the reviewer may have anticipated an accurate prediction of bubble trapping in a fluid environment, similar to the comprehensive analysis we conducted in the static environment. Theoretically, this would involve repeating the analysis of the same physical quantities and establishing their quantitative relationships, including gas layer deformation, bubble dynamics, and liquid film drainage. Such an analysis offers not much innovation in methodology. On a practical and technical level, however, this presents significant challenges. The gas-liquid interface of the air layer undergoes dynamic deformation in response to the flowing liquid (Journal of Fluid Mechanics, 2018, 835, 45-85; Physical Review Letters, 2008, 100, 246001), and bubble movements fluctuate with the water flow, making their precise motions, especially near the surface, difficult to accurately describe. Such level of predictions falls beyond the scope of this study but remain an area of potential interest for future study.

Furthermore, if the goal of predicting bubble trapping in flow environment is to evaluate the performance of the current MA surface, we have already successfully demonstrated rapid bubble capture under flowing conditions.

Minor comments

1. Page 5: Please add errors to 1.5 ms, 71.5 ms, and 1.54 ms (Is this the same as 1.5 ms? If yes, why don't you use the same rounding?).

Response: We have standardized the rounding to maintain consistency in the revised manuscript. For the data of 1.5 ms and 71.5 ms in “Ultrafast bubble capture” section, error values were not included because we were simply describing the results from Fig. 1d, which represents typical model tests. However, for each condition in Fig. 1e, we have provided the numbers and errors, and provided an additional 100-repetitive bubble capture experiment on the MA surface, as shown in Fig. R3, to enhance the statistic significance of our results.

2. Page 5 (and caption of Fig. 1F): For S_c , is this a characteristic size or a roughness? It appears that you refer to it as both, which can be confusing for people in other fields, as roughness is also defined as the actual surface area normalized to the projected surface area. Please clarify and also elaborate on the benefits of normalizing with respect to S_0 .

Response: Indeed, S_c encompasses both characteristic sizes and roughness, stemming from the data diversity in previous researches. Some studies specify a precise characteristic size (Advanced Materials Interfaces, 2020, 7, 1901599; Advanced Materials, 2021, 33, 2101855; Langmuir 2009, 25(24), 14129–14134), while others provide only roughness parameters, particularly those involving solely nanostructured surfaces (Physical Review Letters, 2016, 117, 094501; Physics of Fluids, 2016, 28, 057103). To address this, we have made efforts to include all relevant information and have chosen to use the maximum roughness characteristic structural size for comparison.

In addition, regarding the normalization for S_{min} , we believe it enables clear visualization of the scale differences among different surface structures (up to 10^4 times). In accordance with the reviewer’s suggestion, we have provided a more detailed explanation of this parameter in the revised manuscript (“ S_c is the maximum characteristic structural size on each surface and $S_{min} = 0.1 \mu\text{m}$ is the minimum characteristic size among the all”).

3. For bubble capture, there is the question of how fast the bubble merges with the plastron, but there is also the subsequent air transport to prepare the surface for the next bubble, which is important for continuous bubble capture. You define capture time as “the interval from initial contact to forming a spreadable three-phase contact line,” but given the transport argument above, another logical choice would be the interval from initial contact until the system is ready to accept the next bubble. Please provide an estimate for this second definition and elaborate on how the transport/build-up may be a bottleneck for continuous bubble capture.

Response: In our work, for the second definition of capture time mentioned by reviewers, that is, a bubble with a radius of 2 mm on the MA surface takes 7.2 ms from initial contact to full spread, whereas this process takes 77.6 ms on the FH surface. It can be observed that the time difference between the second and the first definitions is approximately 6 ms for both surfaces. This indicates that the discrepancy during the spreading stage is insignificant compared to that during the bouncing process, which can be differ by an order of magnitude. Furthermore, our

primary focus is on the drainage phase of the liquid film, as it is the crucial stage where the microstructure plays a key role. Therefore, we believe the current definition and comparison of capture time align best with the focus of this study.

Based on the applied definition of bubble capture time in our work, we re-extract the capture times from previous papers using images or videos, and provide a unified capture time in the following Table R2 and the updated Fig. 1f in the revised manuscript.

Regarding the bottleneck for continuous bubble capture, the gas storage capacity between microstructures on any super-aerophilic surface is finite. Inevitably, this leads to bubble accumulation, forming a surface bulge that can cause new bubbles to bounce off. Nevertheless, this isn't a concern for the intended drag reduction application, where the focus is to maintain a gas layer of a certain thickness by rapidly capturing bubbles. Once this layer is saturated, whether or not new bubbles are captured isn't as critical. We think that in other applications, such as bubble elimination, it would be far more important to consider overcoming the bottleneck for continuous bubble capture. Furthermore, as evidenced in our flow environment capture experiments, we have demonstrated rapid bubble spreading and transportation on our surface, facilitating swift recovery of the gas-liquid interface to the original thickness, ensuring a favorable gas-liquid interface state for continuous and efficient bubble entrapment.

Table R2. Summary of bubble capture time in previous work

Classification	Article	Materials	Structure Characteristic Size* (mm)	Bubble Diameter (mm)	Unified Capture Time (t_c , ms)
1 Porous membrane	Soft Matter, 2019, 15, 5819-5826	Hydrophilic porous membrane + hydrophobic nanoparticle coating	0.005	2.6	4.0
2 Nano-coated plate	Physical Review Letters, 2016, 117, 094501	Glass substrate + hydrophobic nanoparticle coating	0.0001	2.0	3.1 - 146.2
3 PTFE plate	Physics of Fluids, 2016, 28, 057103	PTFE substrate	0.0050	1.5	115.0 ± 17.0

4	Rough Teflon surface	International Journal of Mineral Processing, 2007, 81, 205–216	Sandpapered commercial Teflon substrate	0.0500	1.5	2
		Physical Chemistry Chemical Physics, 2013, 15, 2586-2595	Sandpapered commercial Teflon substrate	0.0050		105.0 ± 4.0
				0.0600	1.5	37.0 ± 3.6
				0.1000		2.0 ± 0.5
5	Micro-nano surface		Micropillared Silicon substrate + silane modification	0.0050	2.1	41.0
				0.0100	2.2	145.0
		Advanced Materials Interfaces, 2020, 7, 1901599		0.0500	2.0	30.0
			Microconical Silicon substrate + nano-coating	0.2040	2.6	5.7
				0.0440		2.0 – 5.0
		Advanced Materials, 2021, 33, 2101855	Glass substrate + nanoparticle deposition	0.0305	1.8	2.0 – 22.8
				0.0163		2.0 – 41.3
				0.0126		2.0 – 42.5
	Nature Communications, 2021, 12, 5358	Glass substrate + powder coating	0.0500	2.0	1.0 – 10.0	
	Langmuir 2009, 25(24), 14129–14134	Silicon substrate + etched nano texture + chemical modification	0.0150	3.0	4.4	
6	Nanofiber	ACS Omega	PMMA	0.0006	4.0	203.0

* Structure Characteristic Size is the maximum characteristic structural size on each surface.

4. Your introduction goes over the development in the field of bubble capture, and you also repeatedly emphasize that you are pushing the limit for capture time. However, as you mention in the introduction, previous work in the field includes micro-nano structuration, so it is unclear why previous microstructures were not as effective as those in the presented work.

Response: We appreciate the reviewer's critical insights, and give us chance to illustrate the novelties of this work. Previous studies on bubble capture using micro-nano structures demonstrates some utility, yet they fall short in consistency and performance compared to our work. It seems that these studies may not fully address the rapid bubble capture mechanism, leading to a lack of targeted design in surface structures. As a result, most of the reported surfaces are random micro-nano structures, causing significant variations in capture times. Furthermore, the inadequate gas layer interface effects between surface structures result in poor drainage performance. This makes it challenging for the liquid film to drain out quickly, a stage we consider crucial in determining whether bubbles can rupture rapidly. In light of this, we specifically focused on enhancing the drainage process when designing our targeted microstructure. Our findings indicate that the introduction of microstructures significantly enhances liquid film thinning, and we've designed targeted periodic surface structures that yield even better results.

5. Page 7: Is the precision of 129.9% justified? Please provide an error estimate.

Response: We appreciate the reviewer's suggestion to consider the variability of our data. To provide reliable estimates of the increase in drainage velocity, we detail below our sampling method and the estimation approach based on the sampled data. First, we selected the sampling region at a height of 0.1 mm above the bubble's upper surface, along a horizontal line of length $1.2 * D_0$, which means the distance where the film pressure P_f has reduced to 0 and D_0 is the bubble diameter. The absolute horizontal velocity along this line (with a sample size of 100 discrete points) was used to represent the drainage velocity across the entire liquid film. The drainage velocity was collected at five heights where the bubble ascended to 0.9 mm, 0.7 mm, 0.5 mm, 0.3 mm, and 0.1 mm from the tip of the microcones, respectively. At each height, the averaged drainage velocities were obtained for three operating conditions at $x = L/2$, $x = L/6$, and $x = 0$. Subsequently, the acceleration ratios of $x = L/6$ and $x = 0$ relative to $x = L/2$ were calculated. Finally, the averaged acceleration ratios at the five heights were computed as the overall acceleration ratios, along with their standard errors. In this way, we avoid the limitation of partial evaluations and ensure the accuracy of our overall assessment of the enhancement in liquid film drainage speed.

Following this statistical approach, we estimated the acceleration ratio and its error for the released position at $x = L/6$ are $69.3\% \pm 16.4\%$. At $x = 0$, the acceleration ratio and its error are $90.4\% \pm 18.5\%$. The reported overall acceleration ratio of 129.9% in the origin manuscript was slightly above the upper limit of its confidence interval, which was due to differences in sampling methods. In the revised manuscript, we have adopted data and errors based on the above-mentioned more robust statistical approach. We believe that this method can efficiently enhance the reliability of our data.

6. Page 7: “We highlight a sharp reduction in capture time t_c from 100 ms to 1.5 ms near TPLs.” Here your surface shows a variation in capture times similar to other works. Is this also the cause of variation in other works? This is relevant for understanding why no one before you achieved similarly small capture time variation.

Response: We thank the reviewer for the insightful comments. We believe that the cause of observed variation in our capture time is one of the reasons for similar variation in other works. The corresponding experiments were specifically designed to elucidate the precise location of bubble bursting. Our results have confirmed that the three-phase contact line serves as an active site for bubble bursting, as it rapidly attains the critical rupture thickness during the liquid film thinning. We think that previous studies might have overlooked the effects of the liquid film dynamics during the drainage stage, leading to the absence of specific design that could enhance liquid film drainage. Additionally, there are other reasons contributing to such fluctuations in capture time on the previous aerophilic surfaces, such as the random structures and surface wettability.

7. Page 8: 162° —Please add the errors to your numbers. I’ve seen in your supporting information that you have them all, so please add them to the main article to make it easier for the reader to judge the reliability/trustworthiness of the study.

Response: Following the reviewer’s constructive suggestion, we have added the errors of apparent contact angle and advancing angle in the main text.

8. Page 8: As I understand, your theory (eq. 1) is a 2D model. How does the actual 3D case differ from your model? Please also make it clearer that you are using “parallel triangular ridge protrusions” (i.e., a different geometry than your initial cones) for this part of the study. It was only obvious to me after reading the methods section on page 19.

Response: We appreciate the insightful comments regarding our theoretical framework. Firstly, we would like to clarify that Eq. 1 in our manuscript is indeed based on a 3D model, rather than a 2D one. Specifically, Eq. 1 is derived from a 3D system that considers four microcones as a fundamental unit, as illustrated in Supplementary Fig. 16(a) of SI. We apologize for any confusion that may have arisen from our previous description.

For the “parallel triangular ridge protrusions” used in the model surface depicted in Fig. 2, they were employed specifically in the Section of “Enhanced Water Film Drainage within a Characteristic Unit” to verify our liquid film evolution theory. The choice of parallel ridges over microcones was driven by the need to address the occlusion problem encountered in high-speed camera recordings. By using parallel ridges, we were able to avoid front and back occluded projections that could obscure precise bubble contact points, thereby ensuring clarity and accuracy in our experimental setup and analysis.

9. Page 11: “Still maintains 98.8%”—what Re is this for?

Response: We appreciate the reviewer’s careful review. The Reynolds number in this test is chosen as 2083. We have now explicitly included this information in the revised manuscript, to ensure that it is clear and easily accessible for readers.

10. Conclusion: “The most basic microstructure” — Define how to measure "most basic," or perhaps just call it a “simple conical structure” throughout the manuscript.

Response: Following the reviewer’s suggestion, we’ve mostly used “simple conical structure” throughout the manuscript, though not everywhere. Our notion of “basic” stems from the surface structure design and fabrication approach. Traditional micro-nanostructured surfaces often rely on complex methods like nanoparticle deposition, electrodeposition, femtosecond laser, and liquid flame spray, which involve aggregation from nanoscale particles to form random microstructures with complicated regulation. In contrast, our approach utilizes direct 3D printing to create cone-shaped surfaces at the hundred-micron scale, offering a more straightforward solution for large-scale surface fabrication.

11. Can you perform continuous flow-free capture? What is the impact of MA substrate tilting on bubble capture and transport?

Response: This is an interesting issue and we have accordingly expanded our study to include continuous flow-free capture, where the surface, after capturing an individual bubble, was not removed from the water to restore the original gas layer but instead immediately prepared for capturing the next bubble. Fig. R9 presents the experimental images of 34 consecutive tests on the MA surface. In the absence of external flow, bubbles are continuously absorbed by the surface, ultimately coalescing into a prominent surface bubble that cannot be expelled due to confinement. We also find that the bubble capture time increases with the number of trapped bubbles (the numbers of second column in Fig. R9). This is attributed to the fact that upon adsorption, the gas layer thickens to form a surface bubble, and the process shifts from microstructures capturing the bubble to the merging of two bubbles. Notably, when the absorbed gas volume exceeds the encapsulation capacity within the microstructure, a protruding bubble gradually causes subsequent fresh bubbles to come into contact with this bubble surface instead. At this point, the surface structure ceases to enhance the drainage

process. Ultimately, bubbles fail to make contact with the surface as they bounce and slide away on the surface of the protruding bubble (as observed during the 34th capture attempt). We have incorporated these findings into the revised manuscript, see Supplementary Fig. 4 of SI.

Next, we conducted experiments on bubble capture and transport on a tilted surface. The surface was submerged and inclined at an angle of 20° with continuous bubble capture. We found that the initial bubble capture time is longer than the 1.5 ms observed on a horizontally placed surface, primarily due to the additional distance the bubble had to ascend along the inclined surface. However, the bubble was still rapidly captured upon first contact with the surface. On the other hand, after consecutively capturing 38 bubbles, the surface bubble

Fig. R9. Continuous bubble capture on the MA surface. The experiment was repeated 34 times until the bubble could no longer be captured and slid off the surface.

experiences a buoyancy-driven upward motion, leading to its eventual detachment from the surface. The surface is restored to its initial state, thus the capture time experiences a sudden drop for the 39th bubble, as shown in Fig. R10. The tilted surface facilitates the detachment of the surface bubble, thus restoring the interface morphology between the microstructures and significantly enhancing the continuity of bubble capture.

Fig. R10. Continuous bubble capture on the tilted MA surface. (a) The image sequence of 50 consecutive bubble capture experiments. The MA surface was inclined at an angle of 20°. (b) The capture time as a function of the number of bubbles captured.

12. Fig 1f: Please make it clearer that the small numbers are paper references and that your structure also belongs to the micro-nano group.

Response: In accordance with the suggestion, we have revised the figure to make it explicit that the superscript numbers represent references to other papers. Additionally, we have clearly indicated that our structure also belongs to the micro-nano group, aligning with the categorization used in the field.

13. Fig 3g: Please include $Re > 3750$ and $Re < 2083$, which I believe you have data on but only put in the extra figures. It won't take up more space, and it will give the reader a better idea of the system's limitations. Please also add the injection rate so it is easier to compare with Fig 4h.

Response: We thank the reviewer for the thoughtful comments. In accordance with the suggestion, we have incorporated these results of $Re > 3750$ and $Re < 2083$ (original Extended Data Fig. 19) in Fig. 4g, and added the injection rate accordingly. We would like to clarify that for $Re < 2083$, we observed minimal difference compared to the scenario at $Re = 2083$, as a capture efficiency of 100% is already achieved at this point. When $Re > 3750$, a notable decline in capture efficiency is observed on the MA surface. This is attributed to the loss and degradation of the gas layer on the MA surface due to water flow impact, accompanied by severe fluctuations in the gas layer structure. We hope that these findings provide readers with

a comprehensive understanding of the limitations associated with this surface.

14. Fig 4f: "...at varying locations along the surface after a 5-second bubble capture process." What does this mean? Is it the time it takes to establish a steady state? And starting from a completely wet surface or starting from a plastron? Or?

Response: We would like to provide additional clarification on the procedure and rationale behind the time mentioned in our manuscript. We start from a surface with a plastron, as the aerophilic property of the surface causes a gas layer to form spontaneously when the flow fully wets the channel. While it does take some time for continuously captured bubbles to accumulate and fill the gas layer, this process can be quite rapid. From our observations, bubbles begin leaving the surface and entering the collection cylinder within approximately 1 second after gas injection.

Here, the "5-second" duration refers to the time we allowed before capturing images, ensuring that we observed either a fully stabilized gas flow or uncaptured bubbles under steady-state conditions, as shown in Fig. 4e and 4f. Yet, this does not imply that it takes 5 seconds to establish the gas layer. Instead, this duration was chosen to ensure consistency in our experiments.

15. Fig 4c-d: I saw the two figures are not aligned with each other, and I wonder if this is the case for more figures. Improve it if you feel perfectionistic.

Response: We thank the reviewer for pointing out our oversight. We have addressed the alignment issue in Fig. 4c-d and have thoroughly checked all figures in the manuscript for similar layout issues.

16. Page 20: Please explain how you provide gas bubbles for 12 hours straight. It says: "a submerged syringe pump." I'm not sure what this is, but it suddenly struck me that it was not clear to me how you provide air for 12 hours straight. If you use a syringe, it has to hold $3 \text{ mL/min} \cdot (60 \cdot 12) \text{ min} \sim 2 \text{ L}$, which is a syringe bigger than I have ever seen. I expect that I have misunderstood, but more readers will misunderstand, so I would suggest that you make this clearer. It should be easy.

Response: We use a 30-mL syringe with a syringe pump to maintain a consistent output rate of 1 mL/min. Every 30 minutes, the syringe is briefly disconnected (≤ 10 seconds) for air replenishment, ensuring minimal disturbance to the surface air layer and continuity in the 12-hour bubble-trapping experiments. Following the reviewer's suggestion, we have incorporated a description in the "Bubble capture and transportation in water flow" section of the Methods.

17. Page 20: "The flow rate could be adjusted from 0 to 1.6 L/min to achieve a Re number range of 0 to 3333." However, you investigated higher Re numbers, so how did you do this?

Response: We thank the reviewer for pointing out this error. We have corrected it to 2.6 L/min

in the revised manuscript. The specific operating conditions employed are also outlined in Supplementary Fig. 21(b) of SI, for clarity and reference.

18. Ex Data Fig 2b: What is the unit on the x-axis? Does this range actually match the A-A in (a)?

Response: The unit on the x-axis in Extended Data Fig. S2(b) (revised Supplementary Fig. 2(b)) is μm , as indicated in the figure. Next, the test range does not precisely align with A-A in (a), as it was intended to provide a quick overview of the data in Supplementary Fig. 2(b) for readers. To avoid ambiguity, we have revised this figure by removing the A-A line, and clarified that the roughness measurements are for the surface depicted in Supplementary Fig. 2.

19. Ex Data Fig 2c: “10 μL hangs on the FH surface.” In the photo, it looks very much like a sessile droplet and not a pendant/hanging droplet. Please clarify.

Response: We appreciate this careful review and have revised "hangs on" to "sessile droplet" in Ex Data Fig 2c, which is now Supplementary Fig. 2.

20. Ex Data Fig 3: “ t_0 ms” in the figure. Is the “ms” unit supposed to be there in the 2nd column?

Response: Following the reviewer’s comment, we have removed the “ms” unit in the 2nd column in Extended Data Fig. 3, which is now Supplementary Fig. 6.

21. Ex Data Fig 8: What is the reference point for this “contact location,” i.e., where is "contact location" = 0?

Response: In Extended Data Fig. 8, the contact position = 0 corresponds to the tip of the first protrusion, and the value of contact position signifies the displacement of the surface from this reference point. To enhance clarity, we have revised the x-axis label to “Displacement” in the Supplementary Fig. 12, thereby facilitating a more intuitive understanding of the statistical data presented.

22. Ex Data Fig 22: “all present nearly straight lines at all three injection rates.” In (a) 0.017 mL/s, the curve does not look straight. Please comment and correct if you made an error.

Response: We have re-conducted the experiment for the condition of 0.017 mL/s in Extended Data Fig 22(a) and corrected the original fluctuation curve. The updated figure, which is now Supplementary Fig. 25(a), shows a nearly straight line at this injection rate.

23. The work reminds me of the work by Prof. Quéré on bouncing droplets, where they also identify various regimes for bouncing and sticking, which depend on bridge formation. It would be interesting to have a short reflection on how looking at techniques for reducing bouncing of water droplets can inspire further developments in the field of gas bubble capture.

a. Mousterde, T., Lecointre, P., Lehoucq, G. et al. Two recipes for repelling hot water. Nat

Response: We also have noticed Prof. Quéré's work on bouncing droplets. Indeed, there are similarities between the bouncing and capture of bubbles and the bouncing and adhesion of droplets, specifically in terms of the role of liquid or gas bridge formation in facilitating adhesion or causing bouncing. This similarity in mechanism allows us to draw insights from the process of liquid bridge formation to better understand the role of gas bridge in bubble capture behavior. Nevertheless, the differences between bubble capture and droplet bouncing are also quite apparent. Given that bubbles are situated in a liquid environment, coupling effects among deformed bubbles, surfaces, and fluids are pronounced during the capture process. This aspect is often not considered in studies on droplet bouncing, as the influence of the atmospheric environment on droplet dynamics is generally neglected. To address this gap, specialized experimental methods and theoretical models are necessary in our study to understand the liquid film dynamics with the microcone-textured super-aerophilic surfaces.

We have cited the paper as Ref. [41] and added the above discussion in the Conclusion of the revised manuscript.

“Interestingly, the bouncing and capture of bubbles share some similarities in liquid/gas bridge formation with the bouncing and adhesion of droplets⁴¹. However, the dynamics of the three-phase contact line and adsorption during bubble capture, influenced by the surrounding liquid, are more complex than droplet adhesion, warranting special attention.”

24. The title could be made more informative, such as: “Ultrafast bubble capture on microcone-textured aerophilic surfaces.” Even better if there is mention of the nanostructuration, which may be hinted at by calling it “super-aerophilic” instead; similar to the field of water wetting, where superhydrophobic implies that some structuration has been applied to the materials.

Response: We greatly appreciate the reviewer's insightful suggestion for the title. In accordance with the reviewer's suggestion, we have revised the title to “*Ultrafast bubble capturing on microcone-textured super-aerophilic surfaces*” to reflect the key aspects of our study effectively. We have also revised the text to ensure clarity and coherence throughout the manuscript.

25. Consider having an extra round of proofreading done, focusing on writing easily digestible sentences.

Response: Following this suggestion, we have thoroughly proofread our manuscript, focusing on ensuring clear and concise presentation while maintaining scientific rigour.

Point-to-point responses to reviewers' comments

We sincerely appreciate all the reviewers for their time and effort in evaluating our manuscript. We especially appreciate Reviewer 1's critical and detailed feedback, which has helped us refine our claims and strengthen key aspects of our study. At the same time, we are also grateful to Reviewers 2, 3, and 4 for their thoughtful assessments and overall support for the publication of our manuscript. In this revision, we have carefully addressed all the concerns, and revised the manuscript accordingly, where appropriate.

Reviewer: 1

Over the course of the review, I believe it is now clear from opinions of the reviewers and the authors that there remains two main points of contention.

First, the manuscript does employ the use of nanostructures, albeit claims in stabilization of the Cassie-state, they should still be explicitly described to avoid misleading readers. This is as of yet (see last section of the review attached) not entirely clear in the primary sections of the manuscript.

Second, the surfaces proposed appear to work primarily with fast moving free bubbles, and not so well with slow moving bubbles (in fact they take an order of magnitude more time than state-of-the-art nanotextures). However, it is also difficult to justify if they are indeed the best with slow bubbles without a direct comparison to the state of the art. Therefore, claims should be made within such limits until more evidence is clearly presented. As far as glaco is concerned, it also has an unclear chemistry makeup, and that makes it a difficult material to benchmark.

With a reorganization of the manuscript around its key points of discovery, and even with glaco surfaces as a control, I would be happy to recommend this work for publication. However, the claims must be accurate and not as universal-sounding as they currently are, which I would still suggest a major revision on. Reduction of the universal-sounding claims is helpful to prevent premature limitation of the field and avoid confusing the benchmark set in current literature.

Response:

We sincerely appreciate the reviewer's thorough evaluation of our revised manuscript and the constructive feedback. We have thoroughly addressed the points raised and fully comprehended the reviewer's intentions, especially the two main aspects: 1) providing an explicit description of nanostructures everywhere; 2) ensuring our claims are limited to free-rising bubbles while discussing performance variations at both fast and slow bubble velocities.

We have undertaken a major revision, explicitly stating "nano/micro-structured super-aerophilic surfaces", focusing more specifically on discussing free-rising bubbles, refining claims to avoid overgeneralization, reducing the universal-sounding claims, such as "surpassing the most capture times" in the Abstract and ensuring that all statements are appropriately scoped.

I have looked through the descriptions of this work again, and found that, while the new claims are more muted, the impact is also slightly lessened since it is now accepted (by both reviewers and authors) that the effect observed is only dominated by fast moving bubbles and cannot apply to slow moving variants (Figure R2b illustrates this). Moreover, as many reviewers have pointed out, the lack of mentioning nanostructuring does appear to introduce a lot of confusion among us. At this point, the authors will also likely agree that this is the case. The authors have made the effort to change the title slightly, but it still feels insufficient since superaerophilic surfaces can be made “simply” using microstructures too! More significantly, when bubbles are slowly moving, the surfaces that the authors have shown (MA and FH) are significantly less efficient in rupturing bubbles compared to the state-of-the-art (Table R1). Therefore, while this work captured my original attention with regards to their “highly performing ultra-fast” bubble capture terminologies, I now feel that the novelty falls under “Understanding bubble capture with fast moving rising bubbles using micro-cones” and not in absolute performance. This also falls in line with my original suspicions. Naturally, it is difficult to expect microstructures to “outperform” nanostructures, and therefore I believe it is important to state “when” they do outperform and otherwise. Having that level of clarity is essential in efforts to make comparisons with the state of the art. I do still believe that by shifting focus away from nano- vs. micro- to fast vs- slow bubbles, this manuscript still warrants publication in Nature Communications. However, this would require re-drafting around the new results found – and framing the logic around how “most” bubbles are often freely flowing and moving at high speeds.

Response:

We thank the reviewer for these concrete and detailed suggestions for revision. We fully understand the concerns regarding the scope of applicability and clarity of expression in our study. We have carefully revised the manuscript to more precisely highlight the innovative aspects and applicability of our work.

Key revisions include: focusing on the capture mechanism of free-rising bubbles, clarifying the role of nanostructures, discussing the performance of surfaces with different bubble approach velocities in the Results, and revising or refining expressions in the Title, Abstract, Introduction and Conclusion.

Detailed comments are listed below:

1. *Response 1: “After careful consideration, the title is revised to “Ultrafast bubble capturing on microcone-textured super-aerophilic surfaces” by removing “simple.” This change aims to specify microcones used as textures, and indicate that “super-aerophilic” is intended as a juxtaposed condition, rather than a subordinate one, to “microcone-textured” for achieving ultrafast bubble capturing. We have also ensured that this revision implicitly acknowledges the contribution of nanoparticles, while highlighting the central role of the microstructures.”*

Q: I am not sure if this revision would acknowledge the contribution of nanoparticles, as the micro-cone texturing description says nothing of the presence of nanostructures. In fact, having superaerophilicity as a description can still apply to pure micro-textured surfaces. For clarity

and to avoid misleading intents, terminologies such as “nano” or “hierarchical” or something more explicit should be included for completeness.

and

“Current super-aerophilic surface designs often emphasize complex nanostructures, while the fundamental role of microtextures in first-contact bubble capture has not been well understood.”

Q: This revision does not illustrate how bubbles are freely moving under much higher velocities than what was investigated in the past (all prior literature discussed). That has to be a point of comparison as differences between slow moving bubbles (See Figure R4) shows great differences in performance between nano-texturing and micro-texturing.

Response:

We thank the reviewer for the critical and insightful comments.

Following the reviewer’s suggestion, we revise the title to explicitly include nanostructure for completeness as “Understanding ultrafast free-rising bubble capturing on nano/micro-structured super-aerophilic surfaces”.

Regarding the second question, our bubble velocity range nearly covers what was investigated in the current literature in the manuscript. Ours: 0.16 to 0.31 m/s in the original manuscript, 0.16 to 0.37 m/s in this version (updated Fig. R1) vs. Current literature (Table R1): 0.14 to 0.37 m/s (with only one exception case down to 0.04 m/s). More importantly, our study and all the compared cases in Table R1 are free-rising bubbles. As we have explicitly stated “free-rising” bubbles in the title and throughout the context to limit discussion under this scenario, we feel it would be better to keep the current narrative. Specifically, compared to current studies on free-rising bubbles, which have extensively examined the role of nanostructures, the influence of microstructures has been relatively underexplored, which represents a critical research gap.

Nevertheless, we fully respect the reviewer’s feedback and adjust the referred sentence as *“While the role of nanostructures during the collision of free-rising bubbles with super-aerophilic surfaces is well established, the fundamental contribution of microtextures in promoting initial capture, even before contact, has yet to be fully understood.”*

We also find it necessary to include physical characteristics of the bubble rising process in the Introduction, thereby setting the stage for the subsequent discussion on microstructures' role in this rising stage, where nanostructures do not have an impact in this context.

2. Response 2: *“After reviewing the Ref. [27] (Adv. Sci., 2024, 2403366), we would like to point out that the claimed similarity between perfluoroalkylated silane functionalized micropillars (not nanostructure-coated) and hierarchical nanostructures occurs only during the final stage---contact line spreading, rather than entire bubble capture process.”*

Q: This is not true. Figure 1h shows how having nanostructures significantly diminishes the “as-measured” rupture time: See the break in the x-axis. Considering the nature of the more “delicate” experiments (needle-attached bubble), the differences are evident.

Response:

We initially did not understand well the statement in comment 3 of the previous review “Recent studies have demonstrated that nanostructure-coated micropillars behave similarly to bare nanostructures, which suggests that the controlling factor in the rapid capture may be the nanostructures rather than the microstructures alone (Adv. Sci. 2024, 2403366).” As a result, our previous response focused on the interpretation of “similarly”.

We appreciate the reviewer specifying that this comment is about Fig. 1h. We agree that, for the needle-attached bubble, having nanostructures significantly diminishes the “as-measured” rupture time.

3. Response 3: “*We report the rising bubble-induced large deformation of the entrapped gas layer, rapidly thinning the liquid film to its rupture threshold and thus achieving an ultrafast bubble capture in down to about 1ms with a simple array of microcones, decorated with nanoparticles as a convenient example to obtain super-aerophilicity, surpassing the most capture times.*”

As the authors accept, and also shown in Figure R4, a fast moving freely rising bubble cannot be compared to many examples shown before – either much slower moving rising bubbles or needle-attached bubbles. As a result, can they truly claim that the capture time by microstructured cones is really the “fastest”? In Figure R4ci and cii, significant differences are observed since the Cassie state in a more “immersed” microstructured surface is already weaker. I also did not see an explicit comparison between bubble velocity and its influence on rupture time (original Query 8). There should be some “link” between this work and prior work using lower bubble approach velocities before a reasonable comparison is made?

On a related note, I really appreciate the tabulation of R1, despite not in graphical form, which has helped me appreciate the findings of this work. That table would certainly do well as supporting information. Adding the estimated “approach velocity” will help to distinguish the authors’ primary finding even more!

Response:

The reviewer's questions helped us reassess the rigor of our arguments. After collecting and reviewing the bubble velocities reported in the current literature in Table R1, we find that our capture times in the original manuscript can be compared with those in the literature. It is important to highlight that both our study and all the cases in Table R1 involve free-rising bubbles. Specifically, the terminal velocity typically ranges from 0.14 to 0.37 m/s, while our similar range for free-rising bubbles was originally 0.16 to 0.31 m/s and now extends from 0.16 to 0.37 m/s by adding new experiments (update Fig. R1). We believe that all of these bubbles align with the "fast moving" bubbles referred to by the reviewer.

Additionally, there is an exceptional case of low velocity ~ 0.04 m/s presented in Table R1 (Nature Communications, 2021, 12, 5358). Out of both rigor and curiosity, we took on the challenge of conducting an extra comparison experiment, lowering the free-rise bubble velocity to 0.04 m/s. We found that, at this velocity, the capture time on the MA surface is still around 1.67 ms (Fig. R1b), while the literature only provides a range of capture times (1-10 m/s) without giving a direct correspondence to 0.04 m/s. Based on all these comparison results, we find it would be appropriate to say that our results for "rising bubbles" "outperform most of the capture times". Yet we have not stated this in the manuscript to avoid making an overclaim.

In accordance with the reviewer's suggestions, we update Fig. R1 (was Fig. R2 in the first response) by explicitly presenting bubble velocity and adding new results data. We would like to point out that since Fig. R1c is needle-attached bubbles, it could not be included in comparing with free-rising bubbles in the prior literature in Table R1 (added as Supplementary Table 1).

Supplementary Note: Regarding the ultra-low velocity of 0.04 m/s in NC, 2021, 12, 535, this case is somewhat special because, at such low speeds, a free-rising bubble is still in the brief acceleration phase moving from 0 and hasn't yet reached its steady velocity, where the buoyancy and drag forces acting on it balance out. More rigorously speaking, when studying the capture of a free-rising bubble, it would be more reasonable to wait until it reaches steady velocity, as before it can be influenced by undesired factors such as gas injection speed and the adhesion of the needle tip. This is why the bubble velocities in other literature listed in Table R1 happen to fall within a relatively higher range. The point is not on the absolute velocity, but on reaching a steady or near steady state, which would be a common understanding in this field. By the way, given that the focus of this study (NC, 2021, 12, 535) is not on bubble capture but rather on anti-foaming, it makes sense that they expand the velocity range in this way. Back to the bubble capture issue, it would be better to exclude this extremely low-velocity scenario to avoid interference with bubble motion caused by hardly-controlled experimental artifacts.

Table R1. Summary of free-rising bubble capture time in pure water

(Compared to the last revision, we have reclassified the surfaces into three major categories, added release distance and bubble velocity, and corrected some values)

Classification	Refs.	Materials	Structure Characteristic Size* (mm)	Bubble Diameter (mm)	Bubble Releasing Distance (mm)	Bubble Approaching Velocity (m/s)	Unified Capture Time (t_c , ms)
Micro-nano surfaces	Advanced Materials, 2021, 33, 2101855	Rough porous microstructures + deposited nanoparticles	0.0440	1.8	2.5	0.14	2.0 – 5.0
			0.0305				2.0 – 22.8
			0.0163				2.0 – 41.3
			0.0126				2.0 – 42.5
	Nature Communicat ions, 2021, 12, 5358	Micro- agglomerates + sprayed superamphiph obic silica nanoparticles	0.0500	~1.9	0.28–73.7	~0.04 – 0.25	1.0 – 10.0
Langmuir 2009, 25(24), 14129– 14134	Etched micro- nano textured silicon substrate + chemical modification	0.0150	~2.6	3.0	~0.16 [#]	4.4	
Soft Matter, 2019, 15, 5819-5826	Microporous membrane + hydrophobic nanoparticle coating	0.005	2.6	2.4	0.20	4.0	
Microstructure d surfaces (Inherently hydrophobic or with chemical modifications)	Physics of Fluids, 2016, 28, 057103	Rough PTFE substrate	0.0050	1.5	> 4.8	0.33	115.0 ± 17.0
	International Journal of Mineral Processing, 2007, 81, 205–216	Rough Teflon substrate	0.0500	1.5	300	0.37	3.0 (revised according to the abstract)
			0.0010				84.0

							(added for completeness)
	Physical Chemistry		0.0050				105.0 ± 4.0
	Chemical Physics, 2013, 15, 2586-2595	Rough Teflon substrate	0.0600	1.5	250	0.35	37.0 ± 3.6
			0.1000				2.0 ± 0.5
			0.0100	~2.2			145.0
	Advanced Materials Interfaces, 2020, 7, 1901599	Micropillars/microprotrusions + submicro features + silane modification	0.0500	~2.0	25	0.35	50.0 (changed from 30 to 50 according to Fig. 3g)
			0.1680	~2.0			5.7
Nanostructured surfaces	ACS Omega, 2022, 7, 39959-39969	PMMA nanofiber mat	0.0006	~2.3	3.6	0.16	203
	Physical Review Letters, 2016, 117, 094501	Silicon substrate + nanoparticle coating	0.0001	2.0	3.0-10.0	0.2 - 0.3	3.1 - 146.2

* Structure Characteristic Size S_c is the maximum characteristic structural size on each surface.

~ denotes extracted data from plots or videos of the literature, as these data are not directly provided.

The approaching velocity is estimated from bubble size and releasing distance.

Fig. R1. **a** Capture time of free-rising bubbles on MA and FH surface at varying releasing distances (i.e. 60mm, 20 mm, 15 mm, 10 mm, and 5 mm), resulting in approach velocities of 0.37, 0.31, 0.30, 0.25, 0.16 m/s. While immediate bubble capture occurs on MA surface with average capture times in a range of 1.46 to 1.75 ms, bubble bounce is observed on FH surfaces in these cases. **b** Capture time of free-rising bubbles at a specific low approach velocity of 0.04 m/s with releasing distance of ~ 1.74 mm. Immediate bubble capture is observed on MA surface with average capture time of 1.67 ms and small standard deviation. No bubble bounce occurs on FH surfaces with an average capture time of 4.9 ms and large standard deviation. **c** Capture time of needle-fixed bubbles on MA and FH surfaces under approach velocity < 0.001 m/s (even an order of magnitude smaller than Fig. 1h in Adv. Sci. 2024, 2403366). No bubble bouncing occurs on FH surfaces. This needle-fixed case can not be compared with free-rising cases in **a** and **b**. The data box displays the mean value with standard deviation, and the black vertical line reaches the maximum and minimum values.

4. Response 7: *“We would like to take this opportunity to give a brief clarification. It was not our intention to omit the discussion on nanoparticles; rather, our use of the term “aerophilic” in the original title was meant to emphasize the necessity of forming a plastron on the microstructured surface for instant bubble capture and subsequent absorption. However, the formation of this plastron does not necessarily require nanoparticles.”*

Yes, indeed this is very important to clarify. I hope that the authors will make the effort to classify the surface appropriately, also in the rest of the manuscript, to avoid confusion.

Response:

We have thoroughly revised the manuscript to ensure that all relevant descriptions are clear and precise to avoid any potential confusion.

5. Response 8: *“Results in Fig. R2 indicate that the capture time on the MA surface remains relatively stable at around 1.5 ms, which is still significantly shorter than that on the FH surface under different velocities. These results have been added within Supplementary Fig. 3 of SI.”*

This is good work and strongly illustrates where the speciality of these nanoparticle-decorated micro-cones are useful and more effective – at bubbles that are approaching at rapid speed. There is definite merit in presenting these in the supporting information to guide future research in the area that will need to take these factors into account.

However, I must say now, at this junction, that the rupture time involved in very slow-moving bubbles (Figure R2b) are now significantly slower (MA: 59.6 ms, and FH: 85.4 ms) than the state of the art per Table 1. Several of these referenced papers in Table 1 have a rupture time of < 10 ms (reference ID 1,2,4, 5). Given this comparative experiment, I now hesitate to say I am convinced that the rupture time is significantly improved from the state-of-the-art listed. Logic of the argument:

- 1) Slow moving bubbles – MA: 59.6 ms vs. FH: 85.4 ms vs. state-of-the-art: < 10 ms.
- 2) Fast moving bubbles – MA:1-2 ms vs. FH: 50-60 ms vs. state-of-the-art: Unknown.

As discussed before, the claim perhaps can still be valid if the state-of-the-art approach velocities can be found and used as comparisons – but a more conservative claim will likely be that these MA surfaces work best only with rapidly moving bubbles. Such bubbles should be distinctively described in the title, abstract, introduction, and conclusions. Otherwise, there could be room for confusion. As I described before, these findings are unexpected, and perhaps show that, while the manuscript may not have advanced state-of-the-art performance, but perhaps advances something more important: the understanding of how different bubble velocities impact so-termed bubble absorbing surfaces.

Response:

Specifically in response to *“the rupture time involved in very slow-moving bubbles (Figure R2b) are now significantly slower (MA: 59.6 ms, and FH: 85.4 ms) than the state of the art per Table 1, Several of these referenced papers in Table 1 have a rupture time of < 10 ms (reference ID 1,2,4, 5).”* we feel compelled to clarify that Figure R1c that was original Figure R1b

(needle-fixed bubble with slow velocity) can not be compared with state of the art per Table R1 (free-rising bubble with fast velocity), given that both bubble release method and final velocity range are different. Only our original results plus Figure R1a and R1b, describe the same scenario as the existing literature, making the comparison of capture time meaningful. Furthermore, as we understand, the final approach velocity would be a superficial difference between free-rising and needle-fixed bubbles; it is not a fundamental distinction. A more comprehensive and fundamental comparison is provided as follows:

	Free-rising bubbles Our original work plus Fig. R1a and b	Needle-attached/fixed bubbles Our Fig. R1c
State of the Art:	All the 10 papers in Table R1 	Adv. Sci., 2024, 2403366 Terminal Velocity:	Ours: 0.16-0.37 m/s vs. literature : 0.14-0.33 m/s (with an exceptional slow velocity of 0.04 m/s)	Ours: 0.001 m/s vs. literature : 0.025-0.03m/s
Determinants of velocity:	Fundamental principles of fluid dynamics	Actively controlled external parameters such as gas injection rate
Bubble rising distance:	2.4-300 mm	~ 0 mm
Bubble shape:	Free and near perfect sphere before contact; Hemisphere at the moment of contact;	A balloon-shaped bubble attached to the needle tip before and even at the moment of contact;
Bubble deformation:	Significant and affected by natural factors such as Buoyancy and surrounding fluid field	Constrained by a fixed point
Local fluid field during bubble rising:	Significant impact on the bubble behavior (dominated by microstructures)	Uncertain and difficult to quantify
Bubble rupture:	Influenced by interplay of bubble deformation, fluid field, microstructures long before contact, and surface feature at the moment of contact	Mainly influenced by surface feature at the moment of contact

↑
Our study scope

Fig. R2. Comparison of the mechanisms underlying bubble capture between free-rising bubbles and needle-attached/fixed bubbles.

Based on these analyses, even when free-rising and needle-fixed bubbles generate similarly slow velocities, their capture times should be compared with caution. Unlike the free-rising process, which follows fundamental hydrodynamic principles, in the needle-attached scenario, an external gas injection rate determines the bubble expansion rate, subsequently affecting the approach velocity and ultimately influencing the rupture time. It means that the rupture time is highly sensitive to the initial gas injection rate, a factor externally imposed rather than naturally controlled by fluid dynamics. Analysis on this effect would be a good point for further investigation on needle-fixed bubbles.

Building on the above, we now find it would also be better to introduce a certain separation distance between the bubble and the surface even in the scenario of free-rising bubble capture, to minimize potential external influences such as gas injection rate and needle-tip adhesion. Under typical free-rising bubble experimental conditions, it appears unlikely that the bubble

velocity would drop as low as 10^{-2} m/s (the majority of the literature reports values around 10^{-1} m/s). We remain cautious about cases in the literature that report extremely short release distances and ultra-low velocities, as well as our own supplemented data on 0.04 m/s, given that multiple external factors may come into play, and are challenging to control.

Accordingly, we feel that using bubble velocity to classify existing studies is not sufficiently rigorous. It is also challenging to define what qualifies as a fast or slow bubble. We respectfully suggest categorizing them according to how the bubbles are released. Based on this logic, our MA surface is among the best-performing surfaces under free-rising bubble conditions:

1) Attached bubbles that are outside our study scope (slow moving bubbles suggested by the reviewer) – MA: 59.6 ms vs. FH: 85.4 ms in Fig. R1c, which can not be compared with state-of-the-art in Table R1 that are all free-rising bubbles and mostly “fast” bubbles. In this experiment of our first revision, the relative velocity between the bubble and the plate was close to zero (<0.001 m/s), which is significantly lower than the 0.25–0.3 m/s in Fig. 1h of Adv. Sci. 2024, 2403366. This condition is nearly equivalent to a stationary bubble resting on the surface, where liquid film thinning is primarily driven by Van der Waals and electrical double-layer forces, resulting in a slow bubble rupture process.

2) Free-rising bubbles that fall within our study scope (fast moving bubbles suggested by the reviewer) – MA: 1-2 ms vs. FH: 38-68 ms vs. state-of-the-art: 1-145 ms. In this velocity range (0.14-0.37 m/s), no clear correlation was observed between approach velocity and capture time from both of our results and prior work.

To eliminate potential confusion, we have followed the reviewer’s suggestion by focusing on the mechanistic analysis rather than overemphasizing performance. Where necessary, we have been more cautious in presenting our findings. To further clarify the scope of our study, we have explicitly restricted it to free-rising bubbles in the title, abstract, introduction, and conclusion. Within this specific domain, we emphasize the role of microscale structures, as the effects of nanoscale structures have already been extensively discussed in previous literature.

Specifically, we have added a comparative discussion on particularly slow-moving free-rising bubbles and needle-attached bubbles in Results (Page 7 in the revised manuscript). We hope this paragraph not only highlights the significance of other bubble capture studies (Adv. Sci. 2024, 2403366) but also encourages researchers to consider the capture mechanisms of different bubble types.

6) Response 10: *“The visual method generally identifies the start time when the bubble first touches the surface and the end time when the three-phase contact line forms (typically marked by a sudden decrease in the bubble contact angle). It is more accurate than the velocity method because the change in velocity may lag behind the bubble rupture event. While this method may be subject to some error due to limited spatial resolution, the error is basically less than 1 ms. Therefore, we find that the visual method is more accurate and reliable for our study.”*

Responses to Query 10 should be included in a Supporting Discussion of some form, as this is good information (alongside Table R1) that will inform future measurements on which methods

are more “conservative”, for instance, the velocity detection algorithms.

Response:

In accordance with the reviewer's comments, we have included our reply to Query 10 into the revised SI as Supplementary Table 2.

7) Response 11: *“In this case, the lower boundary of the liquid film is the bubble surface, and there is no added medium in the water column, so the pure bubble surface can be approximated as the slip boundary.”*

Despite the use of water, it is often unclear if contamination happens enough to lead to a no slip condition. This should be appropriately discussed and justified. While the prefactor will not lead to significant influence on general phenomena or laws governing film rupture, it should also not lead to confusion.

See how the immobile boundary condition tends to be the dominant assumption (<https://journals.aps.org/prl/abstract/10.1103/PhysRevLett.122.194501>):

“Phenomenological features of the thinning film, including the inversion of curvature (dimple) and the dynamic evolution of the dimple profile, agreed well with theoretical prediction assuming the immobile boundary condition [15]. Even freshly generated bubbles with fully mobile surfaces during the rise in bulk exhibited an immobile boundary condition when colliding with a solid surface [16,17]. The discrepancy between experiments leaves a gap in researchers’ understanding of air-water interfaces.”

Response:

We appreciate the reviewer for the meticulous examination of our article. In the revised SI, we have included a discussion on the issue of boundary conditions. We elaborate that under the specific conditions of this study (i.e., no added media, free-rising bubbles), the bubble surface can be approximately considered as a mobile boundary, consistent with treatments in the current literature (Adv. Colloid Interface Sci., 2016, 235, 214-232; Langmuir, 2019, 35, 8294-8307). Additionally, we have cited relevant literature (Phys. Rev. Lett., 2012, 122, 194501) to illustrate that even though clean bubble surfaces can exhibit complete mobility during free rising in some cases, they may transition to immobile boundary conditions upon collision with solid surfaces.

Added content in SI:

“In this case, the lower boundary of the liquid film is the bubble surface, which can be approximated as the slip boundary (no added medium in the water). This approximation is based on the fact that, in a pure water environment, the gas-liquid interface cannot withstand shear stress, particularly in the absence of contaminants or surfactants. However, it should be noted that clean bubble surfaces may transition to a no-slip conditions upon collision with a solid surface⁵, suggesting that the slip boundary applicability may be constrained by specific conditions. Additionally, gas-liquid interface mobility at the thin-film level remains controversial: millisecond-scale bubble coalescence^{6,7} has been speculated to be a result of

interfacial mobility, yet direct observations continue to support the immobile boundary condition⁸. The precise description of boundary conditions in this context also requires further exploration.”

8) Response 12: *“Additionally, from an application perspective, most cases involve the capture of free-rising bubbles. If a bubble is not captured on first contact, it is likely to be carried away by the water flow, making subsequent capture unlikely. Therefore, we believe that our focus should remain on exploring the capture mechanisms that incorporate how the surface features influence the bubble-rising process, rather than on discussing the rupture of high-speed bubbles.”*

I strongly agree that the message should be retuned exactly towards this application of freely rising bubbles. The claims for universal “really-fast” bubble rupture is obviously a little different compared to what the authors have postulated in the first draft. I think we do all accept that at this point. However, there remains to be some work towards the re-drafting of the manuscript to focus on the use of freely rising bubbles while still acknowledging the existence of prior work.

Response:

We sincerely appreciate the reviewer’s acknowledgment of our revisions and constructive suggestions. We fully agree that the information in the manuscript should be more precisely focused on the scenario of free-rising bubbles. In accordance with the reviewer's suggestions, we have thoroughly revised the manuscript to place emphasis on this specific scenario while also acknowledging and citing previous research findings.

9) Response 13: *“At the moment of rupture, the effect of nanoparticles or nanostructures remains consistent across both MA and FH surfaces due to their comparable minimum feature sizes. On the other hand, during the drainage phase before rupture, the microstructure irreplaceably contributes to confining the gas to a constrained space, within which significant deformation of the gas layer can occur, thereby accelerating drainage.”*

I do agree with this revised statement only to a certain extent. The authors, again, left out the fact, that this works only with fast moving bubbles. Otherwise, MA and FH share the same performance – which is arguably an order of magnitude slower than the currently reported state-of-the-art. Throughout my initial review and my current re-review, this has always struck me as the main difference this paper has compared to prior work (Table R1). Authors should obviously check Table R1 references to compare approach velocities, and thereafter, set their work apart in the sense that freely moving rising bubbles are used – and how those bubbles can be captured. I believe that the authors would also acknowledge that without a comparative experiment using state-of-the-art surfaces, it cannot be proven if those will not work with faster moving bubbles. Therefore, some conservative assumptions should be listed to ensure the continuity of accuracy in bubble rupture literature.

Response:

Thanks for the insightful comments, which have helped us refine the scope of our study. As stated previously, the bubble type in our study (free-rising bubbles) and the velocity range (10^{-1} m/s, Fig. R1a) are consistent with prior work (Table R1). The case of ultra-slow free-rising bubbles (10^{-2} m/s, Fig. R1b) should be compared with caution, as it is more susceptible to external influences. Meanwhile, ultra-slow needle-fixed bubbles (10^{-3} m/s, Fig. R1c) fall outside the scope of our comparison.

Following the reviewer's suggestion, we have emphasized the velocity range where microstructures play a significant role in the Results section. We also provide additional clarification regarding ultra-slow free-rising bubbles:

“The extent to which the mechanism of accelerated film drainage is affected by bubble release distance and approach velocity warrants further consideration. For the free-rising bubbles discussed in both our study (approach velocity $v=0.16-0.37$ m/s) and the cases presented in Fig. 1f, the release distance generally ranges from 2.4 to 300 mm, with corresponding $v=0.14-0.37$ m/s (Supplementary Table 1). Within these comparable velocity ranges, bubbles have enough time and space to deform, generating sufficient pressure that accelerates liquid film thinning, all driven by non-contacting microstructures. When bubbles are released over a very short distance with an ultra-low approach velocity of around 0.04 m/s (Supplementary Fig. 3b), MA surface maintains a short capture time of ~ 1.7 ms, while on FH surfaces, bubbles no longer rebound, resulting in a capture time decreasing to the same order of magnitude as that on the MA surface, albeit with greater variance. However, as release distance is further reduced beyond the above range, the effectiveness of microstructures in driving drainage may diminish. Notably, ultra-short release distances have a non-negligible probability of introducing uncontrolled factors, such as gas injection speed and needle tip adhesion, necessitating cautious interpretation of these results. Another scenario involves needle-fixed bubbles^{23,41}. While these can also produce very low velocities, the bubble dynamics differ fundamentally from free-rising bubbles, making direct comparison within the same theoretical framework inappropriate. This refined needle-fixed bubble setup is particularly well-suited for analyzing instantaneous rupture upon surface contact²³.” (Page 7 in the revised manuscript)

In accordance with the reviewer's comments, we focus on clarifying mechanism rather than emphasizing performance in the revised manuscript. Compared to existing studies on free-rising bubbles, our contribution lies in uncovering how microstructures influence flow field around, affect deformation of both the bubble and gas layer, thereby accelerating film drainage and facilitating bubble capture. This perspective differs from prior works, which primarily focus on the role of nanoscale structures.

11) Response 15: *“However, during the drainage stage, microstructures are the dominant factor, primarily through their role in enhancing film drainage. While the reviewer correctly highlighted the fast bubble capture under high We or Bo number conditions as a significant contribution, our core objective has always been to uncover a novel mechanism by which microstructures accelerate bubble capture efficiency through optimizing liquid film dynamics.”*

Once again, I believe the description lacks universality. “During the drainage stage...” refers

to only high We or Bo number conditions. This does not apply as universally as the statement claims. As recommended above, the core message of the manuscript needs to be refined to reflect this “unique” condition, despite of the fact that it may in fact be the more “applied” condition. At low We or Bo number, many state-of-the-art surface (Table R1) show a clear reliance on nanotexturing, and not microtexturing. Therefore, there needs to be more care towards claims made.

Response:

We appreciate the reviewer’s thorough evaluation and valuable insights. We fully acknowledge that "drainage stage..." applies to high We or Bo number conditions rather than being universally valid. We also note that, like our study, the prior works referenced in Table R1 also largely fall within the relatively high We or Bo number range.

Based on the reviewer’s suggestion, we have carefully revised the manuscript, highlighting the velocity range in which the film drainage mechanism is at work, and how it would be influenced by the approach velocity.

12) Response 17: *“In terms of where drainage dominates, our results indicate that drainage is a combination of both macroscopic and local effects. The macroscopic aspect is mainly influenced by the overall bubble dynamics and approach speed. However, the local effects, such as bubble deformation and structure-caused local thin film, play a crucial role in determining the accurate detailed evolution of the liquid film.”*

It is important here to also include this description in the revised version of the manuscript. If the Scheludko equation does indeed fit the experiments better, that would mean that the local film drainage behavior might be dominating. Listing this clearly would help the progress and understanding of future research in the area.

“These findings suggest that the protruding features, especially microstructures, can change the film thickness distribution, thus influence the time required for the liquid film to drain.”

I am sure the authors mean “tips of microstructures”. This would be a good location to also include the actual lengthscale involved at the tip.

Response:

As pointed out by the reviewer, the Scheludko equation provides a good explanation for the experimental results, suggesting that local liquid film drainage behavior may play a dominant role in the process of bubble capture. In accordance with the reviewer’s comments, we have explicitly included this in the revised manuscript while ensuring a coherent narrative (page 9):

“ On a global scale, energy conversion within this multi-interface system, particularly the dynamic interaction among bubble, liquid film, and gas layer, serves as a critical mechanism through which periodic microconical arrays facilitate drainage and enhance bubble capture. Within each unit, this process is driven by the synergetic effects at the liquid-gas interfaces, where we isolate the correlation between deformations in entrapped gas layers and bubble

dynamics (shape and speed). At a more localized scale, the protruding features of the microstructure, as discussed in the Fig. 2f, dictate the local film thickness, thinning the liquid film at the tips and thereby influencing the drainage dynamics, ultimately affecting the time required for liquid drainage.”

Furthermore, "the protruding features" that we referred to means the entire microstructure, encompassing both the tips of the microstructures and the overall contours of the microstructure. The actual length scale involved at the tips is about 10 μm , limited by the 3D printing resolution (nanoArch P140, BMF Precision Tech Inc.). Since the bubble diameters in this study are all larger than at least one microstructural unit, it is not only the tips of the microstructures that influence the distribution of liquid film thickness; the overall contours of the microstructure also contribute to the uneven liquid film formation.

13) Response 18: *“For the microconical surface, the three-phase contact lines pinned on microstructures provide a force that can resist pressure, which can be expressed as $F\sigma=2\pi ac\sigma\cos\theta E,(S10)$ here ac is the cross-section radius of the microstructure where the three-phase contact line formed.”*

I believe that this equation is ok to use, but needs refinement as the so-termed microstructures, are no longer just “microstructural”. The presence of nanostructures will cause the contact line to be smaller than what it seems since the nanostructures themselves will only be in partial contact with the liquid – i.e. of a certain solid fraction.

Response:

We fully understand the reviewer's concern regarding the potential impact of nanostructures on the contact line length. To address this concern, we would like to further clarify our derivation logic and rationale.

Firstly, due to the hydrophobicity of the surface, the liquid does not fully wet the interior of the nanoparticles, resulting in a Cassie wetting state. Under these conditions, the morphology of the contact line does not strictly follow the topography of the nanoparticles, but instead forms a nearly circular contact line that connects the most protruding nanoparticles, as shown in Figure R3. Based on our roughness measurements (Supplementary Fig. 2), the roughness of the nanoparticle clusters is approximately 1.78 μm . To estimate the contact line length, we consider the middle cross-section of the microcone, where the diameter is 346 μm . When accounting for the nanostructures' roughness, the effective diameter is approximately $346 + 1.78*2 = 349.56 \mu\text{m}$. The corresponding circumferences are 1086.99 μm (without nanoparticales) and 1098.17 μm (with nanoparticales), respectively, indicating that the nanostructures alter the contact line length by only about 1%. This suggests that we could neglect their direct impact on the contact line length.

Fig. R3. Schematic diagram of the three phase contact line on a microcone.

Secondly, the influence of nanostructures on the hydrophobic properties of the surface has been incorporated through the introduction of the advancing contact angle (θ_{adv}). As a parameter that comprehensively reflects both surface chemical properties and morphology, θ_{adv} inherently accounts for the effect of nanostructures. Although the geometric details of nanostructures are not explicitly included in the equation, their contribution on surface hydrophobicity and contact line behavior is captured through the relationship of θ_{adv} and θ_E in the equation.

In summary, we believe that the current equation is reasonable in terms of theoretical derivation and practical application. The impact of nanostructures on surface properties has been appropriately introduced by using the advancing contact angle. We have provided a more detailed explanation in the revised SI Note 5 to enhance the discussion on the contribution of nanostructures:

“Notably, although surface nanostructures can affect the estimation of contact line length, under Cassie wetting conditions, this error is only about 1% and can therefore be neglected. We have considered the influence of nanostructures on surface hydrophobicity by introducing the advancing contact angle (θ_{adv}). The θ_{adv} is a parameter that comprehensively reflects both the surface chemical properties and morphology. The impact of nanostructures on surface hydrophobicity and contact line behavior is indirectly reflected through the effect of θ_{adv} on the θ_E value in Equation S12.”

Conclusions: In general, I believe the authors have discovered a niche area that is worth exploring and publishing their findings on. This centres around fast moving millimetric

bubbles. While it was not originally helpful to Reviewers that the claims were made around “simply” microtexturing, I believe the findings around approach velocity still bears merit. Looking through the key parts of the manuscript, I was still able to find remnant statements from the original draft that could be construed as misleading without stating these primary difference:

(1) Nanostructuring should be clearly illustrated. (2) Approach velocities that show the

differences in performance between MA and FH should be clearly stated. The performance is clearly not universal for all bubble approach velocities since they are much slower when the bubble is approached slowly (Figure R2b).

Abstract:

“Current super-aerophilic surface designs often emphasize complex nanostructures, while the fundamental role of microtextures in first-contact bubble capture has not been well understood.”

- True likely only for fast bubble approach velocities.

“The simply microtextured surface reported in the present work supports...”

- Again, this detail should be revised to not be microtextured only.

- Clearly the nanostructuring helps in stabilization of the surface.

Introduction:

- The investigated range of bubble approach velocity is still missing from the Introduction section. This must be added to help improve the context of the findings.

- “Compared to silane functionalized micro-pillars in 20 μm , as an example, the capture time of a fixed bubble decreases modestly from ~ 14 ms to ~ 5 ms on the hierarchical nanostructured surface. On the other hand, hydrophobic microprotrusions can capture free bubbles an order of magnitude faster than the nanoparticle-decorated surfaces.”

Conclusions:

“We underscore that rapid bubble capture does not rely solely on complex and finely detailed nanostructures.”

- Not true for different bubble approach velocities.

“These findings, together with our proposed mechanism of microstructures that facilitate accelerated film drainage, collectively offer a fundamental understanding of the ultrafast bubble capture process.”

- Only true under high approach velocity.

Response:

We thank the reviewer’s recognition of our efforts and the detailed guidance for improving the manuscript. We fully understand the concerns raised and have carefully revised the manuscript to ensure the accuracy and clarity of our findings. Below are our specific responses and revisions:

1. We have strengthened the description of the nanostructure throughout the text, particularly in the Title, Abstract, Results, and Conclusion, emphasizing the crucial role of nano/micro-structures, rather than purely microstructures, in achieving fast capture. For instance, we have changed "simply microtextured surface" to "micro-nano structured surface" in the abstract to accurately reflect the surface characteristics.

2. In the "Enhanced water film drainage within a characteristic unit" section of the results, we have expanded a discussion on the performance of MA and FH surfaces at different approaching velocities, particularly comparing three scenarios: free-rising bubbles with approach velocities between 0.14 and 0.37 m/s, free-rising bubbles with ultra-low approach velocity, and needle-fixed bubbles with ultra-low speed. We have emphasized that in the second scenario, the results should be considered with a degree of caution, as multiple external factors may introduce variability that is challenging to control for.
3. We have carefully reviewed the entire manuscript to ensure that all potentially misleading expressions have been corrected. We have clearly stated that the microstructures play a significant role in the scenario of free-rising bubbles at high speeds. These additions will help readers better understand the scope and limitations of our study.

We believe these revisions significantly enhance the accuracy and rationality of the manuscript, while also addressing the reviewer's concerns.

We have made the following revisions on the pointed sentences (along with additional modifications throughout the manuscript):

Abstract:

“While the role of nanostructures during the collision of free-rising bubbles with super-aerophilic surfaces is well established, the fundamental contribution of microtextures in promoting initial capture, even before contact, has yet to be fully understood.”

“The present nano/microstructured surface supports...”

Introduction:

- We find it inappropriate to compare needle-attached and free-rising bubble here, so we only keep:

“An opposing trend, even, exists where hydrophobic microprotrusions can capture free-rising bubbles (approach velocity is ca. 0.35 m/s) an order of magnitude faster than the nanoparticle-decorated surfaces”

and highlight the work of *Adv Sci* 11, 2403366, 2024 in Results:

“Another scenario involves needle-fixed bubbles^{23,41}. While these can also produce very low velocities, their dynamics differ fundamentally from free-rising bubbles, making direct comparison within the same theoretical framework inappropriate. This refined needle-fixed bubble setup is particularly well-suited for analyzing instantaneous rupture upon surface contact²³.”

Conclusions:

“We underscore that rapid capture of free-rising bubbles does not rely solely on complex and finely detailed nanostructures.”

“These findings, together with our proposed mechanism of microstructures that facilitate

accelerated film drainage, collectively offer a fundamental understanding of the ultrafast capture process of free-rising bubbles.”

Reviewer: 2

The authors have produced a noteworthy manuscript on the impacts of microtexture on bubble capture underwater. This work has clear significance and displays an impressive quantification of the gas layer and interface profiles.

While I had some concerns with their writing regarding the relative contributions of micro- and nanotexture, they have since both moderated the tone and conducted additional rounds of experimentation. Given that, I find no further major issues with this publication.

Response: We greatly appreciate the reviewer's constructive feedback, which has helped strengthen our work. We are delighted and grateful for the reviewer's recognition of our manuscript.

Reviewer: 3

I am satisfied with the responses given by the authors to the comments in my first report. The paper can be published as it is.

Response: We are delighted that our responses have addressed your concerns and appreciate your recommendation for publication. We sincerely thank the reviewer for the insightful comments and constructive suggestions, which have greatly helped improve our manuscript.

Reviewer: 4

I have reviewed the point-by-point responses and the revised version of the manuscript by Hu et al. and carefully considered the points raised by all four reviewers, with particular attention to the comments made by reviewer 4. The authors have satisfactorily addressed the concerns raised by reviewer 4 and appear to have responded comprehensively to the feedback from the other reviewers as well. Collectively, these revisions have significantly improved the manuscript, and I now believe it meets the standards for publication in *Nature Communications*.

Response: We are grateful for the reviewer's recommendation for publication in *Nature Communications*. The reviewer's exceptionally detailed and constructive comments have been invaluable in strengthening and refining our manuscript, and we are truly grateful for your time and effort dedicated to this review.

However, I would like to highlight two minor issues:

- 1) There appears to be a grammatical issue in the revised abstract: 'surpassing the available most minimum capture times.' I suggest revising this for improved clarity.

Response: We thank the reviewer for pointing out this grammatical issue. For this specific statement, we find it to be too universal-sounding and have therefore deleted it. We also

carefully reviewed the manuscript to address all the grammatical issues.

2) The figure titled 'Fig. R10. Continuous bubble capture on the tilted MA surface' is missing from the revised supplementary information. Please ensure that it is included for completeness and clarity.

Response: Fig. R10 has now been included in the revised SI as supplementary Fig. 5 for completeness and clarity.

Finally, I would like to acknowledge the authors' considerable effort in addressing the reviewers' comments and strengthening the manuscript. I also encourage the authors, in future work, to revisit their statement: "We feel it's unrealistic to design a structure that can capture all bubble sizes." Exploring this challenge could lead to exciting new opportunities in this field.

Response: We truly appreciate the reviewer's thoughtful recognition of our efforts in revising the manuscript. We definitely agree that designing a structure capable of capturing all bubble sizes is an exciting direction for future work. It could potentially lead to more fundamental solutions for the bubble capture, which are certainly worth exploring.